# Boosting classical and quantum nonlinear processes in ultrathin van der Waals materials

Xiaodan Lyu[1,2,12], Leevi Kallioniemi [1,12], Hongbing Cai [3], Liheng An[1], Ruihuan Duan [4], Shuin Jian Wu [5], Qinghai Tan [6], Chusheng Zhang[1], Ruihua He[7], Yansong Miao [7], Zheng Liu [4], Alexander Ling[5,8], Jesus Zúñiga-Perez [1,2] ✉ & Weibo Gao [1,2,9,10,11] ✉

Understanding and controlling nonlinear processes is crucial for engineering light-matter interaction and generating non-classical light. A significant challenge in ultra-thin nonlinear materials is the marked diminution of the nonlinear conversion efficiency due to the reduced light-matter interaction length and, in many cases, the centrosymmetric crystalline structures. Here we relax these limitations and report a giant boost of classical and quantum nonlinear processes in ultrathin van der Waals materials. Specifically, with a metal-nonlinear material heterostructure we enhance classical second-harmonic generation in h-BN flakes by two orders of magnitude. Moreover, we have engineered a metal-$SiO_2$-nonlinear material heterostructure resulting in a remarkable two orders of magnitude augmentation of the quantum spontaneous parametric down-conversion (SPDC) in $NbOCl_2$ flakes. Notably, we demonstrate SPDC in a 16 nm-thick $NbOCl_2$ flake integrated into the proposed structure. These findings simplify on-chip quantum state engineering and accelerate the use of van der Waals materials in nonlinear optoelectronics.

The integration of active photonic materials such as III-V semiconductors onto complementary metal-oxide semiconductor (CMOS) and dielectric photonic circuits has faced longstanding challenges. In this context, the ease of exfoliation and transfer of van der Waals materials onto nearly any material platform has driven the development of integrated photonic devices based thereon. The use of van der Waals materials as integrated nonlinear devices began more than 10 years ago when graphene was exploited for fabricating modulators and photodetectors[1-3]. More recently, the advent of transition metal dichalcogenides (TMDs) has further enriched this field. Nonlinear processes have been demonstrated in TMDs[4-9], thus making it possible to realize ultrafast optical switching[10,11]. Beyond classical nonlinear processes, van der Waals materials have been reported to facilitate optical parametric amplification and oscillation[12,13], and spontaneous parametric down-conversion (SPDC)[14,15]. Notably, the demonstration of SPDC in two-dimensional layered materials offers significant

[1]Division of Physics and Applied Physics, School of Physical and Mathematical Sciences, Nanyang Technological University, Singapore, Singapore. [2]Majulab, International Research Laboratory IRL 3654, CNRS, Université Côte d'Azur, Sorbonne Université, National University of Singapore, Nanyang Technological University, Singapore, Singapore. [3]Department of Physics, University of Science and Technology of China, Hefei, China. [4]School of Materials Science and Engineering, Nanyang Technological University, Singapore, Singapore. [5]Centre for Quantum Technologies, National University of Singapore, Singapore, Singapore. [6]School of Microelectronics, University of Science and Technology of China, Hefei, China. [7]School of Biological Sciences, Nanyang Technological University, Singapore, Singapore. [8]Department of Physics, National University of Singapore, Singapore, Singapore. [9]School of Electrical and Electronic Engineering, Nanyang Technological University, Singapore, Singapore. [10]Quantum Science and Engineering Centre (QSec), Nanyang Technological University, Singapore, Singapore. [11]Centre for Quantum Technologies, Nanyang Technological University, Singapore, Singapore. [12]These authors contributed equally: Xiaodan Lyu, Leevi Kallioniemi. ✉e-mail: jesus.zuniga@ntu.edu.sg; wbgao@ntu.edu.sg

advantages over bulk nonlinear materials both in terms of integration, due to their bond-free attachment to the underlying substrates[16], as well as in terms of relaxed phase-matching conditions, which need not be satisfied for subwavelength nonlinear active regions[17,18].

Despite the significant interest in ultra-thin van der Waals materials for nonlinear applications, their small nonlinear conversion efficiency remains the strongest limiting factor. The reasons are the small interaction length inherent to significantly subwavelength nonlinear materials and, in many cases, the centrosymmetric crystalline structure, which constrains their second-order nonlinear susceptibility to be zero. This is particularly the case in TMDs and h-BN flakes formed by an even number of monolayers[4,19,20]. Furthermore, even for some noncentrosymmetric van der Waals materials, such as TMDs containing odd layers, a decrease in nonlinearity with increasing thickness is often observed, which is ascribed to the significant modification of the electronic structure driven by strong interlayer electronic coupling and dielectric screening [4].

Recently, efforts have been made to enhance the nonlinear effects in ultra-thin van der Waals materials either by breaking the crystal inversion symmetry through twisted flakes or by coupling them to photonic structures. In the case of the twisted flakes approach[21–23] one must resort to intricate flakes manipulations (e.g., the use of micro rotators), which is complex and renders the approach difficult to implement and incompatible with other basic functionalities. In the case of photonic resonators, two options can be considered. First, one can employ low cavity quality factor (Q) photonic structures, including metasurfaces and dielectric resonators, which enable a moderate enhancement but rather broadband operation[24–29]. Alternatively, one can resort to high Q resonant cavities that allow, potentially, for larger enhancements but sacrifice the broad bandwidth due to their resonant character[30,31]. This spectrally narrow enhancement prevents exploiting the full potentiality of subwavelength light sources, which can display a broad emission bandwidth as a result of the relaxed phase matching condition[14]. Thus, ideally one would like to merge the advantages of both approaches without necessitating complicated clean-room processes.

Herein we develop an approach to enhance nonlinear processes in ultra-thin materials via the engineering of the field distribution around the nonlinear material. This approach is independent of the material symmetry and, thus, of general applicability. We utilize nonlinear material/metal and nonlinear material/dielectric/metal heterostructure configurations to modify appropriately the electric field distribution at the van der Waals materials position. Our heterostructures play thus a double role: to enhance the intensity of the electric field while keeping a broadband response, similar to low Q cavities, and to enlarge the electric field gradient, which acts a second sizeable source of polarization. This second ingredient is essential to achieve a comparable nonlinear response between thin films with odd and even layers in materials with AA' stacking. To illustrate the general applicability of our heterostructures we chose two van der Waals materials: hBN, which is a widespread 2D material, whose centrosymmetric or non-centrosymmetric character is layer number dependent, and NbOCl$_2$, which is a van der Waals material exhibiting one of the largest nonlinear responses to date (Supplementary Table 2). We observe a giant enhancement of second harmonic intensity on both h-BN, i.e., from some nanometers to tens of nanometer-thick layers, and NbOCl$_2$ flakes. Leveraging nonlinear transfer matrix simulations[23,32] we have evaluated quantitatively the SH intensity on different heterostructures and substrates, getting insight into the interplay between dipolar and quadrupolar polarization contributions. Interestingly, for the SHG in h-BN layers, the interplay between the two contributions results in a nonlinear intensity almost independent of the monolayers parity, erasing thereby one practical limitation. The nonlinear response from the developed structure is at least one order of magnitude (typically about 50 times) larger than on standard SiO$_2$/Si

substrates and the observed enhancement effect is broadband, covering a wide range of pump laser wavelengths. This characteristic is particularly advantageous for ultra-thin van der Waals biphoton sources such as NbOCl$_2$, for which photon pairs exhibit a broad spectrum owing to relaxed phase-matching conditions[17,33]. Finally, we propose and use an easy-to-implement SiO$_2$/Au planar structure that enables to enhance the nonlinear contributions in very thin (~20 nm or less) nonlinear materials. With this method, we achieve an impressive 3 orders of magnitude SHG enhancement in comparison with the conventional flakes on wafer configuration for h-BN layers with thicknesses below 10 nm. When employed to enhance photon pair generation in NbOCl$_2$, the metal/dielectric heterostructure enables us to reduce the NbOCl$_2$ thickness down to 16 nm. This constitutes the realization of SPDC with one of the thinnest nonlinear media among currently reported SPDC sources (Supplementary Table 4).

## Results

### General strategy for enhancing SHG in thin materials

The nonlinear optical processes in a material are governed by its electrical polarization, which is given by a power series of the electric field amplitude. Beyond the linear term [34]:

$$P_{NLO} = P_{2\omega} + P_{3\omega} + \ldots = \chi^{(2)} E_\omega E_\omega + \chi^{(3)} E_\omega E_\omega E_\omega + \cdots,$$

where $\chi^{(2)}$ is the second-order susceptibility tensor. $\chi^{(2)}$ is at the basis of SHG, which not only holds potential for creating nonlinear devices but also is being employed extensively by the 2D community for non-destructive lattice orientation identification using low pump powers[4,35,36]. In the classical SHG process, two photons of frequency $\omega$ interact inside the nonlinear material to give a single photon of frequency $2\omega$. As indicated above, the SHG intensity is related to the pump electric field by the second-order susceptibility tensor that, in turn, can be expanded in terms of its dipole and higher multipole moments. To leading order, we can thus write $P_{2\omega} = \chi_d^{(2)} : E_\omega E_\omega + \chi_q^{(2)} : E_\omega \nabla E_\omega + \cdots$, where $\chi_d^{(2)}$ and $\chi_q^{(2)}$ represent the dipolar and quadrupolar moments of $\chi^{(2)}$[23,37]. Note that the dipolar term couples only to the electric field amplitude (in fact, to the electric field intensity), while the quadrupolar term couples to the electric field amplitude and its gradient (i.e., the spatial variation of the electric field). For a more detailed analysis of this expansion and the quantitative relationship between the dipole and quadrupole moments see note "Dipole moment and Quadrupole moment" in the Supplementary Information. Previous studies have demonstrated quadrupolar enhancement of SHG by using optical dressing with an in-plane photon wave vector (i.e., by breaking inversion symmetry by oblique incidence excitation)[38] and by a two-beam SHG technique on a rough metal surface, on which unequal retardation effects for each beam breaks the initial symmetry [39].

Our strategy consists in finding simple designs that maximize the tradeoff between the electric field amplitude at the nonlinear-material location and the gradient of the electric field at that same position, as illustrated in Fig. 1a. To do so we exploit two well-known facts of metals: first, they display a large reflectivity over a broad wavelength range in the visible and infrared (IR), comparable to Bragg mirrors with few number of pairs or with small refractive index contrast between the Bragg materials (i.e., those forming low Q Fabry-Perot cavities)[40]; second, by imposing a near-zero electric field at their surface, they enable the creation of a strong electric field gradient near the metal, far exceeding for example the field gradient established in thick h-BN or NbOCl$_2$ by residual below-bandgap absorption.

To illustrate quantitatively the enhancement of both magnitudes, in Fig. 1b (left axis) we have plotted the electric field intensity inside 100 nm-thick h-BN deposited on gold and on standard SiO$_2$/Si wafers. Due to the relatively large reflectivity of the h-BN/gold interface compared to the h-BN/SiO$_2$ interface (Supplementary Table 5), the

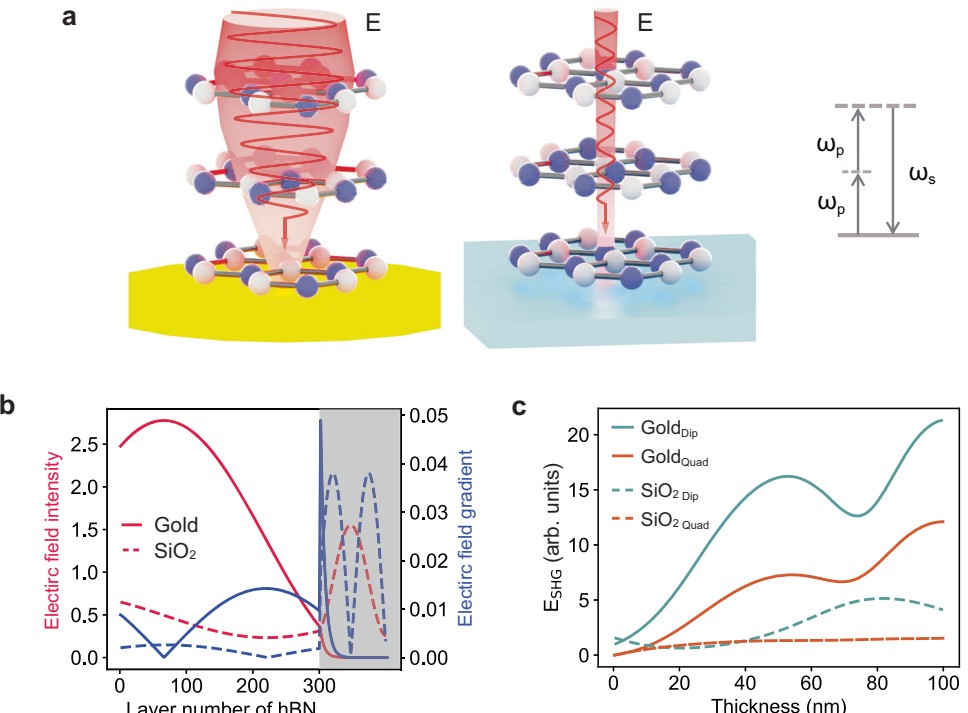

**Fig. 1 | Electric field distribution in centrosymmetric-nonlinear material/metal heterostructures. a** Schematic of incident light distribution on gold film (left) and SiO$_2$/Si substrate (right), showing a magnified electric field amplitude and gradient on the gold film. Inset: Illustration of the second harmonic generation (SHG) process, where two photons with frequency $\omega_p$ are combined to generate a single signal photon with frequency $\omega_s$. **b** Simulation results of the magnitude of pump field intensity (red) and of the gradient of electric field (blue) in a 300-monolayer hexagonal boron nitride (h-BN) flake on gold (line) and SiO$_2$/Si (dashed line). The dark area is the substrate. **c** The simulation results of dipolar (green) and quadrupolar (orange) contributions to the magnitude of SHG electric field of h-BN flakes with odd number of monolayers on gold (solid lines) as well as SiO$_2$/Si (dashed lines). Dip: dipolar; Quad: quadrupolar.

electric field intensity (at the pump wavelength) increases by a factor of 3-5, providing a stronger dipolar contribution to the nonlinear polarization. Interestingly, the electric field gradient also becomes larger when considering h-BN deposited on gold (right axis in Fig. 1b), with an increase that can amount up to a factor of ten compared to h-BN/SiO$_2$. This increased gradient, combined with the enhanced electric field amplitude, magnifies the quadrupolar contribution to the nonlinear polarization.

Nevertheless, one can wonder how important the quadrupolar contribution can be, in absolute terms, compared to the dipolar one. In centrosymmetric materials, for which the dipolar contribution vanishes due to symmetry constraints, the quadrupolar contribution is obviously the dominant one, and it enables the observation of SHG in sufficiently thick materials across which a field amplitude gradient can be established. On the other hand, in non-centrosymmetric materials, the dipolar contribution is in general considered to be the dominant one. To illustrate the actual magnitude of the quadrupolar contribution enabled by the use of metals, we used the nonlinear transfer matrix approach[23] to calculate the individual contributions to the SHG intensity of h-BN displaying an odd number of monolayers (i.e., for a non-centrosymmetric BN) deposited on gold. Figure 1c indicates that, when using gold as a substrate, the quadrupolar response in h-BN is just a factor of 2 to 3 smaller than the dipolar response. The fact that both contributions are of the same order of magnitude highlights the potential effect of the quadrupolar moment in modifying the nonlinear response of a given material, particularly of centrosymmetric materials, where the quadrupolar contribution dominates. Furthermore, because the global enhancement of dipolar and quadrupolar contributions described above stems from two general metal properties—high reflectivity over a broad wavelength range and the ability to generate a strong electric field gradient near the metal—its effect is

observed when depositing the nonlinear active material on a variety of metals (Supplementary Figs. 1, 16 and 17).

## Enhancing SHG of h-BN on metals and polarization properties

To illustrate the benefits of our simple heterostructure for SHG, we transferred h-BN flakes of different thicknesses onto SiO$_2$/Si substrates that were partially coated with 200 nm (Au)/6 nm (Ti) films (Supplementary Fig. 3). Note that Ti is used here to facilitate metal adhesion and plays no particular role in terms of the optical design. A femtosecond-pulsed laser with tunable wavelength around 890 nm was used to conduct the nonlinear measurements on h-BN. More detailed description of the setup can be found in the Methods Section.

Figure 2(a) displays the SHG response of h-BN thin films calculated thanks to the nonlinear transfer matrix formalism (Section "Nonlinear transfer matrix method" in Supplementary Information) as a function of h-BN thickness, for odd (solid) and even (dashed) number of monolayers h-BN, on gold and SiO$_2$/Si substrates. The one to two orders of magnitude enhancement for even number of monolayers h-BN can be predominantly ascribed to the enhancement of both the amplitude of the electric field, due to an increased reflectivity at the h-BN/gold interface with respect to h-BN/SiO$_2$ interface, and its gradient, which both contribute to a larger quadrupolar response. On the other hand, for odd number of monolayers, the potential enhancement if only the dipolar contribution was considered would be typically one to two order of magnitude (attaining its maximum for an h-BN thickness of about 30 nm, for which the dipolar response on SiO$_2$/Si approaches zero). However, due to the out-of-phase quadrupolar contribution (see Supplementary Fig. 2), the overall enhancement is rather a factor 20–30.

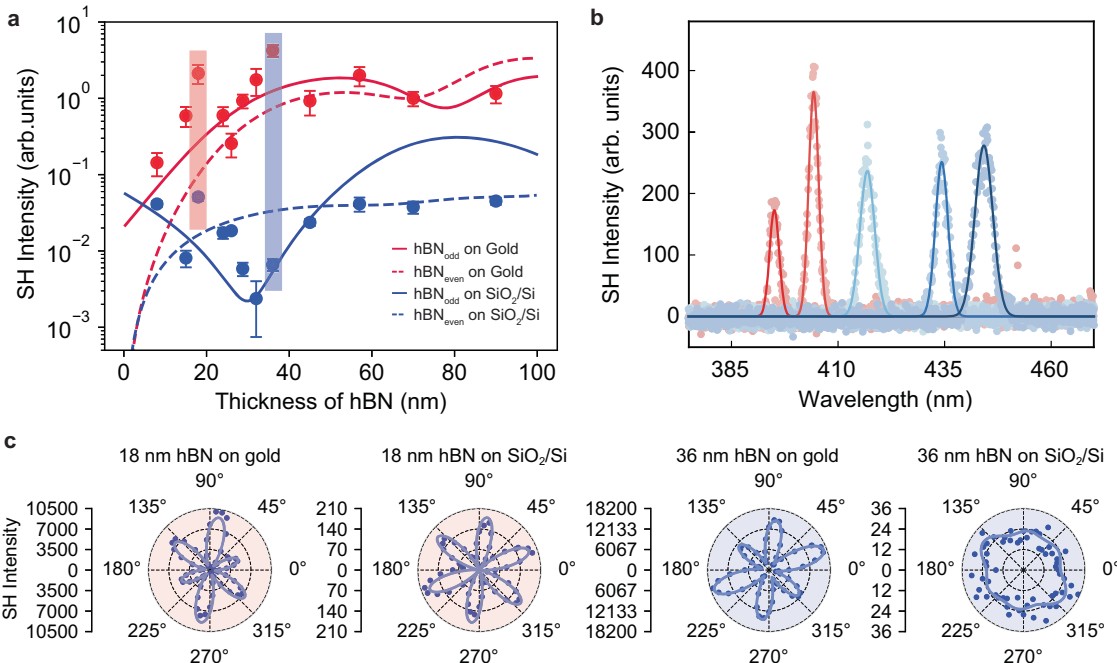

**Fig. 2 | SHG enhancement in centrosymmetric-nonlinear material/metal heterostructures. a** The experimental and simulated SH intensity for h-BN with different thicknesses on gold films and SiO$_2$/Si substrates. Symbols are experimental points while solid and dashed lines are nonlinear transfer matrix simulation results. Error bars represent the standard deviation of SHG values over the six maxima in the polarization-resolved measurements. The pump field in the simulation is 890 nm. All data are normalized to the value for 70 nm h-BN on gold. **b** SHG of h-BN on gold film under variable pumping wavelengths at 790 nm, 808 nm, 834 nm, 868 nm, 890 nm. Solid lines represent Lorentzian fits to the SHG signal spectrum. **c** Polar plot of h-BN SHG intensities as a function of the polarization angle of the incident laser on the same 18 nm (left) and 36 nm thick h-BN flakes (right) transferred onto a gold film and onto a SiO$_2$/Si substrate, respectively. The corresponding experimental points are highlighted by shadowed regions in (**a**), where we assume that both flakes have an odd-layer structure. Solid lines represent fits to the polarization-dependent SHG response under strain.

Thus, to enhance the SHG response of even number of monolayers h-BN (i.e., centrosymmetric) by one to two orders of magnitude, we need to sacrifice a factor of ~2-4 in the SHG signal of odd number of monolayers h-BN (i.e. non-centrosymmetric). Importantly, the enhancement of both contributions on gold substrates results in two SHG intensity curves (Fig. 2a, red curves) that are almost parallel to each other and that differ at most by a factor of 2 for thicknesses larger than 15 nm, erasing thereby the strong dependence of monolayer parity observed on SiO$_2$/Si wafers (Fig. 2a, blue curves).

The experimental data in Fig. 2a, while showing a large dispersion, confirm that employing gold as a substrate systematically enhances the SHG intensity of h-BN by approximately a factor of ~10. For thicknesses around 30 nm, this enhancement can exceed two orders of magnitude. More importantly, the SHG enhancement exhibits a more than 100 nm wide spectral range based on different excitation wavelengths, with SHG ranging from 395 nm to 445 nm, as illustrated in Fig. 2b. This broad spectral range results from the broadband metal reflectivity at visible and IR wavelengths, and from the electric field amplitude being zero close to the metal surface, imposing thereby a strong field amplitude gradient for all considered wavelengths.

To further ensure that the SHG response arises from the h-BN, we performed polarization-dependent SHG measurements, which exhibit a distinct six-fold symmetry (Fig. 2c), in agreement with the D$_{3h}$ point group of h-BN and some TMDs[4,21,41]. To discard any potential contribution to the SHG signal from the metal surface nonlinearity, SHG was recorded under the same conditions from bare gold. The SHG response from the bare metal is about 20 and 80 times smaller than the SHG signal from h-BN/gold heterostructures with h-BN thicknesses of 36 nm and 65 nm, respectively, as shown by polarization-resolved measurements on these two h-BN flakes at different locations

(Supplementary Figs. 4–6). These findings corroborate that the measured SHG signal indeed originates mostly from the h-BN flakes. Furthermore, since the measurements on SiO$_2$/Si and gold substrates were performed on exactly the same h-BN flakes (Supplementary Fig. 3), the azimuthal orientation of the maxima/minima in the polarization-resolved measurements for a given thickness coincide on both substrates (Fig. 2c), as dictated by the common crystallographic orientation with respect to the polarized incident laser. Noticeably, the polar plot of the SHG is fully symmetric on the flat SiO$_2$/Si substrate while an asymmetry is observed systematically on gold films. This asymmetry degree varies, however, from location to location within the same flake (Supplementary Figs. 4 and 5). This observation is tentatively ascribed to the inhomogeneous strain induced by the irregularities of the gold film whose roughness, while in the nanometer scale, is five times larger than that of SiO$_2$/Si substrates (root mean square roughness of 2.4 nm Vs 0.5 nm, respectively). To minimize the effect of the polarization-resolved SHG response asymmetry of h-BN on gold, each experimental point in Fig. 2a corresponds to the SHG value averaged over the six maxima in the polarization-resolved measurements, both for data on gold and on SiO$_2$/Si substrates.

Overall, the results in Fig. 2 confirm that the SHG intensity of h-BN on gold originates from the h-BN and that it is enhanced by one to two orders of magnitude compared to h-BN on SiO$_2$/Si thanks to the interplay between enhanced dipolar and quadrupolar contributions.

## Compatibility of SHG metal and 2D twisting enhancements

To further tailor the nonlinear response and symmetry breaking of h-BN, we combine our material-on-metal approach with twist engineering, which is known to break inversion symmetry by physically rotating h-BN sections with respect to each other. To do so we investigated the SHG of h-BN homostructures with two different stacking

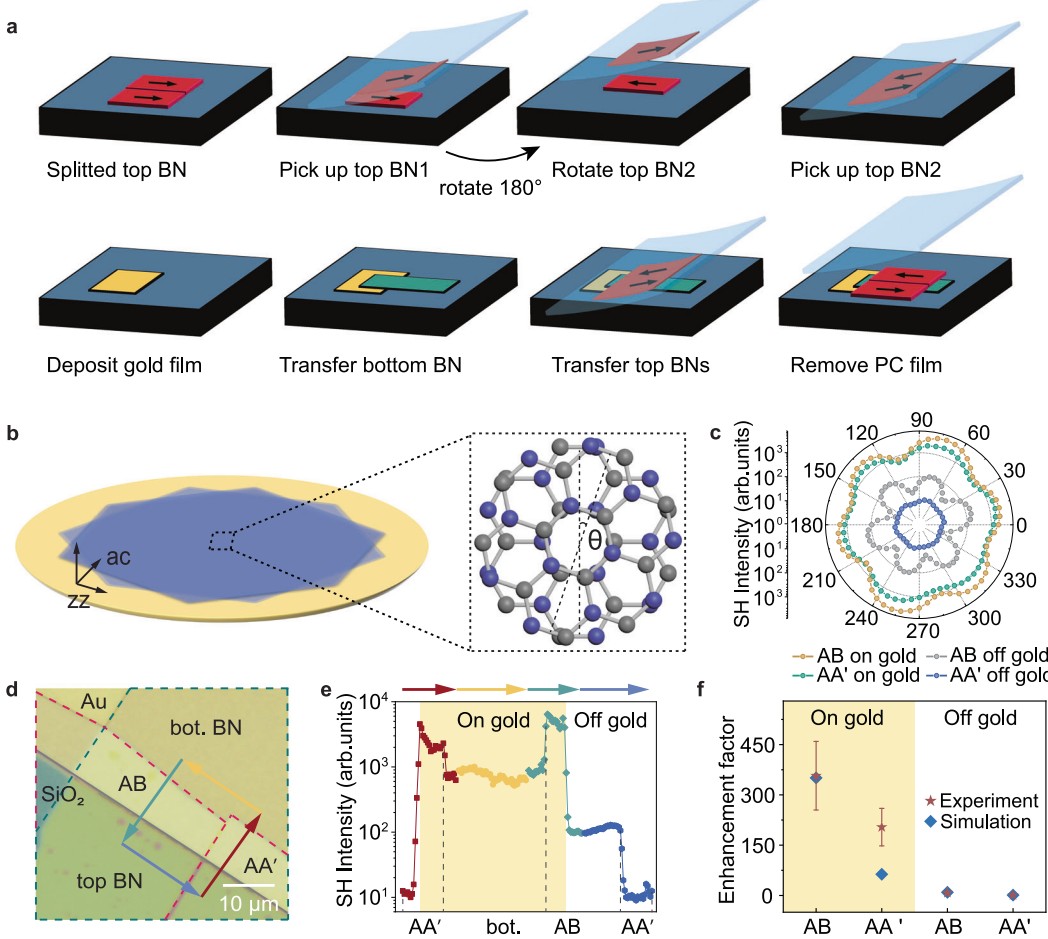

**Fig. 3 | SHG enhancement in twisted h-BN on gold film. a** Detailed process for preparing the h-BN homostructure. It illustrates the step-by-step method used to assemble the h-BN homostructure. PC: polycarbonate. **b** Schematic of h-BN homostructures (blue) on gold film (yellow). A homostructure is formed by rotating the top h-BN flake relative to the bottom one by an angle. Inset: Rotation of h-BN lattice with a θ angle. **c** The experimental results of SHG intensity as a function of polarization angle for the homostructures on gold and SiO₂/Si; AA' stack corresponds to 0° and AB stack corresponds to 60° stacking angle. **d** Bright field optical microscopy image of h-BN homostructures. Red dashed line: top layer h-BN. Green dashed line: bottom layer h-BN. **e** SHG intensity along a closed loop including all the h-BN homostructures (AA' and AB)/substrates(gold and SiO₂/Si) combinations. Different combinations are represented by unique colors and symbols, each corresponding to the path depicted by a line of the same color in (**d**). The dashed lines mark the boundaries between different regions. **f** Summary of the enhancement factors from different areas in (**d**) with respect to the 0° homostructure on SiO₂/Si substrates, measured at a 72° polarization angle: AB (60°) homostructure on gold; AA' (0°) homostructure on gold; AB (60°) homostructure on SiO₂/Si; AA' (0°) homostructure on SiO₂/Si. The error bars represent the standard deviation of the enhancement factor within each region.

angles on gold and on SiO₂/Si substrates (Fig. 3a–f). Specifically, we selected a 49 nm thick h-BN flake and separated it into two pieces. One piece of 49 nm h-BN layer was picked up while the other piece of 49 nm h-BN layer, left on the initial substrate, was rotated by 180° (equivalently, 60°) before stacking it side by side with the unrotated one. This pair of rotated h-BN layers was subsequently and simultaneously transferred on top of a third 46 nm h-BN layer, which had a rotation of 0° with respect to the two top initial layers (Fig. 3a). In this way, h-BN homostructures with 0°- and 60°-rotation interfaces can be compared within the same sample and under exactly the same experimental conditions, as evidenced by the polarization-dependent SHG (Fig. 3c).

The presence of an additional dipolar SHG signal arising from the h-BN interface at which the local inversion symmetry is broken by the layers twist[23] leads to a notable increase of SHG intensity, magnified by approximately 357 times, compared to the off-gold AA' stack (Fig. 3e and f). Note that the value of the enhancement depends on the location of the non-centrosymmetric interface within the entire structure (see numerical simulations in Supplementary Figs. 7 and 8), and can be engineered to accumulate in-phase the individual effects of

subsequent AB interfaces distributed along the stack thickness[22,23]. Besides, an optimum choice of the top and bottom h-BN pieces parity for the actual h-BN flakes would result in a further improvement (18 times more for 95 nm h-BN flakes, see Supplementary Fig. 9b, d and f).

In addition to the enhanced bulk effects from dipolar and quadrupolar contributions -arising due to the increased electric field magnitude and field gradient-, the additional twisted interface-related SHG further amplifies the overall SHG response. This mechanism is certainly interesting and an additional source of SHG enhancement compatible with our approach. However, it is also much more complicated, which highlights our approach of substrate engineering as a powerful and simple way of enhancing the nonlinear signals from van der Waals integrated materials.

## SPDC enhancement in nonlinear-material/metal structures

In ultrathin materials (largely subwavelength), phase-matching constraints can be assumed to be relaxed and, thus, one can expect SPDC to provide pairs of photons spreading over a broad wavelength range[14,17,18]. Furthermore, under certain conditions a correspondence

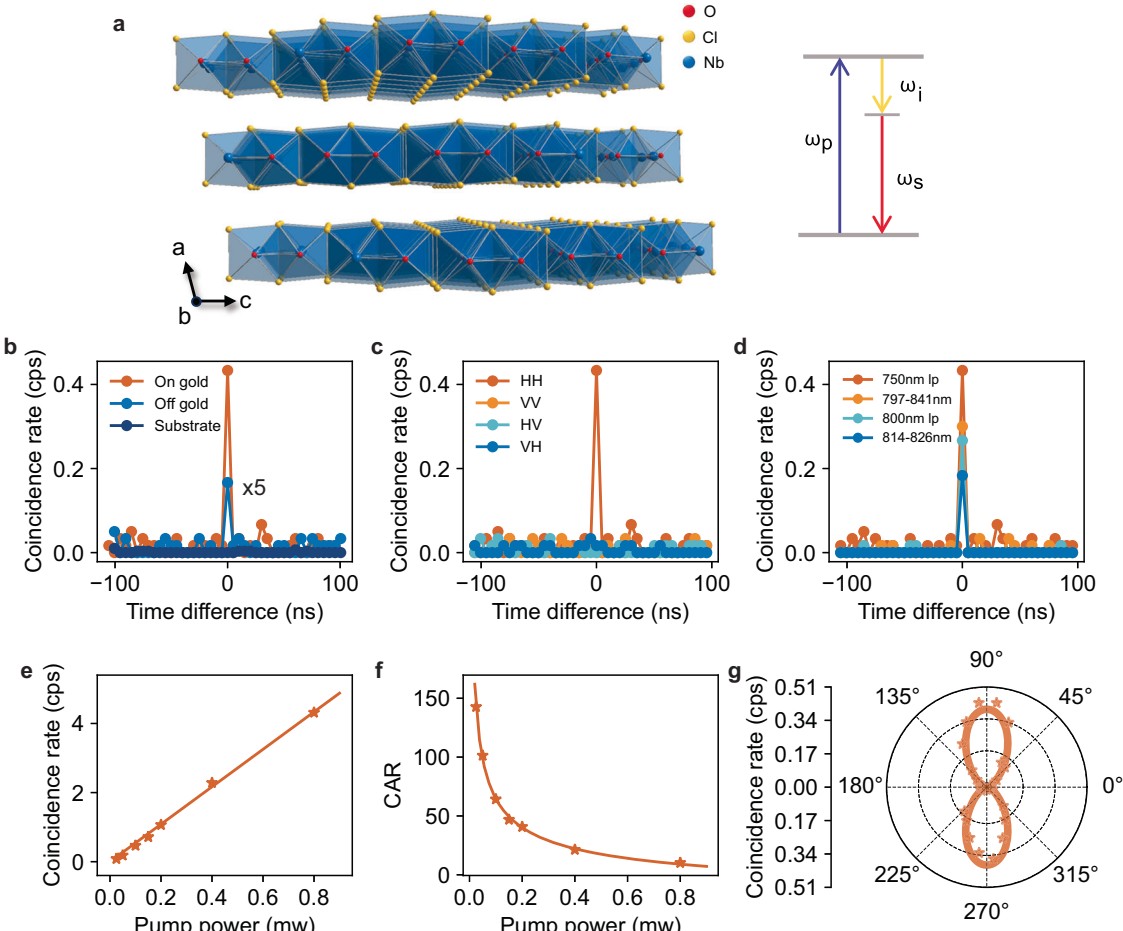

**Fig. 4 | Characterization of photon pair generation from NbOCl₂/metal heterostructures. a** Schematic drawing of the crystal structure of $NbOCl_2$, denoting the polar axis b, nonpolar axis c, and out-of-plane axis a. Red, oxygen atoms; yellow, chlorine atoms; blue, niobium atoms., Inset: An illustration of the quantum spontaneous parametric down conversion (SPDC) process, in which a photon ($\omega_p$) incident upon a nonlinear crystal is spontaneously converted into signal and idler photons of lower frequency ($\omega_i$ and $\omega_s$). **b** Coincidence counts rate from 275 nm thick $NbOCl_2$ on gold (orange), $NbOCl_2$ on $SiO_2/Si$ (blue) and the bare $SiO_2/Si$ substrate (dark blue). **c** Coincidence rate in H-V basis from $NbOCl_2$ on gold substrate. H-axis is defined as the polar axis (b-axis), and the fundamental wave is also polarized along the H-axis. **d** Coincidence count rate from $NbOCl_2$ on gold film with different wavelength integration ranges. lp long pass. **e** Power dependent coincidence counts rate from $NbOCl_2$ on gold film, showing a linear response. The solid line represents the fitted curve. **f** Power dependent coincidence-to-accidental ratio (CAR) from $NbOCl_2$ on gold film, indicating that a clear correlation peak above the classical limit (CAR>2) is obtained from the sample. The solid line represents the fitted curve, which exhibits an inverse relationship. The results in (**d**–**f**) are measured under co-polarized configuration. **g** Coincidence count rate as a function of the fundamental wave polarization in the co-polarized detection configuration. Points are experimental results, and the broad line is a theoretical fitting.

between sum-frequency generation (SFG) and SPDC intensity can be established[42], meaning that the larger the SFG intensity is, the larger the SPDC rate one can expect. For these two reasons, our material-on-metal approach based on broadband enhancement of SHG should be of use with subwavelength SPDC nonlinear materials.

To illustrate the potential of our approach we used $NbOCl_2$ as the nonlinear van der Waals material due to its high nonlinear coefficient (Supplementary Table 2) and potential for subwavelength SPDC emission[14]. We thus transferred an $NbOCl_2$ flake (275 nm thick) onto a gold film. A continuous-wave (CW) laser centered at 409 nm was used as the fundamental wave and a Hanbury Brown-Twiss setup[43] was used for all coincidence counting experiments. The SPDC response from $NbOCl_2$ was first measured when transferred onto an $SiO_2/Si$ substrate, showing a coincidence peak above the background when the time difference between the two detector channels is set to 0 ns, which is a signature of biphoton generation (Fig. 4b). Interestingly, the $NbOCl_2$ flake on gold exhibits a tenfold increase in the coincidence count rate. The polarization-independent characteristic of the Au film obviates the need for orientation-specific alignment between the $NbOCl_2$ flake and its substrate, thereby not influencing the efficiency of the SPDC

emission. As with the SHG process discussed in previous sections, the presence of the metal just below the $NbOCl_2$ redefines the boundary conditions of the fundamental wave, inducing an enhancement of the electric field and higher-order contributions in the nonlinear process, while also increasing the reflectivity at the signal and idler wavelengths.

To evidence that the photon pairs are generated by SPDC, we examined several critical aspects of the correlated photon pair generation process. Both the signal and idler photons are emitted with polarization along the crystallographic b-axis, demonstrating that $NbOCl_2$ exhibits mainly type-0 SPDC emission (Fig. 4c). Figure 4d shows the coincidence count rate from $NbOCl_2$ on the gold film across different detection wavelength ranges, with all measurements conducted under the co-polarized configuration. Wider detection bandwidth results in larger coincidence rate, consistent with broadband biphoton generation due to relaxed phase-matching conditions[17]. On the other hand, the narrower the detection bandwidth the fewer the accidental coincidences, increasing thereby the coincidence-to-accidental ratio (CAR)[14,44], as shown in Supplementary Fig. 10. The observed biphoton coincidence rate is linearly proportional to the fundamental wave pump power, as shown in Fig. 4e, while the CAR is inversely proportional to it

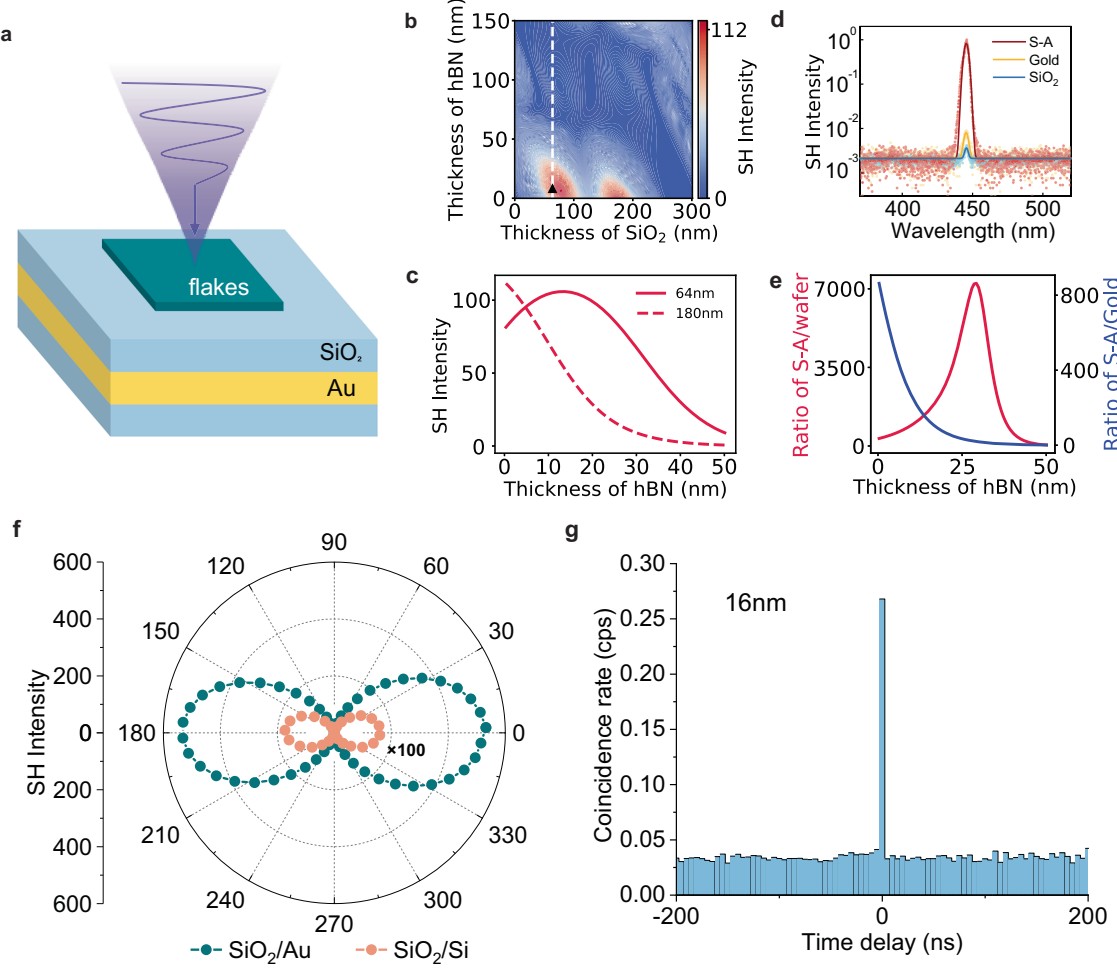

**Fig. 5 | Characterization of enhanced nonlinear processes from few-monolayer flakes on dielectric/gold heterostructures. a** The schematic of NbOCl₂ flake on SiO₂/Au structure. **b** SHG intensities as a function of h-BN and SiO₂ (on top of gold) thicknesses for h-BN flakes with an odd number of layers, deposited on a substrate made of a SiO₂/Au/Si wafer structure. The dashed line corresponds to the calculation range for the solid line shown in (**c**). **c** Simulated SHG intensities for h-BN on the SiO₂/Au/Si wafer structure, considering different h-BN and SiO₂ thicknesses: 64 nm for odd-layered h-BN (solid), 180 nm for odd-layered h-BN (dashed). **d** Experimental results highlighting the SHG intensities for an 8 nm h-BN flake on SiO₂/Au (red), gold (yellow), SiO₂/Si (blue) substrate. **e** Enhancement factor of SHG intensity from h-BN flakes on 64 nm SiO₂-Au

compared to Si wafer $\left(\frac{I^{SH}_{S-A}}{I^{SH}_{wafer}}, \text{red}\right)$ and enhancement factor of SHG intensity from h-BN flakes on 64 nm SiO₂/Au compared to gold $\left(\frac{I^{SH}_{S-A}}{I^{SH}_{gold}}, \text{blue}\right)$, both involving h-BN flakes with an odd number of layers. **f** Polarization dependent SHG measurement of 16 nm NbOCl₂ flake on 64 nm SiO₂/Au substrate (green) and SiO₂/Si wafer (orange). The 16 nm NbOCl₂ flake on SiO₂/Au substrate shows an enhancement of SHG intensity by a factor 306. **g** SPDC coincidence counts rate on 16 nm NbOCl₂ flake on 64 nm SiO₂/Au under co-polarized configuration with an enhancement factor 100 (lower bound) in a 100-minute measurement.

due to the increased background photons (Fig. 4f), which is a typical indication of nonclassical photon pair emission[17]. Finally, we analyzed the coincidence count rate relative to the polarization of the fundamental wave. When rotating the polarization of the fundamental wave with respect to the NbOCl₂ flake and measuring SPDC in the co-polarized setting, the generated pattern exhibits a two-lobed symmetry (Fig. 4g), consistent with the C₂ crystal group characteristic of NbOCl₂[45] and with our SHG experiments on NbOCl₂ (Fig. 5f). Compared with recent approaches to enhance SPDC in LiNbO₃ layers, which made use of resonant metasurfaces etched either onto the LiNbO₃[46] or onto a dielectric film atop[47], our approach is much easier to implement and paves the way for more efficient studies of photon frequency entanglement across a broader range of operation wavelengths.

## Pushing nonlinear efficiency of nanometric sources
Up to now, for the SHG measurements with h-BN we have dealt with layers several tens of nanometers in thickness. However, many applications (e.g., quantum sensing and quantum communication) and

certainly ultimate miniaturization require nonlinear sources below 20 nm or even 10 nm thickness[48]. Unfortunately, if such thin layers were deposited onto the gold substrates discussed previously, although the SHG signal would be larger than on standard SiO₂/Si substrates (Fig. 2a), it would be substantially smaller than if the h-BN layer was just double in thickness. As shown in Fig. 2a, when the layer thickness becomes 15 nm, the SH signal reduction is approximately fivefold in comparison with a layer thickness of 30 nm.

To strengthen the nonlinear response of films from several monolayers to tens of monolayers thick, we have designed a structure consisting of a SiO₂/Au/Si wafer stack that incorporates a planar dielectric Fabry-Perot cavity between the material and the metal (Fig. 5a). Through nonlinear transfer matrix calculations, we anticipate a significant enhancement originating from magnified dipolar and quadrupolar contributions to the polarization at the h-BN location, compared to the signal obtained on the gold film directly, for h-BN flakes containing both odd and even numbers of monolayers (Fig. 5b and Supplementary Fig. 11). In the case of very thin layers, where the

gradient of the electric field is smaller, the quantitative enhancement of SHG again becomes parity dependent. Very thin flakes with an odd number of monolayers tend to favor SHG over even-layered flakes, and optimized structures need to be fabricated depending on parity (Fig. 5c). Note that for the current Air/SiO$_2$/Gold configuration if we assume that the reflection phases at each of the two interfaces differ roughly by $\pi$, the minimum resonant cavity thickness is not $\lambda/2n_{cav}$ but $\lambda/4n_{cav}$. This resonant condition would correspond to a thickness in the order of 153 nm for the 890 nm pump and about 77 nm for the SH waves. For a monolayer-thick h-BN (i.e., for vanishingly-thin h-BN with odd number of monolayers), the optimum SiO$_2$ thicknesses are around ~76 nm and 166 nm (Fig. 5b), so very close to the resonant condition for the SH waves and to the condition for a maximum of the electric field pump amplitude (maximizing the dipolar contribution). Optimizing either of these two effects results in a similar computed SHG signal. Furthermore, as shown in Fig. 5b, for h-BN containing odd-number of monolayers and deposited on a SiO$_2$/gold substrate, the h-BN thickness maximizing the SHG can be extremely thin (in the order of monolayers).

In contrast, for h-BN with vanishingly small even number of monolayers (Supplementary Fig. 11), the optimum SiO$_2$ thickness (121 nm) lies between the resonant conditions for the SH waves and the pump. Besides, contrary to h-BN containing odd number of monolayers, very thin even-parity h-BN does not maximize the SHG response. Indeed, to maximize the quadrupolar moment controlling the response of centrosymmetric h-BN, we need to have a non-negligible h-BN thickness, leading to an optimum SiO$_2$ thickness of 51 nm for an optimum h-BN thickness of 59 nm.

To experimentally validate our findings, we transferred an 8 nm h-BN flake onto a 64 nm-thick SiO$_2$ layer deposited on gold. The measured two orders of magnitude enhancement of the SH signal compared to the same flake deposited directly onto gold (Fig. 5d) confirms the efficiency of the SiO$_2$/Au structure in enhancing the nonlinear response of very thin h-BN, which still remains broadband (Supplementary Fig. 12). Furthermore, this straightforward and versatile method can also be extended to enhance the SHG for thicker h-BN layers, providing three orders of magnitude enhancement compared to standard h-BN on SiO$_2$/Si substrate (red curve in Fig. 5e). This is of practical importance for the 2D community to analyze the parity of thick h-BN samples in which spatial thickness variations on the order of the monolayer are expected, as illustrated in Supplementary Fig. 13a. Monitoring the SHG photon count rate after the transfer of a monolayer flake onto a "thick" h-BN revealed a pronounced distinction between the SHG intensity of the 17 nm flake and its augmented state post-monolayer addition (Supplementary Fig. 13b). This stark intensity contrast not only underscores the impact of the additional layer but also confirms the even-layered structure of the flake, consistent with our simulations. In comparison to the threefold contrast reported in previous studies for similar layer thickness differences[23], our SiO$_2$/Au structure approach markedly improves the contrast ratio by an order of magnitude.

To verify the applicability of the approach for enhancing quantum nonlinear processes too, we transferred a 16 nm NbOCl$_2$ flake onto the same 64 nm SiO$_2$/Au/Si wafer structure. The characterization of the classical SHG process, under co-polarized configuration measurement, displays a giant enhancement of over 300 times compared to 16 nm NbOCl$_2$ deposited on SiO$_2$/Si wafer (Fig. 5f). The coincidence counts per second measured from the 16 nm NbOCl$_2$ flake on 64 nm SiO$_2$/Au/Si structure is presented in Fig. 4g. A distinct coincidence count peak, with a CAR exceeding 8 (Supplementary Fig. 14), was detected in the co-polarized configuration with a 0.25 mW pump power. A lower bound enhancement factor of 100 was obtained in a 100 min measurement. To our knowledge, this structure features a nonlinear medium that is three times thinner than previously reported values[14], paving the way to the development of real ultra-thin integrated quantum devices. A quantitative comparison of "thin" active regions showing SPDC is provided in Supplementary Table 4.

## Discussion

We have proposed an easy-to-implement method for enhancing the nonlinear response of both centrosymmetric and non-centrosymmetric van der Waals materials that involves manipulating the amplitude, phase, as well as gradient of the pump field. By engineering the light component we achieved a substantial boost in the light-matter interaction resulting in a notable increase in the nonlinear response, as observed for both SHG and SPDC (Supplementary Table 6). For centrosymmetric materials like h-BN, which typically exhibit lower second-order susceptibility values due to their symmetry, our approach has enabled an impressive increase of about two orders of magnitude in the SHG response and enabled even-layer h-BN to exhibit SHG responses comparable to those of odd-layer h-BN. This demonstrates that the symmetry constraints on the nonlinear optical response can be effectively mitigated by adequately modifying the electric field distribution within the nonlinear material.

The desired operational bandwidth of a nonlinear optical system is highly dependent on the specific application. For instance, quantum memory of single photons requires an extremely narrow bandwidth to ensure high fidelity and efficiency in photon storage and retrieval. This contrasts with other applications, such as broadband optical parametric amplifiers and supercontinuum generation, which rely on robust operation across a wide range of frequencies and benefit from broader bandwidths. In addition, the growing interest in ultra-thin entangled photon sources, particularly for integrated quantum computing and communication devices, highlights the importance of broadband capabilities. Frequency entanglement, which leverages the relaxed momentum conservation in thin films, is increasingly recognized for its potential in encoding information. In such systems, it is essential to enhance the generation of photon pairs over a broad spectrum of frequencies to maximize entanglement efficiency. Our material-on-metal configurations are particularly well-suited for such broadband operations, with at least 100 nm large bandwidth. The broadband enhancement of quantum nonlinear processes was demonstrated by achieving a ten-fold increase in the SPDC coincidence rate of NbOCl$_2$ layers deposited on gold.

By introducing a SiO$_2$ layer between the van der Waals flakes and the gold film the enhancement for monolayer thick h-BN can be boosted by 3 orders of magnitude compared to SiO$_2$/Si. Thus, this method can accommodate h-BN layers ranging from the monolayer level to hundred nanometers thick. Most importantly, we demonstrate correlated photon pair generation in a 16 nm-thick NbOCl$_2$ layer transferred onto the same SiO$_2$/Au structure, suggesting its potential for achieving SPDC down to the monolayer level.

## Methods
### Sample preparation
The hexagonal boron nitride (h-BN) flakes were prepared by mechanical exfoliation with poly-dimethyl Siloxane (PDMS). The twisted bilayer h-BN was fabricated by the tear and stack method. The bottom layer h-BN was picked up by polycarbonate (PC) film and transferred onto the pre-patterned gold lead. Then another h-BN flake was torn apart partially and picked up by PC film at 70 °C, leaving the other half of the h-BN on the SiO$_2$ substrate. The PC/h-BN stack on the transfer plate was rotated by 180° and stacked on top of the remaining half h-BN. Finally, the twisted h-BN bilayer was picked up at 70 °C and dropped on top of the prepared h-BN flake on the gold to finish the whole structure. The PC film was removed with chloroform wash (Fig. 3a).

The NbOCl$_2$ layers were obtained by mechanical exfoliation of bulk NbOCl$_2$. After that, the exfoliated flakes were transferred onto a polydimethylsiloxane (PDMS) film. We selected appropriate flakes on the PDMS film for the fabrication of the heterostructures.

As determined by AFM, the thickness of gold/titanium substrate used in Figs. 2–4 is 200 nm/18 nm, and the thickness of SiO$_2$ in the SiO$_2$/Au heterostructure of Fig. 5 is 64 nm. The primary role of the Ti

layer is to act as an adhesion promoter, ensuring a strong bond between the gold film and the underlying substrate. The thickness of $SiO_2$ layer on $SiO_2$/Si substrate is 285 nm. The thickness of $NbOCl_2$ shown in Fig. 4 is 275 nm while in Fig. 5 it is 16 nm.

## Second harmonic generation

To perform second harmonic generation (SHG) measurements we utilized a custom-built confocal setup. The samples under investigation were excited using a tunable Ti: sapphire laser. The laser beam passed through a series of optical elements, including a polarizer, a half waveplate, a quarter waveplate, a beam splitter, a second half waveplate, and an objective. The emitted signals from the sample were directed through the beam splitter, a half waveplate, and a 600 nm shortpass filter before being directed to the spectrometer for analysis. Additionally, the half waveplate positioned next to the objective was motorized, allowing for investigations of angle-dependent SHG.

To ensure a more precise and reliable calculation of the SHG enhancement factor, we have utilized Lorentz fitting directly on the raw data to extract the maximum SHG signal values, ensuring that no artificial baseline adjustments impact the analysis. This method allows us to accurately capture the polarization-dependent SHG response for h-BN on different substrates, including gold and $SiO_2$. For each substrate, we identify six maximum values that correspond to the six-fold symmetry characteristic of h-BN. By performing the same measurements on h-BN on and off the gold substrates, we calculate the enhancement factor at each angle. We then compute the mean value and standard deviation to provide a robust estimate of the enhancement factor and its associated uncertainty. This approach ensures that the enhancement factor is derived from the true physical properties of the system, free from the influence of any background subtraction, and accurately reflects the influence of different substrates on the SHG response.

## Coincidence experiments

In our study, photon coincidence measurements were conducted using a specialized Hanbury Brown-Twiss setup. A 409 nm continuous wave (CW) laser was used to excite our samples, utilizing a back-reflection configuration. In the excitation path, the laser passed through a polarizer, a half waveplate, a quarter waveplate, a dichroic mirror, another motorized half waveplate, and a microscope objective (50x magnification, 0.8 numerical aperture). The emitted signals from the sample passed through the same objective, dichroic mirror, a 750 nm long-pass filter, a beam splitter and then were divided into two output channels. Each channel was equipped with a quarter waveplate, a half waveplate and a polarizer before being directed to the avalanche photodiode for analysis. The correlation signals from the two paths are identified by a time tagger (Swabian Instruments). We set the polarization of the fundamental waves aligned to the orientation of the crystallographic polar axis of the $NbOCl_2$ (denoted as b). The lower bound of the enhancement factor for SPDC is determined using the signal-to-noise ratio (SNR), as the SPDC signal from the material off the gold film is at or below the background level. Specifically, it is calculated as: $(C^{on} - \langle A \rangle^{on})/\delta A^{on}$ and $(C^{on} - \langle A \rangle^{on})/\delta A^{off}$, where $C$ is the coincidence counts per second and $A$ is the accidental counts per second for on and off structures, $\langle \rangle$ means mean value and $\delta$ represents the standard deviation. Using this approach, the calculated lower bound values of the enhancement factor are 93 and 153, respectively, confirming a significant enhancement of the SPDC signal. The bin width in our coincidence experiments was set to 5000 ps.

## Nonlinear transfer matrix method calculations

To understand SHG enhancement we use nonlinear transfer matrix simulations[23,32] in which the only approximation is that the pump field is unaffected by the second-order nonlinear process, the so-called undepleted-pump approximation. This formalism enables us to calculate precisely the pump field distribution (i.e., its spatially varying amplitude as well as its spatially varying gradient, as illustrated in Fig. 1a) imposed by the different interfaces and materials employed, especially on metal surfaces. It also allows us to introduce the nonlinear polarization sources at the level of each h-BN monolayer, facilitating the analysis of the effect of the actual parity of monolayers (odd or even) as well as of rotated heterostructures[23,49]. Noticeably, phase shifts between different waves involved in the process (pump, nonlinear sources and second-harmonic) are inherently included in this formalism. Thus, phase-matching conditions limiting SHG in bulk crystals are naturally retrieved.

## Nonlinear susceptibility tensor of NbOCl2

The crystal structure of $NbOCl_2$ belongs to the $C_2$ space group, which determines the form of its second-order nonlinear susceptibility tensor:

$$\begin{bmatrix} 0 & 0 & 0 & d_{14}^{(2)} & 0 & d_{16}^{(2)} \\ d_{21}^{(2)} & d_{22}^{(2)} & d_{23}^{(2)} & 0 & d_{25}^{(2)} & 0 \\ 0 & 0 & 0 & d_{34}^{(2)} & 0 & d_{36}^{(2)} \end{bmatrix} \quad (1)$$

where $d_{14}^{(2)} = d_{25}^{(2)} = d_{36}^{(2)}, d_{23}^{(2)} = d_{34}^{(2)}$, and $d_{16}^{(2)} = d_{21}^{(2)}$. When the pump field is incident perpendicularly to the $NbOCl_2$ crystal (i.e., vertically in Fig. 4a), the non-zero term contributing to the polarization in the nonlinear material is $d_{22}^{(2)}$ and $d_{23}^{(2)}$. Considering that $d_{22}^{(2)} >> d_{23}^{(2)}$, the main SPDC response in $NbOCl_2$ is type-0. In type-0 SPDC, the polarization of the pump, signal, and idler photons are all the same.

## Data availability

Relevant data supporting the key findings of this study are available within the article and the Supplementary Information file. All raw data generated during the current study are available from the corresponding authors upon request.

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

## Acknowledgements

This work was supported by Singapore National Research foundation through CRP grants (CRP Award Nos. NRF-CRP22-2019-0004) and QEP Grants (NRF2021-QEP2-01- P01, NRF2021-QEP2-01-P02, NRF2021-QEP2-03-P01, NRF2022-QEP2-02-P14), and ASTAR IRG (M21K2c0116).

## Author contributions

Conceptualization, X.L., J.Z.P., W.G. Methodology, X.L., L.K., J.Z.P., W.G. Material, H.C., L.A., R.D., C.Z., Z.L. Simulation, X.L. Visualization, X.L. Discussion, S.J.W., Q.T., A.L. Writing—original draft, X.L., J.Z.P., W.G. Writing—review & editing, X.L., L.K., R.H., Y.M., J.Z.P., W.G.

## Competing interests

The authors declare no competing interests.
