## [Transparent Peer Review file · Nature Communications]

Boosting classical and quantum nonlinear processes in ultrathin van der Waals material

Corresponding Author: Professor Jesus Zuniga-Perez

Version 0:

Reviewer comments:

Reviewer #1

(Remarks to the Author)

The authors propose and experimentally demonstrate an interesting and elegant approach to boost the nonlinear efficiency in van der Waals material, especially for materials centrosymmetric crystalline structures such as hBN. This is achieved by simply transferring nonlinear materials onto a metal film. Both SHG and SPDC processes can be enhanced by more than two orders of magnitudes. Using this technique, they have demonstrated the photon-pair generation from an NbOC12 flake of only 16 nm, establishing it as the thinnest SPDC source so far. While this work appears very interesting, I would like to invite the authors to address the following concerns before recommending it for publication.

1. The authors claimed that the unique advantage of this material-on-metal approach is broad-band operation, as compared to other approaches using nanocavities or metasurfaces. However, can similar nonlinear efficiency be achieved with nanocavities or metasurfaces with low-quality factors, which also allows broad-band operation? Also, what is the operation bandwidth for material-on-SiO₂/Au configuration the authors introduced in Fig. 4 to further improve the enhancement? Does higher enhancement imply a narrow bandwidth? Additionally, whether a large bandwidth is good or not depends on the specific application; for example, quantum memory of single photons requires extremely narrow bandwidth.

2. Following the first comment, the authors cited references 26 and 29 to support their claims about bandwidth limitations imposed by nanocavities or metasurfaces. It appears these two references are not related to photonic resonances. Are there any previous works that have demonstrated enhanced nonlinear processes with nanocavities or metasurfaces?

3. The authors have discussed the physical mechanism for the enhancement enabled by the material-on-metal approach. If I understand correctly, there are two contributions: (i) field enhancement for the signal/idler photons; and (2) larger electric field amplitude gradient. It appears that both of these contributions are important for the nonlinear enhancement of centrosymmetric materials such as hBN, while the enhancement in non-centrosymmetric material such as non-centrosymmetric material NbOC12 is due to contribution (1) only. If this is the case, it should be discussed more clearly. More importantly, the physics behind field enhancement in contribution (1) is not explicitly provided while the authors have mentioned boundary conditions. Is it simply the Fabry-Perot effect? Is the reason that the material-on-SiO₂/Au approach provides higher enhancement due to the higher Q of the Fabry Perot cavity? Is there such a photonic resonator effect in your approach that you wanted to avoid?

4. The authors performed the classical SHG measurement with hBN and SPDC with NbOC12. Have the authors tried to measure SPDC from hBN? How is the fluorescence from hBN? Does NbOC12 perform better than hBN in terms of SPDC? Which one provides higher nonlinearity? What are the unique benefits of hBN?

5. In Fig. 1c,d, the authors presented that the nonlinear tensor can be modified by the metal film. Does this modification depend on the hBN layer thickness? Is the modification consistent for all the samples-on-metal? The nonlinear tensor looks interesting, any potential for generating polarization entanglement?

6. In lines 227-228, it is stated that the "quantum SPDC generation rate is proportional to the classical SHG rate, the proportionality being controlled by the pumping conditions." This is not true. While there is indeed a classical-to-quantum correspondence between SPDC and SFG (rather than SHG), it is not a simple proportional relationship. SPDC is related to the integration of all possible SFG processes over the whole spatial and spectral spectrum.

7. The term “pump field” in the SHG measurement can be confusing as the same term is used for the SPDC process. It is better to use signal/idler photons or fundamental waves instead.
8. Fig. 3b seems unnecessary because Fig. 3c has included all the information needed.
9. Fig. 3g presents the biphoton polarization dependence. However, SPDC rate (instead of CAR) vs. polarization angle is a more appropriate and standard measurement to confirm biphoton polarization. Also, this figure is not cited in the main text.
10. Figure 3e shows the CAR vs. pump power. How does the SPDC rate depend on the pump power? I recommend making plots of CAR/SPDC rate vs. pump power (rather than showing different correlation plots under different powers).

Reviewer #2

(Remarks to the Author)

In their work “Giant boost of classical and quantum nonlinear processes in ultrathin van der Waals materials”, Lyu et al are investigating the intensity of second harmonic generation (SHG) and spontaneous parametric down conversion (SPDC) in thin-film material systems, namely hBN and NbOCl₂, that are placed onto or close to a gold surface and show that the efficiency of SHG and SPDC is enhanced.

Overall, I do not think the results are very novel (similar works done before, see below) and I also believe the impact for the community is also quite limited (the involved physics is rather trivial, see below). At the same time, from a scientific point of view, the manuscript is not well written. That means the analysis that is done is not clear and convincing, and the results are not clearly interpreted, and it is also not clear what is the main claim of the paper, in terms of the physics that is discovered. The manuscript is a rather chaotic gathering of different experimental and theoretical results, corresponding to different combinations of nonlinear material and photonic systems, where a clear and coherent understanding of the physics and interpretation of the results are not presented.

This also makes judging the impact of the paper rather hard, not knowing what the exact claim of the paper is. Hence, to some extent my judgment on the novelty and the impact of the paper is based on what I tried to infer myself by looking at the theoretical and experimental plots and trying to associate a clear physical explanation to the observed results. From my understanding, I could conclude that the observed effect is simply due to having a mirror on one side of the nonlinear material, which would increase the pump field intensity by a factor around 4 at some places and also allows for collecting light that is generated and is propagating in the forward direction without the mirror. This is a rather trivial effect. There are already many works done on hybridizing van der Waals/2D materials with resonator- or waveguide-based nanophotonic structure (see the following recent reviews [<https://doi.org/10.1039/D3NH00246B>; <https://doi.org/10.1038/s41578-023-00558-wj>]), among which there are also works which are very similar to the works done here, with 2D materials placed on metallic mirror [<https://doi.org/10.1016/j.matt.2024.04.043>] and also within a cavity with dielectric Bragg mirrors [<https://doi.org/10.1002/adom.202300907>], both demonstrating enhanced SHG. This rather takes away the general novelty of this current work. From the point of view of physics discussed, it is clear that there is always a trade-off between the bandwidth (quality factor) of the resonance and its efficiency. The lowest limit of a bad cavity, is one where one mirror is missing in the cavity and you just have one mirror. So instead of getting an enhancement in efficiency that comes from many reflections in the cavity over a narrow bandwidth, one would get an enhancement caused by one reflection which would work over a large bandwidth. Such effects are very trivial, and pretty much are the cornerstone of nonlinear optics in the last many decades and very well understood. At the first look, I thought maybe the results come from a special interaction between the van der Waals material and the metal (which would have been interesting), but that seems to not be the case, as the results have no special connection to having van der Waals materials or even having a metallic mirror. As I understand the paper, the same effects can be seen in any thin film nonlinear material and any type of mirror (Bragg mirror for example). With this, I do not think the result is very novel and impactful. Again, this is based on my understanding of the physics of the system, which was unfortunately not clearly offered in the manuscript.

Nonetheless, below I give my detailed comments on what I see as the major issues in the manuscript. Although, even by implementing them, as I currently understand the physics of this system, I do not think that changes the general novelty and impact of the work. Nonetheless, here are my comments:

- First a general comment, which is a problem that drastically reduces the quality of the manuscript. Overall, the authors are using references in a very uncaredful way. For example, line 42-45: “Beyond classical nonlinear processes, van der Waals materials facilitate correlated photon pair generation by four-wave mixing (FWM) [14], optical parametric amplification and oscillation [15], and spontaneous parametric down-conversion (SPDC) [16-21].” Most of the references here are not really van der Waals materials. Specifically references [16] to [19] are either lithium niobate or III-V semiconductors. Reference [15], as far as I know, does not reach an OPO regime, but is rather an OPA. Also, in line 41-42, they say “Nonlinear processes have been demonstrated in TMDs [4-9], thus making it possible to realize ultrafast optical switching and parametric amplification [10-13].” Which one of references from [10] to [13] contain parametric amplification? Also, in line 45-48 “Notably, the implementation of SPDC in ultra-thin subwavelength materials offers a significant advantage by relaxing momentum conservation criteria [20, 21], thereby alleviating the size limitations imposed by bulky nonlinear crystals that limit their on-chip integration [22, 23].” References [22-23] do not seem to be relevant for the sentence before it. The authors seem to be talking about the advantage of not having phase-matching restrictions for nonlinear effects in thin materials, then they cite reference [22-23] in integrated lithium niobate and silicon, which does not seem to be about nonlinear effects. Line 64-65, they say “the spectrally resonant character of the enhancement limits their operation bandwidth [26, 29]”. Does reference [26] employ any resonant photonic structure? Also, line 146-147, reference [28] is not based on resonant nanocavities, it is

based on phase-matching, in contrast to what is stated in the manuscript there. In line 149-150, references [35-37] do not show the SHG polarization dependence for hBN or TMDs, as far as I can see, so they do not relate to the statement before them. Overall, I think the authors should very carefully reexamine “all” of their references and make sure that they are mentioned in the correct context and with the correct explanation.

- A comment about a main claim of the paper: The authors say in the abstract, and I also in the manuscript something similar, that “The demonstration of SPDC in 16nm NbOCl₂ flakes establishes our device as the thinnest SPDC source so far.” I am not sure if it is in the policy of Nature journals to make such claims. Nonetheless, I also do not think it is straightforward to say this device is only 16 nm thick, since you have to consider the thickness of your whole device, involving the metal and the SiO₂ thin films, and then you need to compare at least to some other state of the art works, including recent works on thin film lithium niobate and GaAs and also the corresponding metasurfaces in those platforms (see [<https://doi.org/10.1364/PRJ.474387>] as a place to start, but there are many works in these directions in the last few years).
- This is in my opinion the most major issue: The physics of the nonlinear interaction is quite unclear and the analysis is very confusing. To put it very shortly, what is the role of $\chi^{(2)}_d$ and $\chi^{(2)}_q$? The authors start talking about them in line 115-116, and emphasize their importance and make it sound like the enhanced effect is somehow coming from an interplay of these two effects. Then in lines 125-128 they say “Our strategy relies in maximizing the electric field amplitude at the nonlinear-material location and shrinking down the scale at which electric field gradients are considerable, both effects being controllable thanks to the presence of a neighboring metal surface, which imposes strict boundary conditions on the electric field.” Which again emphasizes the physical importance of $\chi^{(2)}_d$ and $\chi^{(2)}_q$. Then the rest of the analysis and their simulations seem to not involve these quantities at all. The simulations seem to be based on nonlinearity of single monolayer hBN, which they are not including in the initial explanation of the physics. Is the dipolar and quadrupolar bulk nonlinearity somehow connected and can be derived from the single monolayer nonlinearity? If so, these relations should be rigorously derived and some actual numbers should be associated to these quantities. From my understanding, the connection between the physics explained in the motivation and the simulations are lost. At the end, the authors look at many different platforms, and it is not clear in any of them, what is actually the source of enhancement.
- What is special about using the hBN and NbOCl₂ materials? If the source of the nonlinearity is $\chi^{(2)}_d$ and $\chi^{(2)}_q$, which are bulk quantities, are these numbers particularly larger in hBN and NbOCl₂ compared to say materials like lithium niobate or GaAs? Numbers should be given and discussed.
- In line 148-150, the authors say that “The polarization-dependent SHG measurement exhibits a distinct six-fold symmetry (Figure 1c), in agreement with the D_{3h} point group of h-BN and some TMDs [26, 35-38].” Since the authors seem to be claiming that there is a dominant contribution from $\chi^{(2)}_q$, the question is, does the quadrupolar $\chi^{(2)}$ tensor have the same tensor symmetry as the dipolar one? The authors should cite a work that has investigated the tensor symmetry of the quadrupolar $\chi^{(2)}$ either for hBN or generally showing that it has the same tensor symmetry of the dipolar $\chi^{(2)}$.
- Another important point that should be discussed: Can one rule out if the SHG/SPDC enhancement is coming from the metal surface nonlinearity? What is the pump beam diameter? The authors give a number in the supplementary, that say 2 by 2 meters, which is obviously a typo and should be corrected. I would guess they meant 2 by 2 micrometer is the pump beam size. In such a focusing regime, one would have some non-negligible longitudinal polarization component for the field, which might generate longitudinal and transversal field components through the surface nonlinearity of the gold, that might then be collected in reflection. The experimental test would be to do the same work without the hBN or NbOCl₂, and just with the metal-based structures.
- A major concern, which also relates to the previous two points, is that in fact the measured polarization measurement results in the presence of the metal, shown in Fig 1(c), and also in Fig S6 of the supplementary, do not match the expected 6-fold symmetry of SHG from hBN. The authors explanation of “This phenomenon might be tentatively ascribed to the inhomogeneous strain induced by the irregularities of the gold film (see Figure S6 for more examples of this asymmetry).” is not convincing and is also not supported by any evidence. How irregular is the surface of the gold? What type of strain can it cause? And how is strain expected to affect the nonlinear tensor in 2D materials? A much more probable cause could be that the SHG is coming from different contributions of materials with different $\chi^{(2)}$ tensors. One strong candidate is the surface nonlinearity of the gold. Yet these are not really investigated. I think the authors should justify their strain argument with some more substantial reasoning and citing some similar studies on that topic, and at the same time do some reasoning on why it cannot be other contributions like gold’s surface nonlinearity.
- In plots like 1(b) or 1(e), why is the signal generated for the case of hBN on SiO₂ around 0? I assume the authors subtracted the average value for this measurement. But for an analysis where one is interested in the ratio of two signals, one should not do a subtraction from both signals, otherwise how can the ratio be calculated? I think the authors should show the actual measured values and not the average-subtracted values, which is not useful for the analysis of enhancement.
- Why is there Ti layer under gold? Is that needed for bonding? Would be good if this is explained too.
- In the experiments, how do the authors determine/measure if a hBN flake has odd or even number of layers? Is it just from the SHG measurements or is it some direct measurement approach like AFM? This should be mentioned.
- In Fig 2(e), a combination of bad image quality, bad choice of colors, and lack of legends, makes this plot completely non-understandable and consequently any physics that is tried to imply from it is lost.
- Another major issue is the following: The enhancement factor, as shown in Fig 2(a) is rather a meaningless figure of merit. If I understand correctly, what is measured is how much SHG is generated for a certain thickness of a material, comparing the same thickness when placed on SiO₂ and gold. For such a system, where the authors are claiming that everything is dependent on the intensity of the field at the surface, it could easily be that for a specific thickness, the gold substrate has a dramatically different field at the surface compared to SiO₂ substrate, due to simple Fabry Perot effects. For example, I can imagine if the thickness of the SiO₂ layer is systematically changed, this enhancement factor could easily change. So that does not mean that the gold substrate is creating a dramatic enhancement in the field when considering all thicknesses. This can in fact even be seen in the simulations shown in Fig. 2(b), where the overall enhancement of generated SH, when considering all thicknesses and averaging the results in a way, seems to be about an order of magnitude. This order of magnitude enhancement more or less makes sense, since reflection from the interface makes the electric field generally 2

times stronger at some points, and this makes the intensity of fundamental harmonic pump 4 times stronger, which makes SHG intensity about 16 times stronger. Then one can consider that reflection is collecting also waves generated in the transmission direction and sends them to the reflection side, creating a factor of 32. So such enhancement factors are easily explained by the effect of the mirror, which is not a really an interesting effect. I suggest the authors perform such a study, where the thickness of hBN is fixed but the SiO₂ thickness is changed, to find a fair ratio between the efficiency of the system with and without gold. Overall, I think that this enhancement factor at equal thicknesses is not a reasonable figure of merit to judge the enhancement, and a more general figure of merit should be defined that can somehow take out this very sensitive thickness-dependent interferences to be able to judge how much the gold substrate is affecting the system.

- This one order of magnitude enhancement is also stated for the SPDC measurement (line 237-238), where again the same explanation as above follows, where the main contribution to enhancement is coming from reflection, and has not profound connection (that I can see) to the dynamics of dipolar, surface, or quadrupolar nonlinearity.

- A major issue for analysis of SPDC: Line 267-269 the authors say "The observed biphoton emission rate has an inverse relation to the pump power, which is a typical indication of nonclassical photon pair emission [16]." Where is this shown? The authors are just showing CAR as a function of pump power. Those are not the same thing. Also, why are the authors analyzing CAR for studying the polarization dependent response in figs 3(f) and (g)? Fundamentally the CAR should not be polarization dependent. The fact that it is here raises many questions. The effects that the authors are trying to describe would only make sense by using the absolute coincidence rate as the quantity for interpretation. This analysis with CAR is very confusing and raises many questions about why CAR is polarization dependent. CAR is the coincidence to accidental ratio, and having more CAR does not mean a more efficient process.

Reviewer #3

(Remarks to the Author)

In this manuscript, the nonlinear optical properties of both hBN and NbOCl₂ flakes are investigated with different substrates, which make distinct nonlinear optical behaviours, especially the nonlinear optical enhancement effect. The authors studied SHG and SPDC for hBN and NbOCl₂ flakes respectively. They found that the Au substrate show obvious signal augmentation, compared to SiO₂/Si substrate. Further, SiO₂/Au/Si substrate demonstrate outstanding performance, which can be used to enhance nonlinear optical signal for very thin flakes. The theoretical explanation of field amplitude and gradient is reasonable. There are many literatures about the nonlinear optical enhancement effect of van der Waals materials. The enhancement factors of some of them are larger than the results reported here. However, this work is interesting and systematic, which could be considered for publication in NC. My questions and comments are as follows.

1, The nonlinear optical signal from Au film should be considered. That is, to measure the signal of Au substrate simultaneously, and the enhancement factor needs to be reevaluated.

2, Are there possibilities for the proposed substrates to enhance single layer van der Waals material?

3, In Fig. 1, why to change the thickness of hBN, which makes the results be not comparable.

4, In Fig. 2, the 49 nm thick hBN flake was separated into 4 pieces, not 2 pieces, right?

5, The phenomenon of broadband enhancement is a little bit overemphasized.

6, For SiO₂/Au/Si substrate, the field distribution has a dependence on wavelength. Whether such SiO₂/Au/Si structure also maintains broadband characteristics like Au substrate?

7, I suggest the authors use picosecond laser to conduct measurements, with higher wavelength resolution.

8, The SEM/AFM images of Au and/or SiO₂ films can be provided, whose flatness may have an impact on the nonlinear optical process.

9, As for the writing, there are many long sentences, which lead to difficulty in reading.

Version 1:

Reviewer comments:

Reviewer #1

(Remarks to the Author)

The authors comprehensively revised the manuscript, including the text and figures. Additional experiments and theoretical analysis are provided to support their claims. The updated results regarding nonlinear enhancement with a metal substrate appear quite interesting. While most comments from all reviewers have been addressed appropriately, the following points need to be clarified before I could recommend it for publication.

1. The authors discussed several different structures and many different experimental conditions. The physical mechanism and the achieved enhancement are different for these structures. I recommend including a table summarizing the features of these structures, including the nonlinear enhancement factors, physical mechanisms, benefits, and limitations.

2. In Line 117, the authors provide an equation describing the nonlinear polarization with contributions from dipole and quadrupole moments. Can the authors provide references for this equation? Additionally, have previous works in any nonlinear system proposed other approaches to enhance the quadrupolar term?
3. The authors mention that the enhanced quadrupolar response in non-centrosymmetric materials can be 2-3 times smaller than the dipolar response. Does this indicate that the proposed approach to enhancing the nonlinear response is more suitable for centrosymmetric materials?
4. In Line 183, the authors claim that the dipolar enhancement can reach two orders of magnitude for an odd number of monolayers. Why is the enhancement so large for just a reflection-based mechanism?
5. In Fig. 2c, the polarization dependence of SHG in the hBN-on-SiO₂/Si platform looks totally different for 18 nm and 36 nm thicknesses. What is the underlying physical mechanism behind this difference? Are these results for odd layers or even layers?
6. In Line 253, the authors show that the enhancement in twisted hBN layers can reach an enhancement of 357 times, which is very impressive by using gold only. However, the physical mechanism is not well described. Is this enhancement compared to the off-gold AB stack? For the AB stack where the inversion symmetry is broken, the dominant contribution should be dipolar. How does the gold bring such a high enhancement?
7. In Fig. 1c, the authors present the SHG contributions of dipole and quadrupole moments for the gold substrate. I recommend including the results for the SiO₂ substrate for a direct comparison.
8. What is the operating bandwidth for the SiO₂/Au/SiO₂/Si structure where the Fabry Perot effect is involved? Is the operating bandwidth reduced as compared to the hBN-on-Au structure?
9. Please correct the typos in citing the figures: (1) Line 203, Fig. 1c -> Fig. 2c. (2) Line 412, Fig. 4f -> Fig 5f.

Reviewer #2

(Remarks to the Author)

The authors have presented very detailed responses to my questions, including new results and new explanations added to the manuscript, and have resolved my main concerns about the manuscript. The results and the main message of the paper are now much clearer. I am now convinced that the results have novelty and will be of interest to the field. Hence, I can now support the publication of this work in Nature Communications.

I only have a few comments that I think the authors should consider in their final version:

-In line 83 of the manuscript the authors say „hBN, which is the most widespread 2D material today“. I think such statements are rather subjective. Taking out the word „most“ will resolve the problem.

-In line 158, the authors said „Furthermore, because the global enhancement of dipolar and quadrupolar contributions described above stems from two general metal properties“. It is not clear here what the two general metal properties are.

-In the last paragraph of the „Methods“ section „Sample preparation“, lines 468-473, I think the figure numbers should be updated. I think the description there refers to the figure numbers from the previous version of the manuscript.

-In lines 3-5 for page 6 of the Supplementary file, the authors say „This enhancement is demonstrated in Figure 2c of the main text. This enhancement is demonstrated in new Figure 2a of the main text.“ I think these sentences need to be updated for the new figure numbers. Probably there is also a repetition in sentences, and also the word „new“ in there is probably unnecessary. Overall, I suggest the authors look at the figure numbers in the texts again, to make sure there is no inconsistencies with the many new added figures.

-After equation (30) in the supplementary, the authors say „The propagation wavevector of matrices Φ for 3ω is $3k_0$ and the refractive indexes need to be reevaluated accordingly,“. I am not sure where the third harmonics are coming into this calculation. If it is an error then it should be corrected. If not, maybe a short explanation helps, since I cannot see where the third harmonic is getting into the calculations.

-In line 508-510, the definition for calculating the lower bound for the enhancement factor of SPDC seems rather an unusual definition as an enhancement factor. It would be good if the authors can add a short explanation there about why this definition is used for enhancement factor and what it means.

-In line 69 of the main text, the authors say „See table S2“. I think here they meant table S1, which makes sense compared to the context of the sentences before. I suggest to the authors to carefully examine all these naming for tables and figures in the text to make sure they are referred to correctly.

-I think the formula (1) in the supplementary is from the following reference [<https://doi.org/10.1364/OL.44.005792>], where as I understand they had hBN on SiO₂ substrate. I do not think it is valid to use that formula for evaluating $\chi^{(2)}$ values for systems of 2D materials on different types of substrates, especially ones like in this work that show extra field enhancement effects. I assume the data now in table S2 for $\chi^{(2)}$ values of hBN and NbOCl₂ are based on this formula. If so, maybe the authors can say more clearly in that first supplementary section with formula (1), which of their experimental data for hBN and NbOCl₂ are actually used to find these $\chi^{(2)}$ value. I am not sure if this formula is meant exactly for finding the bulk $\chi^{(2)}$, since it is used in [<https://doi.org/10.1364/OL.44.005792>] for finding the surface $\chi^{(2)}$ for hBN. So, the authors should explain how they derived bulk $\chi^{(2)}$ values for table S2 based on formula (1) (maybe state if they used an effective thickness) and state explicitly which of their experimental data they are using for this task.

-The caption for figure S11 in the supplementary needs to be updated. It refers to a figure in the main text that seems not to be there now. Also, it compares to the result on gold film, which is not shown in the same figure, and it would be good to remind the reader in that caption where that result is.

- Caption of figure S12 says „Wavelength dependent of SHG measurement from material on SiO₂/Au configuration.“ The authors should mention in this caption which material is that, hBN or NbOC12, and for which SiO₂ thickness.
- In caption of figure S13, for (a), is that a microscope image? That should be mentioned in the caption.
- In caption of figure S17, for (b), the authors should mention what is the dielectric and what is its thickness.

Reviewer #3

(Remarks to the Author)

As it has been improved a lot, I recommend that this manuscript be accepted for publication in Nature Communication.

Version 2:

Reviewer comments:

Reviewer #1

(Remarks to the Author)

The authors have thoroughly addressed my comments. I recommend accepting this manuscript for publication in Nature Communications.

Reviewer #1 (Remarks to the Author):

General comments:

The authors propose and experimentally demonstrate an interesting and elegant approach to boost the nonlinear efficiency in van der Waals material, especially for materials with centrosymmetric crystalline structures such as hBN. This is achieved by simply transferring nonlinear materials onto a metal film. Both SHG and SPDC processes can be enhanced by more than two orders of magnitudes. Using this technique, they have demonstrated the photon-pair generation from an NbOCl₂ flake of only 16 nm, establishing it as the thinnest SPDC source so far. While this work appears very interesting, I would like to invite the authors to address the following concerns before recommending it for publication.

General response: We thank you for summarizing our work and recognizing its significance. We also thank you for providing insightful comments and raising some questions that have contributed to improve our manuscript and clarify certain conceptual points. Find below a detailed answer to each of your questions.

Comment #1.1: The authors claimed that the unique advantage of this material-on-metal approach is broad-band operation, as compared to other approaches using nanocavities or metasurfaces. However, can similar nonlinear efficiency be achieved with nanocavities or metasurfaces with low-quality factors, which also allows broad-band operation?

Response #1.1: We thank you for raising this question, which was not sufficiently addressed in the first version of the manuscript (given that all three referees raised it somehow).

We acknowledge that similar nonlinear efficiency can potentially be achieved using nanocavities or metasurfaces with low-quality (Q) factors, which also enable broadband operation. However, our material-on-metal approach offers several distinct advantages:

- (1) Our heterostructures are much easier to fabricate than resonant nanocavities or metasurfaces, whose geometric parameters need to be finely tuned to match the emitter wavelength.
- (2) The operational bandwidth of our device spans a wide wavelength range (610 nm to 1100 nm), which is considerably broader than the values reported for low-Q cavities or metasurfaces, when considering similar enhancement factor. Further details on this aspect are provided in Response #1.2.

To provide a quantitative and clear benchmark with other strategies, we have added a new section in the Supplementary Information “*Comparison of enhancement factor and operation bandwidth among different cavities and metasurfaces with relatively low Q*”

factor” comparing our figures of merit with those achieved thanks to low-Q cavities or metasurfaces. The new section reproduces the text and Table shown below in *italic*:

“*Tapajyoti et al. reported that significant SHG can be achieved with amorphous Selenium (Se)-based chalcogenide metasurfaces by exploiting the coupling between lattice and individual particle resonances. At the resonant wavelength of $\lambda = 700$ nm, where field enhancement is maximal, the SHG signal normalized to the incident power shows a 100-fold increase over the nonresonant region ($\lambda = 800$ nm) (Figure R1a). The quality factor of this structure is 175. Figure R1b reproduces the study from Figure R1a but with an increased period of 470 nm and the same particle size of 350 nm, demonstrating broader resonances with lower quality factors, resulting in a two-fold reduction of the field intensity and SHG emissions.*

Figure R1: Resonantly Enhanced Second Harmonic Generation in (a) critically and (b) strongly coupled conditions. Reproduced from 10.1515/nanoph-2021-0277.

Using lithium niobate (LN) membrane metasurfaces, Lun et al. achieved a SHG efficiency of 2.0×10^{-4} , which is significantly higher than previous LN implementations, with a narrow operational bandwidth of 10 nm and a Q factor of 75. Alternatively, Lucca et al. demonstrated high-Q resonances across the visible spectrum using quasi-bound states in the continuum (qBICs) on h-BN and leading to a 100-times enhancement, though this requires adjustments in resonator shapes for each wavelength.

Mudassar et al. reported broad wavelength-dependent SHG enhancements in MoS₂ metasurfaces (690 nm – 735 nm; 825 nm-950 nm) without specifying the enhancement factors. Flavia et al. demonstrated a nonlinear optical response over a broad spectral range of 350 nm to 550 nm using a new type of metal oxide barium titanate-based nonlinear metasurface. However, the absolute enhancement peaks at a relatively low value of 16. Anna et al. have reported MoS₂ nanodisks engineered to couple Mie resonances with C-excitons, which enables 23-fold enhancement of SHG intensity compared to monolayer MoS₂. This enhancement, while covering a broad wavelength range, is strongly wavelength-dependent and peaks at a 23-fold increase around 900 nm.”

The figures of merit discussed above have been now summarized in Table S1 of the new Supplementary Information, so that the reader can easily compare between those works (mostly based on metasurfaces) and ours. The table is shown below as Table R1:

Material	Structure	Enhancement factor	Operation bandwidth	Q factor	Ref. (in SI)
a-Se	MS	100	340 nm- 360 nm	175	[4]
Lithium Niobate	MS	80	470 nm – 480 nm	75	[5]
hBN	BIC	100	Narrow	300	[6]
MoS₂	MS	-	690 nm – 735 nm; 825 nm-950 nm	Low (not mentioned)	[7]
poly-BaTiO₃	MS	16	350 nm – 550 nm	-	[8]
MoS₂	nanodisk	1-23	400 nm – 500 nm	Low (not mentioned)	[9]

Table R1: Comparison of enhancement factor and operation bandwidth among different cavities and metasurfaces with relatively low Q factor.

- (3) From the fundamental point of view (i.e. beyond the pure quantitative benchmark), the use of metal-based heterostructures exploits not only the enhanced electric-field intensity, as the photonic structures presented in the previous paragraphs, but also the enhanced electric field gradient, induced by the zero electric field imposed by metals close to their surfaces. To clarify this point, which is essential to understand one of the originalities of our work, we have added a completely new section in the main manuscript entitled “***General strategy for enhancing SHG in centrosymmetric and noncentrosymmetric thin materials***”, as detailed in the answer to your Comment 3.

Besides, we have modified the introduction of the main manuscript to draw explicitly a parallelism between our approach and the use of low-Q cavities for broadband operation, but also indicating from the introduction that besides the enhancement of the electric field intensity our method relies on a second ingredient (i.e. the electric field gradient), which is related to the quadrupolar contribution to the nonlinear polarization. The text added in pages 4 and 5 of the main manuscript is shown in *italic* below:

“Recently, efforts have been made to enhance the nonlinear effects in ultra-thin van der Waals materials either by breaking the crystal inversion symmetry through twisted flakes

or by coupling them to photonic structures. In the case of the twisted flakes approach [21-23] one must resort to intricate flakes manipulations (e.g. the use of micro rotators), which is complex and renders the approach difficult to implement and incompatible with other basic functionalities. In the case of photonic resonators, *two options can be considered. First, one can employ low cavity quality factor (Q) photonic structures, including metasurfaces and dielectric resonators, which enable a moderate enhancement but rather broadband operation (See Table S1) [24-29]. Alternatively, one can resort to high Q resonant cavities that allow, potentially, for larger enhancements but sacrificing the broadband width due to their resonant character [26, 30]. This spectrally narrow enhancement prevents exploiting the full potentiality of subwavelength light sources, which can display a broad emission bandwidth as a result of the relaxed phase matching condition [14]. Thus, ideally one would like to merge the advantages of both approaches without necessitating complicated clean-room processes.*

Herein we develop an innovative *and simple* approach to enhance nonlinear processes in ultra-thin materials via *the engineering of the field distribution around the nonlinear material. This approach is independent of the material symmetry and, thus, of general applicability.* We utilize nonlinear material/metal and nonlinear material/dielectric/metal heterostructure configurations to modify appropriately the electric field distribution at the van der Waals materials position. *The roles played by our heterostructures are to enhance the intensity of the electric field while keeping a broadband response, similar to low Q cavities, and to enlarge the electric field gradient, which acts a second sizeable source of polarization. This second ingredient is essential to achieve a comparable nonlinear response between thin films with odd and even layers in materials with AA' stacking. To illustrate the general applicability of our heterostructures we chose two Van der Waals materials: hBN, which is the most widespread 2D material today and whose centrosymmetric or non-centrosymmetric character is monolayer number dependent, and NbOCl₂, which is a van der Waals material exhibiting one of the largest nonlinear responses to date (Table S2).*

We observe a giant enhancement of second harmonic intensity on both h-BN, i.e. from some nanometres to tens of nanometre-thick layers, and NbOCl₂ flakes. Leveraging nonlinear transfer matrix simulations [23, 24] we have evaluated quantitatively the SH intensity on different heterostructures and substrates, *getting insight into the interplay between dipolar and quadrupolar contributions. Interestingly, for the SHG in h-BN layers the interplay between the two contributions results in a nonlinear intensity almost independent of the monolayers parity, erasing thereby one practical limitation.*”

Comment #1.2: Also, what is the operation bandwidth for material-on-SiO₂/Au configuration the authors introduced in Fig. 4 to further improve the enhancement? Does higher enhancement imply a narrow bandwidth?

Response #1.2: Thank you for inquiring about the operation bandwidth for our material-on-SiO₂/Au configuration.

To address your question, we have conducted additional experiments to measure the SHG response of h-BN on SiO₂/Au substrate across various pump wavelengths and performed the associated numerical simulations. As illustrated in the Figure R2a below, the operation bandwidth of the SiO₂/Au configuration remains broad, despite the observed large SHG signals. Additionally, we simulated an 8 nm h-BN layer on both SiO₂/Au and SiO₂/Si configurations, as shown in Figure R2b. The results reveal an enhancement across a wavelength range from 610 nm to 1100 nm, corresponding to an enhancement bandwidth exceeding 490 nm.

Figure R2: (a) Experimental wavelength dependent SHG measurements from material on SiO₂/Au configuration. (b) Simulated wavelength dependent SHG measurements from material on SiO₂/Au and SiO₂/Si substrates.

These new measurements are referred to in the new main manuscript (page 18) and the figure has been added into a new section “**Broadband SHG on SiO₂/Au configuration**” in the Supplementary Information, where it appears as Figure S12:

“To experimentally validate our findings, we transferred an 8 nm h-BN flake onto a 64 nm thick SiO₂ layer deposited on gold. The measured two-orders of magnitude enhancement of the SH signal compared to the same flake deposited directly onto gold (Figure 5d) confirms the efficiency of the SiO₂/Au structure in enhancing the nonlinear response of very thin h-BN, *which still remains broadband (Figure S12).*”

Comment #1.3: Additionally, whether a large bandwidth is good or not depends on the specific application; for example, quantum memory of single photons requires extremely narrow bandwidth.

Response #1.3: Thank you for your valuable observation regarding the application-specific advantages of large and narrow bandwidths. As you point out, quantum memory of single photons is one such application where an extremely narrow bandwidth is crucial to ensure high fidelity and efficiency in photon storage and retrieval. In contrast, broadband capabilities are crucial for a variety of applications that benefit from robust operation across a wide range of frequencies. These include broadband optical parametric amplifiers and supercontinuum generation, which are vital for advancing optical technologies. Moreover, there is growing interest in ultra-thin entangled photon sources for integrated quantum computing and communication devices. Particularly, frequency entanglement, which utilizes the relaxed momentum conservation inherent in thin films, is increasingly recognized for its potential in encoding quantum information into many different channels. In such systems, enhancing the generation of photon pairs across a broad spectrum of frequencies is crucial. Considering this and your notice on specific applications where narrow band is required, we have included a new paragraph in the Conclusions section (page 21), with the new text shown below in *italic*:

“...The desired operational bandwidth of a nonlinear optical system is highly dependent on the specific application. For instance, quantum memory of single photons requires an extremely narrow bandwidth to ensure high fidelity and efficiency in photon storage and retrieval. This contrasts with other applications, such as broadband optical parametric amplifiers and supercontinuum generation, which rely on robust operation across a wide range of frequencies and benefit from broader bandwidths. In addition, the growing interest in ultra-thin entangled photon sources, particularly for integrated quantum computing and communication devices, highlights the importance of broadband capabilities. Frequency entanglement, which leverages the relaxed momentum conservation in thin films, is increasingly recognized for its potential in encoding information. In such systems, it is essential to enhance the generation of photon pairs over a broad spectrum of frequencies to maximize entanglement efficiency. Our material-on-metal configurations are particularly well-suited for such broadband operations, with at least 100 nm large bandwidth. The broadband enhancement of quantum nonlinear processes was demonstrated by achieving a ten-fold increase....”

Comment #2: Following the first comment, the authors cited references 26 and 29 to support their claims about bandwidth limitations imposed by nanocavities or metasurfaces. It appears these two references are not related to photonic resonances. Are there any previous works that have demonstrated enhanced nonlinear processes with nanocavities or metasurfaces?

Response #2: We apologize for mixing the references in the initial manuscript.

We have now corrected this erratum and replaced them by the new references [30] and [31], which discuss SHG enhancement of monolayer TMD on metasurfaces and microcavities, respectively. In the first reference, Nils *et al.* report that a WS₂ monolayer combined with a silicon metasurface hosting bound states in the continuum (BICs) exhibits more than 3-orders of magnitude enhancement in second-harmonic intensity compared to a WS₂ monolayer on a flat silicon film of the same thickness. However, this enhancement is limited to a narrow range around the resonance wavelength of 832 nm. In reference 31, Jared *et al.* reported enhanced SHG from monolayer MoS₂ embedded within an all-dielectric Fabry-Perot microcavity that is resonant at the pump wavelength 818 nm.

Besides the two references 30 and 31, we have added more previous works and summarized them in the table R2.

To ease the quantitative comparison between our approach and previous ones, we have now included in the Supplementary Information a new section entitled “*Comparison of SHG enhancement factors enabled by coupling to resonant photonic structures*” where we summarize the SHG enhancement achieved in previous realizations.

Material	Structure	Enhancement factor	Operation wavelength	Ref. (in SI)
Lithium Niobate	MS	80	470 nm – 480 nm	[5]
WS₂	gratings	400	860 nm	[20]
WS₂	MS	1140	832 nm	[10]
MoS₂	microcavity	10	818 nm	[11]
MoS₂	MS	35	850 nm	[21]
WS₂	MS	100-1000	810 nm	[22]
MoS₂	nanowire	140	800 nm	[23]
WS₂	MS	10	1240 nm	[24]
hBN	BIC	100	Tunable with scaling factor	[6]

Table R2: Enhancement of SHG with cavities or metasurfaces with low bandwidth.

Comment #3.1: The authors have discussed the physical mechanism for the enhancement enabled by the material-on-metal approach. If I understand correctly, there are two contributions: (i) field enhancement for the signal/idler photons; and (2) larger electric field amplitude gradient. It appears that both of these contributions are important for the nonlinear enhancement of centrosymmetric materials such as hBN, while the enhancement in non-centrosymmetric material such as non-centrosymmetric material NbOCl₂ is due to contribution (1) only. If this is the case, it should be discussed more clearly. More importantly, the physics behind field enhancement in contribution (1) is not explicitly provided while the authors have mentioned boundary conditions. Is it simply the Fabry-Perot effect?

Response #3.1: We appreciate your careful reading of our manuscript and your insightful observation regarding the physical mechanisms underlying the enhancement effects. You have correctly identified the two primary contributions to the observed enhancement: (i) field enhancement for the signal and idler photons and (ii) the larger electric field amplitude gradient.

The enhancement described in contribution (1) leverages the inherently high reflectivity of metals across a broad wavelength range in the visible and infrared (IR) spectrum. This behavior is analogous to low-Q Fabry-Pérot cavities formed by Bragg mirrors with a small number of pairs or low refractive index contrast.

In response to your comment, as well as a related suggestion from the second referee, we have added a new section in the main manuscript titled “*General strategy for enhancing SHG in centrosymmetric and noncentrosymmetric thin materials*”. This section provides a detailed discussion on the physical origin and role of each enhancement mechanism (i and ii), offering a clearer and more systematic explanation. Besides, we have modified Figure 1 to quantify the absolute effect of each of these physical mechanisms.

The new section in the main manuscript is entitled “*General strategy for enhancing SHG in centrosymmetric and noncentrosymmetric thin materials*”, which we reproduce below (the added text appears in *italic*):

“General strategy for enhancing SHG in centrosymmetric and noncentrosymmetric thin materials

The nonlinear optical processes in a material are governed by its electrical polarization, which is given by a power series of the electric field amplitude. Beyond the linear term [26]:

$P_{NLO} = P_{2\omega} + P_{3\omega} + \dots = \chi^{(2)}E_{\omega}E_{\omega} + \chi^{(3)}E_{\omega}E_{\omega}E_{\omega} + \dots$, where $\chi^{(2)}$ is the second-order susceptibility tensor. $\chi^{(2)}$ is at the basis of SHG, which not only holds potential for creating nonlinear devices but also is being employed extensively by the 2D community for non-destructive

lattice orientation identification using low pump powers [4, 27, 28]. In the classical SHG process, two photons of frequency ω interact inside the nonlinear material to give a single photon of frequency 2ω . As indicated above, the SHG intensity is related to the pump electric field by the second-order susceptibility tensor that, in turn, can be expanded in terms of its dipole and higher multipole moments. To leading order, we can thus write $P_{2\omega} = \chi_d^{(2)}:E_\omega E_\omega + \chi_q^{(2)}:E_\omega \nabla E_\omega + \dots$, where $\chi_d^{(2)}$ and $\chi_q^{(2)}$ represent the dipolar and quadrupolar moments of $\chi^{(2)}$. Note that the dipolar term couples only to the electric field *amplitude (in fact to the electric field intensity)*, while the quadrupolar term couples to the electric field amplitude and its gradient (i.e. the spatial variation of the electric field). *For a more detailed analysis of this expansion and the quantitative relationship between the dipole and quadrupole moments see note “Dipole moment and Quadruple moment” in the supplementary information.*

Our strategy consists thus in finding simple designs that maximize the tradeoff between the electric field amplitude at the nonlinear-material location and the gradient of the electric field at that same position, as illustrated in Figure 1a. To do so we exploit two well-known facts of metals: first, they display a large reflectivity over a broad wavelength range in the visible and infrared (IR), comparable to Bragg mirrors with few number of pairs or with small refractive index contrast between the Bragg materials (i.e. those forming low Q Fabry-Perot cavities) [37]; second, by imposing a near-zero electric field at their surface, they enable to create a strong electric field gradient near the metal, far exceeding the field gradient in thick h-BN or NbOCl₂ induced by below-bandgap absorption or that created at dielectric interfaces.

To illustrate quantitatively the enhancement of both magnitudes, in Figure 1b (left axis) we have plotted the electric field intensity inside 100 nm-thick h-BN deposited on gold and on standard SiO₂/Si wafers. Due to the relatively large reflectivity of the h-BN/gold interface compared to the h-BN/SiO₂ interface (see Table S5 in SI), the electric field magnitude (at the pump wavelength) increases by a factor 3-5, providing a stronger dipolar contribution to the nonlinear polarization. Interestingly, the electric field gradient does also become larger when considering h-BN deposited on gold (right axis in Figure 1b), with an increase that can amount up to a factor ten compared to h-BN/SiO₂. This increased gradient, combined with the enhanced electric field amplitude, magnify the quadrupolar contribution to the nonlinear polarization.

Still, one can wonder how important the quadrupolar contribution can be, in absolute terms, compared to the dipolar one. In centrosymmetric materials, for which the dipolar contribution vanishes due to symmetry constraints, the quadrupolar contribution is obviously the dominant one, and it enables to observe SHG in sufficiently thick materials across which a field amplitude gradient can be established. On the other hand, in non-centrosymmetric materials the dipolar contribution is in general considered to be the dominant one. To illustrate the actual magnitude of the quadrupolar contribution enabled by the use of metals, we used the nonlinear transfer matrix approach [23] to calculate the individual contributions to the SHG intensity of h-BN displaying an

odd number of monolayers (i.e. for a non-centrosymmetric BN) deposited on gold. Figure 1c indicates that when using gold as a substrate, the quadrupolar response in h-BN is just a factor 2 to 3 times smaller than the dipolar response. The fact that both contributions are of the same order of magnitude highlights the potential effect of the quadrupolar moment in modifying the nonlinear response of a given material, particularly of centrosymmetric materials, where the quadrupolar contribution dominates. Furthermore, because global enhancement of dipolar and quadrupolar contributions stems from two general metal properties, its effect is observed when depositing the nonlinear active material on a variety of metals (Figure S1, S16 and S17).”

Figure 1 | Electric field distribution in centrosymmetric-nonlinear material/metal heterostructures. **a**, Schematic of incident light distribution on gold film (left) and SiO₂/Si substrate (right), showing a magnified electric field amplitude and gradient on the gold film. *Inset: Illustration of the SHG process, where two photons are combined to generate a single signal photon.* **b**, The magnitude of pump field intensity distribution (red) and the gradient of electric field (blue) in a 300-monolayer h-BN flake on gold (line) and SiO₂/Si (dashed line). The dark area is the substrate. **c**, The simulation results of dipolar (green) and quadrupolar (orange) contributions to the magnitude of SHG electric field of h-BN flakes with odd number of monolayers.

Comment #3.2: Is the reason that the material-on-SiO₂/Au approach provides higher enhancement due to the higher Q of the Fabry Perot cavity? Is there such a photonic resonator effect in your approach that you wanted to avoid?

Response #3.2: We appreciate your thoughtful question. As discussed in Response #3.1, the material-on-SiO₂/Au stack does indeed act as a low-Q planar dielectric FP cavity, formed between the material and the metal.

As we detail below, h-BN with odd layers matches well with the FP cavity resonance conditions, whereas h-BN with even layers does not. Therefore, rather than attributing the higher enhancement solely to the higher Q-factor of the FP cavity, we conclude that the observed enhancement in the material-on-SiO₂/Au approach results from a combination of FP cavity resonance and the electric field gradient inside the material.

To clarify this point, we discuss these conditions separately:

- (1) Fabry-Perot cavity resonant condition: The resonance condition for a Fabry-Pérot cavity is written as: $\frac{4\pi nL}{\lambda} + \varphi_1 + \varphi_2 = 2\pi m$, where m is an integer, L is the length of the cavity, n is the refractive index of cavity, λ is the wavelength of waves, and φ_1 and φ_2 are the reflection phases at each interface. In our case, taking into account that the phase shifts at the opposite interfaces (φ_1 and φ_2) differ roughly by π , which leads to resonances whenever the cavity length is an odd multiple of $\lambda/4n$, the cavity length required for a Fabry-Perot resonance is on the order of 153 nm for the 890 nm pump and of about 77 nm for the 445 nm SH waves. To further support this point we have included a table in the supplementary materials detailing the reflectivity coefficients and reflectivities at the various interfaces within our structure, calculated using the formulas $r = \frac{n_i - n_j}{n_i + n_j}$ and $R = |r|^2$, where n_i and n_j are the refractive indices of adjacent materials.

Interface	n_i	n_j	Reflection coefficient r	Reflectivity R	Transmission T
BN/Au	2.12	0.13+5.66 i	-0.7428-0.6469 i	0.97	0.03
BN/SiO₂	2.12	1.47	0.18	0.03	0.97
SiO₂/Si	1.47	3.67	-0.43	0.18	0.82

Table R3: Reflectivity and transmission rate at different interfaces.

- (2) For thin h-BN with an odd number of layers: Nonlinear transfer matrix simulations indicate two optimum SiO₂ thicknesses, around 76 nm and 166 nm. These values correspond very close to the resonant condition for the SH waves and the pump amplitude (maximizing the dipolar contribution), respectively. Optimizing either of these two effects results in a similar computed SH signal and enables the maximization of the SH signal for monolayer-

thick h-BN. It indicates that the FP cavity resonance plays a critical role in enhancement for odd-layered h-BN.

- (3) For thin h-BN with an even number of layers: The optimum calculated SiO₂ thickness is about 121 nm, far from the cavity resonant conditions for either wave (SH or pump). Therefore, the dominant contribution to the polarization is the quadrupolar term (due to the electric field gradient) rather than the FP cavity. Besides, for h-BN with even number of layers, small thicknesses (on the order of some monolayers) do not maximize the SH signal. To maximize the signal, the gradient of the field needs to be significant and h-BN with a non-negligible thickness is necessary. Nonlinear transfer matrix simulations indicate that the optimum thicknesses for even number of monolayers h-BN are 51 nm and 59 nm, for the SiO₂ and h-BN, respectively.

The above discussion has now been introduced in the main text, shown as *italic* below:

Page 17: “To strengthen the nonlinear response of films down to several to tens of monolayers thick, we have designed a structure consisting of a SiO₂/Au/Si wafer stack that incorporates a planar dielectric *Fabry-Perot* cavity between the material and the metal (Figure 5a).”

Page 17: “*Note that for the current Air/SiO₂/Gold configuration, if we assume that the reflection phases at each of the two interfaces differ roughly by π , the minimum resonant cavity thickness is not $\lambda/2n_{cav}$ but $\lambda/4n_{cav}$. This resonant condition would correspond to a thickness in the order of 153 nm for the 890 nm pump and about 77 nm for the SH waves. For a monolayer-thick h-BN (i.e. for vanishingly thin h-BN with odd number of monolayers), the optimum SiO₂ thickness are around ~76 nm and 166 nm, so very close to the resonant condition for the SH waves and close to the condition for a maximum of the electric field pump amplitude (maximizing the dipolar contribution), respectively. Optimizing either of these two effects results in a similar computed SH signal. Furthermore, as shown in Figure 5b, for h-BN containing odd-number of monolayers and deposited on a SiO₂/gold substrate, the thickness maximizing the SH can be extremely thin (in the order of monolayers). On the contrary, for h-BN with vanishingly small even number of monolayers (Figure S11), the optimum SiO₂ thickness (121 nm) lies in between the resonant conditions for the SH waves and the pump, and the actual h-BN thickness does not maximize the SHG response. Indeed, to maximize the quadrupolar moment controlling the response of centrosymmetric h-BN we need to have a nonnegligible h-BN thickness, leading to an optimum SiO₂ thickness of 51 nm for an optimum h-BN thickness of 59 nm.*”

Comment #4: The authors performed the classical SHG measurement with hBN and SPDC with NbOC12. Have the authors tried to measure SPDC from hBN? How is the fluorescence from hBN? Does NbOC12 perform better than hBN in terms of SPDC? Which one provides higher nonlinearity? What are the unique benefits of hBN?

Response #4: Thank you for your questions regarding our choice of materials for the classical SHG and SPDC measurements. Regarding SPDC measurements on h-BN, we have indeed conducted new experiments but found it challenging to obtain a significant SPDC signal from h-BN due to two main reasons:

- 1) h-BN's second order susceptibility is approximately ten times lower than that of NbOCl₂, which significantly limits its intrinsic efficiency.
- 2) h-BN can be centrosymmetric (depending on the parity of the number of layers), which makes it difficult to insure non-centrosymmetry as thickness increases, which is required for effective SPDC.

Figure R3: (a) Spectrum from h-BN flakes with 409 nm pump. Note that we use a long-pass filter at 750nm. (b) Spectrum from NbOCl₂ flakes excited with 409 nm pump.

Besides, when pumped with a 409 nm laser, the fluorescence of the h-BN employed in the current study shows an emission band ascribed to deep defects across the wavelength range where we could expect SPDC, as shown in Figure R3(a) (note that we used a 750 nm long-pass filter for the measurement). While this emission band might be beneficial for applications requiring defect-based emission (e.g. for quantum sensing applications), it will provide a non-negligible background for SPDC experiments that will degrade the signal-to-noise ratio or even, as it seems to be our case, completely mask any potential generation of photon pairs. Therefore, for SPDC we thus opted for NbOCl₂ due to its non-centrosymmetric crystal structure and intrinsically-large nonlinear coefficients.

Despite these challenges, h-BN remains a valuable nonlinear optical material due to its high mechanical strength, thermal stability, and transparency across a broad range of wavelengths, from ultraviolet to infrared. Importantly, our study also aims to showcase that our method can be applied to both bulk centrosymmetric and non-centrosymmetric materials. For example, h-BN with even layer numbers, which typically show low SHG response due to centro-symmetry, exhibited trends and responses comparable to those with an odd number of layers when combined with our

enhancement approach. This underscores our technique's versatility and potential in enhancing the nonlinear optical responses of various materials.

To clarify the material we chose, we have added discussions about the physics underlying the enhancement in the main text:

Page 4: *“To illustrate the general applicability of our heterostructures we chose two Van der Waals materials: hBN, which is the most widespread 2D material today and whose centro- or non-centrosymmetric character is monolayer number dependent, and NbOCl₂, which is a van der Waals material exhibiting one of the largest nonlinear responses to date (Table S2).”*

Comment #5.1: In Fig. 1c,d, the authors presented that the nonlinear tensor can be modified by the metal film. Does this modification depend on the hBN layer thickness? Is the modification consistent for all the samples-on-metal?

Response #5.1: We appreciate your insightful questions regarding whether the modification of the nonlinear tensor by the metal film depends on the hBN layer thickness and whether the modification is consistent across all samples-on-metal. To address these points, we performed additional polarization dependent SHG measurements on samples with two different thicknesses (36 nm and 65 nm) and analyzed the results as follows:

- (1) Rather than depending directly on the hBN layer thickness, we believe the modification of the nonlinear tensor depends primarily on the strain induced during sample fabrication and roughness of substrate. As a result, the strain can vary for samples of different thicknesses. For the 65 nm hBN sample, slightly larger inhomogeneities were observed compared to the 36 nm sample (Figure R4).

Figure R4: SHG measurement from 36nm and 65nm h-BN on Au film.

- (2) Consistency within the same sample: We found that the modification of the nonlinear tensor is consistent within the same sample. For a given sample the general trend in terms of maximum enhancement across different locations remains homogenous, and similarly for the asymmetry between the different intensity lobes

(Figure R5). To minimize the effect of the lobes asymmetry for different polarization angles, each point in the new Figure 2a (both on gold and on SiO₂) represents the SH intensity averaged over the six SH maxima in the polarization-resolved measurements.

Figure R5: SHG measurements from 65 nm h-BN on Au film.

- (3) Influence of surface quality and transfer procedure: The measurements discussed above suggest that to engineer the nonlinear tensor one requires a very strong control on the quality of the substrate surface and of the transfer procedure. In fact, the asymmetry between the lobes in the polarization-resolved SH measurements is systematically larger when h-BN is deposited on gold than when it is deposited on SiO₂. We ascribe this to the surface roughness, which is five times larger for the gold surface than for the SiO₂ (Figure R6).

The information has now been added to the main text, and polarization dependent SHG measurements as well as AFM images of the two surfaces have been included in the Supplementary Information. The new text is shown in *italic* below:

Page 10: “Noticeably, the polar plot of the SHG is fully symmetric on the flat SiO₂/Si substrate while an asymmetry is observed systematically on gold films. *This asymmetry degree varies, however, from location to location within the same flake (Figures S4 and S5). This observation is tentatively ascribed to the inhomogeneous strain induced by the irregularities of the gold film whose roughness, while in the nanometer scale, is five times larger than that of SiO₂/Si substrates (root mean square roughness of 2.4 nm Vs 0.5 nm, respectively). Indeed, to minimize the effect of*

the asymmetry on the polarization-resolved SHG response of h-BN on gold, each experimental data in Figure 2a corresponds to the SHG value averaged over the six maxima in the polarization-resolved measurements, both for data on gold and on SiO₂/Si substrates.”

Figure R6: (a) AFM image of gold film. (b) Height profile of the gold film along the red line indicated in (a). (c) AFM image of SiO₂ film. (d), (e) Height profile of the SiO₂ film along the red(d) and green(e) line indicated in (c). Root mean square roughness is defined as $\sqrt{\langle (h - \bar{h})^2 \rangle}$, where h is the height of each pixel.

Comment #5.2: The nonlinear tensor looks interesting, any potential for generating polarization entanglement?

Response #5.2: Regarding the potential applications in polarization entanglement, we recognize the potential of h-BN for polarization entanglement applications, despite its low second-order nonlinear response. In the nonlinear tensor, for a BN monolayer, the nonzero terms are:

$$\chi_{y'y'y'}^{(2)} = -\chi_{y'x'x'}^{(2)} = -\chi_{x'x'y'}^{(2)} = -\chi_{x'y'x'}^{(2)},$$

which indicates that with x polarized pump field, photon pairs with state $|xy\rangle$ and $|yx\rangle$ can be generated with equal probabilities. On the other hand, with y polarized pump field, photon pairs with state $|xx\rangle$ and $|yy\rangle$ can be generated with equal probabilities. This would enable to generate maximally entangled quantum states, which are Bell states $|\Phi^-\rangle$ and $|\Psi^+\rangle$. While the current response efficiency poses a challenge, the unique symmetry properties of the nonlinear tensor provide a strong foundation for further research into enhancing these materials for practical entangled photon generation.

Comment #6: In lines 227-228, it is stated that the “quantum SPDC generation rate is proportional to the classical SHG rate, the proportionality being controlled by the pumping conditions.” This is not true. While there is indeed a classical-to-quantum correspondence between SPDC and SFG (rather than SHG), it is not a simple proportional relationship. SPDC is related to the integration of all possible SFG processes over the whole spatial and spectral spectrum.

Response #6: We thank you for the critical observation and clarification regarding the relationship between SPDC and SHG. Upon reevaluation, we acknowledge that our original manuscript inaccurately described this relationship.

For classical processes such as SHG and SFG (sum frequency generation), energy and phase matching conditions are stringent, typically resulting in photon generation at specific frequencies determined by the pump frequencies. For the quantum SPDC process, the total energy of the photon pairs (signal and idler) must sum to the pump photon energy. However, the frequencies of the individual photons are not fixed and can vary across a broad spectrum, due to the relaxed phase-matching conditions in thin film. The classical estimation of the SPDC generation rates can be related to the SFG process by: $\frac{1}{\Phi_p} \frac{dN_{pair}}{dt} = 2\pi\Xi^{SFG} \frac{\lambda_p^4}{\lambda_s^3\lambda_i^3} \frac{c\Delta\lambda}{\lambda_s^2}$, where Φ_p is the SPDC pump flux, $\lambda_{x=p,s,i}$ are the pump, signal, idler wavelengths and $\Delta\lambda$ is the nonlinear resonance bandwidth at the signal/idler wavelengths [Optica 6.11 (2019): 1416-1422]. Therefore, SPDC is not appropriately described by a simple proportional relationship to SHG, as initially suggested, but related to the integration of all possible SFG processes over the whole spatial and spectral spectrum.

In response to this, we have revised our manuscript to clarify these differences:

“In ultrathin materials (largely subwavelength) phase-matching conditions can be assumed to be always fulfilled and, thus, one can expect SPDC to provide pairs of photons spreading over a broad wavelength range [14, 17, 18]. Furthermore, under certain conditions a correspondence between the SFG and the SPDC intensity can be established [39], meaning that the larger the SFG intensity is, the larger the SPDC rate one can expect. Because of these two reasons, our material-

on-metal approach based on broadband enhancement of SHG should be of use with subwavelength SPDC nonlinear materials.”

And one additional discussion section “SHG, SFG and SPDC” is added into the supplementary information.

Comment #7: The term “pump field” in the SHG measurement can be confusing as the same term is used for the SPDC process. It is better to use signal/idler photons or fundamental waves instead.

Response #7: Thank you for pointing out the potential confusion arising from our use of the term "pump field" in the context of both SHG and SPDC statements.

In response to your suggestion, we have reviewed the manuscript and agree that distinguishing between the terms used for SHG and SPDC processes will enhance clarity. We replaced the term “pump field” in the context of SPDC measurements with “fundamental wave to describe the initial high-energy photon source that is down-converted into signal and idler photons. This adjustment ensures that the terminology reflects the specific processes and avoids any overlap that might lead to confusion.

These changes have been implemented throughout the manuscript to ensure consistency and improve the reader’s understanding of the distinct processes involved in our study.

Comment #8: Fig. 3b seems unnecessary because Fig. 3c has included all the information needed.

Response #8: We appreciate your attention to the efficiency and clarity of our manuscript and agree with the importance of presenting a concise manuscript.

Initially, we included Fig. 3b to explicitly show the coincidence counts from the bare substrate, which is crucial for demonstrating that the SPDC signal is derived solely from the NbOCl₂ material, thereby confirming that there is no residual contribution from the substrate itself. Fig. 3c, on the other hand, is intended to illustrate the enhancement of SPDC signals when NbOCl₂ is used with and without the gold substrate under identical experimental conditions. This comparison is vital for showcasing the effect of the substrate on SPDC efficiency.

To address your concerns and enhance the clarity, we have revised our approach and combined these insights into a single, more comprehensive figure. This new figure (Figure 4b) now includes SPDC measurements from NbOCl₂ on gold, NbOCl₂ off gold, and of the bare substrate, enabling the reader to directly compare these scenarios within one cohesive visual figure (Figure R7). This

update has been incorporated in Page 14 in the main text to ensure a clearer understanding of the origin of SPDC signals (in *italic* below):

“The SPDC response from NbOCl₂ was first measured when transferred onto an SiO₂/Si substrate, showing a coincidence peak above the background when the time difference between the two detector channels is set to 0 ns, which is a signature of biphoton generation (Figure 4b). Interestingly, the NbOCl₂ flake on gold exhibits a ten-fold increase in the coincidence count rate.”

Figure R7: Coincidence count rate of NbOCl₂ on /off gold (orange/blue) and bare substrate (dark blue).

Comment #9: Fig. 3g presents the biphoton polarization dependence. However, SPDC rate (instead of CAR) vs. polarization angle is a more appropriate and standard measurement to confirm biphoton polarization. Also, this figure is not cited in the main text.

Response #9: Thank you for your insightful feedback regarding Fig. 3g. We appreciate your suggestion to use the SPDC rate versus polarization angle as a more standard measurement to confirm biphoton polarization, which is indeed a conventional approach.

In light of your suggestion, we have revisited our data presentation. We have now included measurements of the SPDC rate as a function of polarization angle.

Figure R8: Polarization dependent SPDC rate with both collection arms set parallel to the pump polarization.

Additionally, we acknowledge the oversight in not citing Fig. 3g within the main text of our manuscript. We have corrected this by including appropriate references to Fig. 3g (now is Figure 4g) in the sections discussing our experimental results:

Page 16: “Finally, we analyzed the coincidence count rate relative to the polarization of the fundamental waves. When rotating the polarization of fundamental wave and measuring SPDC in the co-polarized setting, the generated pattern exhibits a two-lobed symmetry (*Figure 4g*), consistent with the C2 crystal group characteristic of NbOCl₂ [32] and with our SHG experiments on NbOCl₂ (*Figure 5f*).”

Comment #10: Figure 3e shows the CAR vs. pump power. How does the SPDC rate depend on the pump power? I recommend making plots of CAR/SPDC rate vs. pump power (rather than showing different correlation plots under different powers).

Response #10: Thank you for your great recommendation on how to present the data in Figure 3e. We have updated our analysis and now include plots that directly explore the relationship between the CAR/SPDC rate and pump power. The new Figure R9 illustrates this relationship, showing that the photon pair coincidence rate scales linearly with pump power. Additionally, we've measured the CAR across varying pump powers, which demonstrates an inverse dependence on pump power, underscoring the photon pair generation process. These adjustments provide clearer insights into the dynamics of SPDC.

Figure R9: Coincidence count rate (left) and CAR (right) as a function of pump power.

In response to this, we have revised our manuscript (Page 16) to clarify these differences:

“...The observed biphoton *coincidence rate is linearly proportional to the fundamental wave pump power, as shown in Figure 4e, while the CAR is inversely proportional to it due to the increased*

background photons (Figure 4f), which is a typical indication of nonclassical photon pair emission [17]. Finally, ...”

In summary, we sincerely thank you for your valuable suggestions and insightful questions, which have greatly enhanced the clarity and depth of our manuscript. Your feedback has helped us refine our explanations of the underlying physical mechanisms and present key concepts with greater accuracy and precision. We believe that the revisions prompted by your suggestions have significantly strengthened the manuscript.

Reviewer #2 (Remarks to the Author):

General comments:

In their work “Giant boost of classical and quantum nonlinear processes in ultrathin van der Waals materials”, Lyu et al are investigating the intensity of second harmonic generation (SHG) and spontaneous parametric down conversion (SPDC) in thin-film material systems, namely hBN and NbOCl₂, that are placed onto or close to a gold surface and show that the efficiency of SHG and SPDC is enhanced.

Overall, I do not think the results are very novel (similar works done before, see below) and I also believe the impact for the community is also quite limited (the involved physics is rather trivial, see below). At the same time, from a scientific point of view, the manuscript is not well written. That means the analysis that is done is not clear and convincing, and the results are not clearly interpreted, and it is also not clear what is the main claim of the paper, in terms of the physics that is discovered. The manuscript is a rather chaotic gathering of different experimental and theoretical results, corresponding to different combinations of nonlinear material and photonic systems, where a clear and coherent understanding of the physics and interpretation of the results are not presented.

This also makes judging the impact of the paper rather hard, not knowing what the exact claim of the paper is. Hence, to some extent my judgment on the novelty and the impact of the paper is based on what I tried to infer myself by looking at the theoretical and experimental plots and trying to associate a clear physical explanation to the observed results. From my understanding, I could conclude that the observed effect is simply due to having a mirror on one side of the nonlinear material, which would increase the pump field intensity by a factor around 4 at some places and also allows for collecting light that is generated and is propagating in the forward direction without the mirror. This is a rather trivial effect. There are already many works done on hybridizing van der Waals/2D materials with resonator- or waveguide-based nanophotonic structure (see the following recent reviews [<https://doi.org/10.1039/D3NH00246B>; <https://doi.org/10.1038/s41578-023-00558-w>]), among which there are also works which are very similar to the works done here, with 2D materials placed on metallic mirror [<https://doi.org/10.1016/j.matt.2024.04.043>] and also within a cavity with dielectric Bragg mirrors [<https://doi.org/10.1002/adom.202300907>], both demonstrating enhanced SHG. This rather takes away the general novelty of this current work. From the point of view of physics discussed, it is clear that there is always a trade-off between the bandwidth (quality factor) of the resonance and its efficiency. The lowest limit of a bad cavity, is one where one mirror is missing in the cavity and you just have one mirror. So instead of getting an enhancement in efficiency that comes from many reflections in the cavity over a narrow bandwidth, one would get an enhancement caused by one reflection which would work over a large bandwidth. Such effects are very trivial, and pretty much are the cornerstone of nonlinear optics in the last many decades and very well understood. At the first look, I thought maybe the results come from a special interaction between the van der Waals material and the metal (which

would have been interesting), but that seems to not be the case, as the results have no special connection to having van der Waals materials or even having a metallic mirror. As I understand the paper, the same effects can be seen in any thin film nonlinear material and any type of mirror (Bragg mirror for example). With this, I do not think the result is very novel and impactful. Again, this is based on my understanding of the physics of the system, which was unfortunately not clearly offered in the manuscript.

General response: Thank you for your detailed evaluation of our manuscript. We appreciate the time you've taken to analyze our work and to provide thorough feedback. We recognize your concerns on (1) the novelty/significance of our work, (2) the clarity of analysis on the origin of enhancement and interpretation of results, (3) the coherence and organization, and (4) a number of additional details.

We fully agree that the main findings of the article were hidden behind a large number of experimental data and that the underlying physics was not clearly explained, nor highlighted. To clarify the physics on play, which combines enhancement of electric field (which your explanation above recalls) and the enhancement of the electric field gradient, and to present clearly the ingredients necessary to understand the observed enhancement factors, we have included a **completely new section** in the main manuscript entitled “*General strategy for enhancing SHG in centrosymmetric and noncentrosymmetric thin materials*”. Similarly, and with the goal of gaining in clarity, we have largely modified three of the five figures and provided, in the Supplementary Information, a large number of additional measurements (some carried out to respond to some of the questions raised by the referees) that sustain further our claims. Besides, to highlight the potential of our approach and to ease comparison between our results and those previously published, mostly based on the use of metasurfaces or microcavities, we have included in the Supplementary Material two tables providing either the bandwidth of the enhancement or the maximum enhancement achieved in previous works. This has further enabled us to correct many of the references that were wrongly used in the first version of the manuscript. Finally, much of the text in the main manuscript has been revised to improve the logic presentation of the results and the clarity. We hope this new organization and presentation of the manuscript, together with the new explanations and sections introduced, will clarify any misunderstanding and will prove useful for the community.

Below we provide a thorough and clear answer to each of your individual questions.

Nonetheless, below I give my detailed comments on what I see as the major issues in the manuscript. Although, even by implementing them, as I currently understand the physics of this system, I do not think that changes the general novelty and impact of the work. Nonetheless, here are my comments:

Comment #1: First a general comment, which is a problem that drastically reduces the quality of the manuscript. Overall, the authors are using references in a very uncareful way. For example, line 42-45: “Beyond classical nonlinear processes, van der Waals materials facilitate correlated photon pair generation by four-wave mixing (FWM) [14], optical parametric amplification and oscillation [15], and spontaneous parametric down-conversion (SPDC) [16-21].” Most of the references here are not really van der Waals materials. Specifically references [16] to [19] are either lithium niobate or III-V semiconductors. Reference [15], as far as I know, does not reach an OPO regime, but is rather an OPA. Also, in line 41-42, they say “Nonlinear processes have been demonstrated in TMDs [4-9], thus making it possible to realize ultrafast optical switching and parametric amplification [10-13].” Which one of references from [10] to [13] contain parametric amplification? Also, in line 45-48 “Notably, the implementation of SPDC in ultra-thin subwavelength materials offers a significant advantage by relaxing momentum conservation criteria [20, 21], thereby alleviating the size limitations imposed by bulky nonlinear crystals that limit their on-chip integration [22, 23].” References [22-23] do not seem to be relevant for the sentence before it. The authors seem to be talking about the advantage of not having phase-matching restrictions for nonlinear effects in thin materials, then they cite reference [22-23] in integrated lithium niobate and silicon, which does not seem to be about nonlinear effects. Line 64-65, they say “the spectrally resonant character of the enhancement limits their operation bandwidth [26, 29]”. Does reference [26] employ any resonant photonic structure? Also, line 146-147, reference [28] is not based on resonant nanocavities, it is based on phase-matching, in contrast to what is stated in the manuscript there. In line 149-150, references [35-37] do not show the SHG polarization dependence for hBN or TMDs, as far as I can see, so they do not relate to the statement before them. Overall, I think the authors should very carefully reexamine “all” of their references and make sure that they are mentioned in the correct context and with the correct explanation.

Response #1: We appreciate your detailed critical reading regarding the accuracy and relevance of the references cited in our manuscript. We acknowledge that this issue required meticulous revision to ensure accuracy and relevance.

In response, we have revised the corresponding sentences and citations in the main text (in *italic* below):

line 42-45 (in the previous version): We added one new citation in which Alessandro *et al.* devise a novel kind of phase-matching-free miniaturized parametric oscillator operating at conventional pump intensities based on TMDs: “*Beyond classical nonlinear processes, van der Waals materials have been reported to facilitate optical parametric amplification and oscillation [12, 13], and spontaneous parametric down-conversion (SPDC) [14, 15].*”

line 41-42 (in the previous version): We have deleted references [10], [11], [13] and added one new reference in which Kan *et al.* demonstrate a fiber all-optical phase shifter using few-layer 2D material tungsten disulfide (WS₂) deposited on a tapered fiber: “*Nonlinear processes have been*

demonstrated in TMDs [4-9], thus making it possible to realize ultrafast optical switching [10, 11].”

line 45-48 (in the previous version): We have updated the sentence to highlight the advantages of integrating two-dimensional materials, rather than emphasizing the absence of phase-matching restrictions. We updated the references to include a new study focused on the integration of van der Waals materials, along with two others discussing the characteristics of traditional nonlinear crystals with three-dimensional covalent bonding: *“Notably, the demonstration of SPDC in two-dimensional layered materials offers significant advantages over bulk nonlinear materials both in terms of integration, due to their bond-free relationship with the underlying substrates [16], as well as in term of relaxed phase-matching conditions, which need not be satisfied for subwavelength nonlinear active regions [17, 18].”*

line 64-65 (in the previous version): We have replaced the sentences with updated references, which discuss SHG enhancement based on metasurfaces and microcavities.

line 146-147 (in the previous version): As we have revised the main text, this sentence as well as the references are now removed.

line 149-150 (in the previous version): The corresponding reference has been replaced with a new one: Li, Y. *et al.*, Probing symmetry properties of few-layer MoS₂ and h-BN by optical second-harmonic generation. Nano letters, 2013. 13(7): p. 3329-3333: *“To further ensure that the SHG response arises from the h-BN we performed polarization-dependent SHG measurements, which exhibit a distinct six-fold symmetry (Figure 1c), in agreement with the D_{3h} point group of h-BN and some TMDs [4, 21, 38].”*

Comment #2: A comment about a main claim of the paper: The authors say in the abstract, and also in the manuscript something similar, that “The demonstration of SPDC in 16nm NbOCl₂ flakes establishes our device as the thinnest SPDC source so far.”. I am not sure if it is in the policy of Nature journals to make such claims. Nonetheless, I also do not think it is straightforward to say this device is only 16 nm thick, since you have to consider the thickness of your whole device, involving the metal and the SiO₂ thin films, and then you need to compare at least to some other state of the art works, including recent works on thin film lithium niobate and GaAs and also the corresponding metasurfaces in those platforms (see [<https://doi.org/10.1364/PRJ.474387>] as a place to start, but there are many works in these directions in the last few years).

Response #2: We appreciate your critical assessment of our claim regarding the thickness of our SPDC source. Your comments have prompted us to carefully review our claim, to specify what we

meant in a clearer manner, and to present in the Supplementary Information a quantitative comparison with the latest developments in the field.

We acknowledge that our initial claim in the abstract and manuscript, regarding the 16 nm thickness of the NbOCl₂ flakes, may have been misleading if the reader understood that we were speaking about the total system thickness (i.e. including the additional layers of metal and SiO₂). To prevent any misunderstanding, in the new version of the article we have emphasized that we refer to the thickness of the nonlinear medium:

Page 2: *“Notably, we demonstrate SPDC in a 16nm-thick NbOCl₂ flake integrated into the proposed structure.”*

Page 5: *“This constitutes the realization of SPDC with the thinnest nonlinear medium among currently reported SPDC sources (Table S4).”*

Page 20: *“To our knowledge, this structure features a nonlinear medium that is three times thinner than previously reported values [14], paving the way to the development of real ultra-thin integrated quantum devices.”*

For completeness, and in response to your suggestion, we have conducted a comprehensive review of recent developments in the field, summarizing the thicknesses of active layers used in comparable studies (Table R4). This comparison, which is now included in the revised supplementary file as Table S4, substantiates our claim that the NbOCl₂ flakes in our device constitute the thinnest nonlinear medium among currently reported SPDC sources.

Photon pair source	Active region thickness (nm)	Ref. (SI)
AlGaAs (Mie-type)	400	[25]
LN (QOM)	680	[26]
GaAs (quasi BIC)	230	[27]
LiNbO₃ (silica meta-grating)	304	[28]
GaP	400	[29]
3R-MoS₂	285	[30]
3R-WSe₂	350	[31]
NbOCl₂	46	[32]
NbOCl₂	1200	[33]
NbOCl₂	16	This work

Table R4: Comparison of thickness of active region in the photon pair sources.

Comment #3: This is in my opinion the most major issue: The physics of the nonlinear interaction is quite unclear and the analysis is very confusing. To put it very shortly, what is the role of $\text{chai}(2)_d$ and $\text{chai}(2)_q$? The authors start talking about them in line 115-116, and emphasize their importance and make it sound like the enhanced effect is somehow coming from an interplay of these two effects. Then in lines 125-128 they say “Our strategy relies in maximizing the electric field amplitude at the nonlinear-material location and shrinking down the scale at which electric field gradients are considerable, both effects being controllable thanks to the presence of a neighboring metal surface, which imposes strict boundary conditions on the electric field.” Which again emphasizes the physical importance of $\text{chai}(2)_d$ and $\text{chai}(2)_q$. Then the rest of the analysis and their simulations seem to not involve these quantities at all. The simulations seem to be based on nonlinearity of single monolayer hBN, which they are not including in the initial explanation of the physics. Is the dipolar and quadrupolar bulk nonlinearity somehow connected and can be derived from the single monolayer nonlinearity? If so, these relations should be rigorously derived and some actual numbers should be associated to these quantities. From my understanding, the connection between the physics explained in the motivation and the simulations are lost. At the

end, the authors look at many different platforms, and it is not clear in any of them, what is actually the source of enhancement.

We thank you for your insightful comment, which has highlighted the need for a clearer exposition of the underlying physics in our work. We apologize for any confusion caused by our initial explanation. **In Response #3.1**, we will elaborate on how the presence of metals influences the field distribution, and we will provide a detailed comparison of the dipolar and quadrupolar contributions within the material. **In Response #3.2**, we will clarify how the changes in the fields (both in magnitude and gradient) are quantitatively connected to the dipole and quadrupole moments, and how the quadrupolar and dipolar susceptibilities are related to each other.

Response #3.1: As indicated in the original manuscript and recognized by Reviewer 1 the observed enhancements arise due to two key factors: (i) field enhancement for the pump and signal photons, primarily due to the higher reflectivity at the nonlinear medium/metal interface than at the nonlinear medium/dielectric interface, and (ii) an increased electric field amplitude gradient. These two effects are due to the presence of the metal material. Specifically, metals exhibit high reflectivity across a broad wavelength range, similar to Bragg mirrors with few pairs or low refractive index contrast between the Bragg materials. Furthermore, the electric field at the metal surface (or close to it) goes to zero, creating a larger electric field gradient near the metal. To illustrate quantitatively these two facts, which are at the base of our findings, we have added a new panel in Figure 1 where we compare the electric field and the electric field gradient distributions in a 300-monolayer h-BN deposited on Au (solid lines) and SiO₂/Si (dashed lines), respectively, as shown below in Figure R10.

Figure R10: Electric field intensity distribution and its gradient in a 300-layer h-BN on different substrates.

Response #3.2: We now discuss, as you requested, how these enhancements are quantitatively related to the dipole and quadrupole moments of the second order susceptibility and how the different order susceptibilities are related to each other. In h-BN the polarization arises from two components: $P_\alpha^{2\omega} = P_\alpha^{2\omega,d} + P_\alpha^{2\omega,q}$, where $P_\alpha^{2\omega,d}$ is the dipole moment and $P_\alpha^{2\omega,q}$ is the quadrupole moment. These are expressed as

$$P_\alpha^{2\omega,d} = \varepsilon_0 \chi_{\alpha\beta\gamma}^{(2)} E_\beta^\omega E_\gamma^\omega$$

$$P_\alpha^{2\omega,q} = \varepsilon_0 \chi_{\alpha\beta\gamma}^{(4)} E_\beta^\omega \frac{\partial E_\gamma^\omega}{\partial z}$$

where the subscripts α, β, γ denote in-plane polarization directions in Cartesian coordinates, and E^ω is the pump field.

In monolayers of h-BN, governed by D_{3h} point group symmetry, the monolayer quadrupole moments $P_\alpha^{2\omega,q}$ vanish due to the reflection symmetry, rendering $\chi_{\alpha\beta\gamma}^{(4),1L} = 0$.

However, in bilayer configurations, where reflection symmetry is broken, a nonzero quadrupole moment emerges. This arises from the gradient in the electric field across adjacent layers. Consider the nonlinear polarization from a **bilayer unit** including layer j and $j + 1$ of the whole stack. In this unit the dipole moment vanishes due to centrosymmetry, leaving **the quadrupole moment as the leading term**:

$$P_\alpha^{2\omega,q,2L} = P_\alpha^{2\omega,d,j} + P_\alpha^{2\omega,d,j+1} = \varepsilon_0 \chi_{\alpha\beta\gamma}^{(2)} E_\beta^{\omega,j+1} E_\gamma^{\omega,j+1} - \varepsilon_0 \chi_{\alpha\beta\gamma}^{(2)} E_\beta^{\omega,j} E_\gamma^{\omega,j}$$

$$= \varepsilon_0 \chi_{\alpha\beta\gamma}^{(2)} \left(E_\beta^{\omega,j} + \Delta E_\beta^\omega \right) \left(E_\gamma^{\omega,j} + \Delta E_\gamma^\omega \right) - \varepsilon_0 \chi_{\alpha\beta\gamma}^{(2)} E_\beta^{\omega,j} E_\gamma^{\omega,j}$$

$$\approx 2\varepsilon_0 \chi_{\alpha\beta\gamma}^{(2)} d_{BN} E_\beta^{\omega,j} \frac{\partial E_\gamma^\omega}{\partial z}$$

where d_{BN} is the distance between the adjacent two layers. We can see thus that the effective quadrupole susceptibility can be given in terms of the dipolar susceptibility as $\chi_{\alpha\beta\gamma}^{(2),2L,q} = 2\chi_{\alpha\beta\gamma}^{(2),d} d_{BN} \neq 0$.

Now let us consider separately the effects on h-BN stacks containing either even number of monolayers or odd number of monolayers:

- (1) For even parity h-BN: Even-layer material, which owns the centrosymmetry of bulk h-BN material, can still show a weak SHG signal. This is all attributed to the nonzero quadrupolar

moment since the dipole moment vanishes due to centrosymmetry. In our method, we have **enhanced the quadrupolar moment by increasing the gradient of the electric field $\frac{\partial E_Y^\omega}{\partial z}$ as well as by amplifying $E_\beta^{\omega,j}$ within the material (both contribution (i) and (ii))**. This enhancement is demonstrated in new Figure 2a of the main text.

- (2) For odd parity h-BN: Both the dipole and quadrupolar moments contribute to the SHG signal. This system includes a combination of two adjacent layers, akin to the even-layer material, and an additional single layer. In the two-layer configuration, despite the absence of a dipole moment (due to symmetry cancellation), the quadrupole moment remains nonzero (as discussed in (1)). Conversely, in the single-layer configuration, the dipole moment is present, but the quadrupole moment is zero. Thus, the overall nonlinear polarization $P_\alpha^{2\omega}$ can be expressed as $P_\alpha^{2\omega,d,SL} + P_\alpha^{2\omega,q,2L}$, where SL denotes one single layer and 2L denotes two adjacent layers. By placing h-BN on a gold substrate, **enhancements are observed in all of E_β^ω , E_Y^ω and $\frac{\partial E_Y^\omega}{\partial z}$ (both contribution (i) and (ii))**, effectively boosting both dipole and quadrupole contributions. This enhancement is depicted in Figure R11, where a and b illustrate the effects on the SiO₂/Si substrate, and c and d demonstrate the enhancements on the gold substrate. Interestingly, on gold substrates the quadrupolar response in h-BN is only 2 to 3 times smaller than the dipolar response. The close absolute values of the two contributions underscore the significant impact of the quadrupolar moment in altering the nonlinear response of a material, especially in centrosymmetric materials where the quadrupolar contribution is dominant.

Figure R11: (a), (c) The magnitude of SHG electric field from dipole and quadrupole contributions, as a function of the odd total layer number of h-BN on SiO₂/Si (a) and gold (c) substrate. Results for even layers not shown, since the dipole moment vanishes. (b), (d) The phase of SHG electric field from dipole and quadrupole contributions, as a function of the odd total layer number of h-BN on SiO₂/Si (b) and gold (d) substrate.

Besides, to confirm that in our architecture the quadrupolar contribution to the polarization can be comparable in intensity to the dipolar contribution, we have added a second panel in the new Figure 1 where we plot the SHG computed by nonlinear transfer simulations separating each of the contributions. As shown in Figure R12 below, the absolute value of the quadrupolar contribution to the SHG is of the same order of magnitude as the dipolar contribution for an h-BN containing odd number of monolayers and being deposited on gold. Furthermore, for even number of monolayers h-BN, which display SHG signals on gold similar to odd counterparts, the quadrupolar contribution is the first nonzero contribution, highlighting its qualitative and quantitative importance.

Figure R12: The simulation results of dipolar (green) and quadrupolar (orange) contributions to the magnitude of SHG electric field of h-BN flakes with odd number of monolayers.

To highlight these two aspects and explain in detail the physics underlying the enhancements observed in our experimental results, we have added a completely new section in the main manuscript before presenting the experimental results. Besides, as already indicated, we have modified Figure 1 to illustrate each of these physical mechanisms.

The new section in the main manuscript is entitled ***“General strategy for enhancing SHG in centrosymmetric and noncentrosymmetric thin materials”***, which we reproduce below (the added text appears in *italic*):

“General strategy for enhancing SHG in centrosymmetric and noncentrosymmetric thin materials

The nonlinear optical processes in a material are governed by its electrical polarization, which is given by a power series of the electric field amplitude. Beyond the linear term [26]:

$P_{NLO} = P_{2\omega} + P_{3\omega} + \dots = \chi^{(2)}E_\omega E_\omega + \chi^{(3)}E_\omega E_\omega E_\omega + \dots$, where $\chi^{(2)}$ is the second-order susceptibility tensor. $\chi^{(2)}$ is at the basis of SHG, which not only holds potential for creating nonlinear devices but also is being employed extensively by the 2D community for non-destructive lattice orientation identification using low pump powers [4, 27, 28]. In the classical SHG process, two photons of frequency ω interact inside the nonlinear material to give a single photon of frequency 2ω . As indicated above, the SHG intensity is related to the pump electric field by the second-order susceptibility tensor that, in turn, can be expanded in terms of its dipole and higher multipole moments. To leading order, we can thus write $P_{2\omega} = \chi_d^{(2)}:E_\omega E_\omega + \chi_q^{(2)}:E_\omega \nabla E_\omega + \dots$, where $\chi_d^{(2)}$ and $\chi_q^{(2)}$ represent the dipolar and quadrupolar moments of $\chi^{(2)}$. Note that the dipolar term couples only to the electric field *amplitude (in fact to the electric field intensity)*, while the quadrupolar term couples to the electric field amplitude and its gradient (i.e. the spatial variation of the electric field). *For a more detailed analysis of this expansion and the quantitative relationship between the dipole and quadrupole moments see note “Dipole moment and Quadruple moment” in the supplementary information.*

Our strategy consists thus in finding simple designs that maximize the tradeoff between the electric field amplitude at the nonlinear-material location and the gradient of the electric field at that same position, as illustrated in Figure 1a. To do so we exploit two well-known facts of metals: first, they display a large reflectivity over a broad wavelength range in the visible and infrared (IR), comparable to Bragg mirrors with few number of pairs or with small refractive index contrast between the Bragg materials (i.e. those forming low Q Fabry-Perot cavities) [37]; second, by imposing a near-zero electric field at their surface, they enable to create a strong electric field gradient near the metal, far exceeding the field gradient in thick h-BN or NbOCl₂ induced by below-bandgap absorption or that created at dielectric interfaces.

To illustrate quantitatively the enhancement of both magnitudes, in Figure 1b (left axis) we have plotted the electric field intensity inside 100 nm-thick h-BN deposited on gold and on standard SiO₂/Si wafers. Due to the relatively large reflectivity of the h-BN/gold interface compared to the h-BN/SiO₂ interface (see Table S5 in SI), the electric field magnitude (at the pump wavelength) increases by a factor 3-5, providing a stronger dipolar contribution to the nonlinear polarization. Interestingly, the electric field gradient does also become larger when considering h-BN deposited on gold (right axis in Figure 1b), with an increase that can amount up to a factor ten compared to h-BN/SiO₂. This increased gradient, combined with the enhanced electric field amplitude, magnify the quadrupolar contribution to the nonlinear polarization.

Still, one can wonder how important the quadrupolar contribution can be, in absolute terms, compared to the dipolar one. In centrosymmetric materials, for which the dipolar contribution vanishes due to symmetry constraints, the quadrupolar contribution is obviously the dominant one, and it enables to observe SHG in sufficiently thick materials across which a field amplitude gradient can be established. On the other hand, in non-centrosymmetric materials the dipolar contribution is in general considered to be the dominant one. To illustrate the actual magnitude of the quadrupolar contribution enabled by the use of metals, we used the nonlinear transfer matrix approach [23] to calculate the individual contributions to the SHG intensity of h-BN displaying an odd number of monolayers (i.e. for a non-centrosymmetric BN) deposited on gold. Figure 1c indicates that when using gold as a substrate, the quadrupolar response in h-BN is just a factor 2 to 3 times smaller than the dipolar response. The fact that both contributions are of the same order of magnitude highlights the potential effect of the quadrupolar moment in modifying the nonlinear response of a given material, particularly of centrosymmetric materials, where the quadrupolar contribution dominates. Furthermore, because global enhancement of dipolar and quadrupolar contributions stems from two general metal properties, its effect is observed when depositing the nonlinear active material on a variety of metals (Figure S1).

Figure 1 | Electric field distribution in centrosymmetric-nonlinear material/metal heterostructures. **a**, Schematic of incident light distribution on gold film (left) and SiO₂/Si substrate (right), showing a magnified electric field amplitude and gradient on the gold film. *Inset: Illustration of the SHG process, where two photons are combined to generate a single signal photon.* **b**, The magnitude of pump field intensity distribution (red) and the gradient of electric field (blue) in a 300-monolayer h-BN flake on gold (line) and SiO₂/Si (dashed line). The dark area is the substrate. **c**, The simulation results of dipolar (green) and quadrupolar (orange) contributions to the magnitude of SHG electric field

of h-BN flakes with odd number of monolayers.

”

Besides the explanation of the physics provided in the new section of the main manuscript, we have also added an additional section in the Supplementary Information entitled “*Dipole moment and Quadruple moment*” recalling the discussion on the dipolar/quadrupolar contributions for odd and even-parity and showing the quantitative relationship between the quadrupolar and dipolar susceptibilities.

Comment #4: What is special about using the hBN and NbOCl₂ materials? If the source of the nonlinearity is $\chi^{(2)}_d$ and $\chi^{(2)}_q$, which are bulk quantities, are these numbers particularly larger in hBN and NbOCl₂ compared to say materials like lithium niobate or GaAs? Numbers should be given and discussed.

Response #4: We thank you for the valuable suggestion, which will enable readers to understand our choice of materials and compare their intrinsic nonlinear properties with those of other materials.

Our choice of h-BN and NbOCl₂ is intentional: in centrosymmetric materials like h-BN with even layers, the dipolar contribution vanishes due to symmetry constraints, making the quadrupolar contribution dominant and highlighting the potentiality of any method enhancing it. Conversely, in non-centrosymmetric materials like NbOCl₂, the dipolar contribution is typically dominant, which our method can also enhance (as microcavities or metasurfaces can also do). Besides, our method provides broadband enhancement, which is particularly advantageous for ultra-thin entangled photon sources like NbOCl₂, as one will be able to exploit frequency-entangled photon pairs and increase the number of quantum communications channels for one single source. This technologically important aspect is now highlighted in the conclusions, where we have added a new paragraph discussing/comparing applications where either narrow band or broadband enhancement is necessary, as suggested by referee 1.

To address your concerns and to have a more transparent comparison with the existing literature, we benchmarked our nonlinear materials by comparing the second order susceptibility (Table R5), which appears now as Table S2 in the Supplementary Information:

Nonlinear material	$\chi^{(2)}$ (pm/V)	Reference
hBN	42	This work
NbOCl₂	400	This work
GaAs	119	[14]
BBO	2	[15]

MoS₂	300	[16]
WS₂	300	[16]
GaSe	30	[17]
PdSe₂	51.7	[18]
LiNbO₃	40	[19]

Table R5: Comparison of entangled photon pair sources presented in previous studies.

Explanations for the choice of the centrosymmetric and non-centrosymmetric materials have been added in Page 5, Page 7 and Page 20 of the main text (in *italic* below):

Page 5: “*To illustrate the general applicability of our heterostructures we chose two Van der Waals materials: hBN, which is the most widespread 2D material today and whose centro-symmetric or non-centrosymmetric character is monolayer number dependent, and NbOCl₂, which is a van der Waals material exhibiting one of the largest nonlinear responses to date (Table S2).*”

Page 7: “*Still, one can wonder how important the quadrupolar contribution can be, in absolute terms, compared to the dipolar one. In centrosymmetric materials, for which the dipolar contribution vanishes due to symmetry constraints, the quadrupolar contribution is obviously the dominant one, and it enables to observe SHG in sufficiently thick materials across which a field amplitude gradient can be established. On the other hand, in non-centrosymmetric materials the dipolar contribution is in general considered to be the dominant one. To illustrate the actual magnitude of the quadrupolar contribution enabled by the use of metals, we used the nonlinear transfer matrix approach [23] to calculate the individual contributions to the SHG intensity of h-BN displaying an odd number of monolayers (i.e. for a non-centrosymmetric BN) deposited on gold.*”

Page 20: “*By engineering the light component, we achieved a substantial boost in the light-matter interaction, resulting in a notable increase in the nonlinear response, as observed for both SHG and SPDC. For centrosymmetric materials like h-BN, which typically exhibit lower second-order susceptibility values due to their symmetry, our approach has enabled an impressive increase of about two orders of magnitude in the SHG response and enabled even-layer h-BN to exhibit SHG responses comparable to those of odd-layer h-BN. This demonstrates that the symmetry constraints on the nonlinear optical response can be effectively mitigated by adequately modifying the electric field distribution within the nonlinear material.*”

Comment #5: In line 148-150, the authors say that “The polarization-dependent SHG measurement exhibits a distinct six-fold symmetry (Figure 1c), in agreement with the D3h point group of h-BN and some TMDs [26, 35-38].” Since the authors seem to be claiming that there is a dominant contribution from $\chi_{2,q}$, the question is, does the quadrupolar $\chi_{(2)}$ tensor have the same tensor symmetry as the dipolar one? The authors should cite a work that has investigated

the tensor symmetry of the quadrupolar $\chi_{\alpha\beta\gamma}^{(2)}$ either for hBN or generally showing that it has the same tensor symmetry of the dipolar $\chi_{\alpha\beta}^{(2)}$.

Response #5: Thank you for questioning about the symmetry of the quadrupolar tensor. Your question allows us to clarify the relationship between the dipolar and quadrupolar nonlinearities, particularly in the context of h-BN and similar materials. As in Response #3, the quadrupolar contribution $\chi_{\alpha\beta\gamma}^{(2),2L,q}$ is derived considering the interaction between two adjacent layers as follows:

$$\begin{aligned} P_{\alpha}^{2\omega,q,2L} &= P_{\alpha}^{2\omega,d,j} + P_{\alpha}^{2\omega,d,j+1} = \varepsilon_0 \chi_{\alpha\beta\gamma}^{(2)} E_{\beta}^{\omega,j} E_{\gamma}^{\omega,j} - \varepsilon_0 \chi_{\alpha\beta\gamma}^{(2)} E_{\beta}^{\omega,j+1} E_{\gamma}^{\omega,j+1} \\ &= \varepsilon_0 \chi_{\alpha\beta\gamma}^{(2)} E_{\beta}^{\omega,j} E_{\gamma}^{\omega,j} - \varepsilon_0 \chi_{\alpha\beta\gamma}^{(2)} (E_{\beta}^{\omega,j} + \Delta E_{\beta}^{\omega})(E_{\gamma}^{\omega,j} + \Delta E_{\gamma}^{\omega}) \\ &\approx 2\varepsilon_0 \chi_{\alpha\beta\gamma}^{(2)} d_{BN} E_{\beta}^{\omega,j} \frac{\partial E_{\gamma}^{\omega}}{\partial z} \end{aligned}$$

where d_{BN} is the distance between the adjacent two layers. The effective quadrupole susceptibility is therefore obtained from $\chi_{\alpha\beta\gamma}^{(2),d}$ by a factor d_{BN} , indicating that the tensor symmetries of the quadrupolar and dipolar terms are identical under these conditions. This indicates that the tensor symmetry of $\chi_{\alpha\beta\gamma}^{(2),2L,q}$ is exactly the same as $\chi_{\alpha\beta\gamma}^{(2),d}$. Consequently, $\chi_{yyy}^{(2),2L,q} = -\chi_{yxx}^{(2),2L,q} = -\chi_{xxy}^{(2),2L,q} = -\chi_{xyx}^{(2),2L,q}$. This parallelism in tensor symmetry between dipolar and quadrupolar terms was also remarked in reference Yao *et al.*, Sci. Adv. 2021; 7 : eabe8691, which results are consistent with ours.

To address your concerns, we have added this discussion into the supplementary section “**Dipole moment and Quadrupole moment**”.

Comment #6: Another important point that should be discussed: Can one rule out if the SHG/SPDC enhancement is coming from the metal surface nonlinearity? What is the pump beam diameter? The authors give a number in the supplementary, that say 2 by 2 meters, which is obviously a typo and should be corrected. I would guess they meant 2 by 2 micrometer is the pump beam size. In such a focusing regime, one would have some non-negligible longitudinal polarization component for the field, which might generate longitudinal and transversal field components through the surface nonlinearity of the gold, that might then be collected in reflection. The experimental test would be to do the same work without the hBN or NbOCl₂, and just with the metal-based structures.

Response #6: Thank you for highlighting the crucial point regarding the potential contribution of metal surface nonlinearity to the observed SHG/SPDC enhancements. Also, we appreciate you pointing out the typo in our supplementary material regarding the pump beam size. Indeed, the

correct measurement should read 2 by 2 micrometers, not meters. We have corrected this in the documents to avoid any confusion.

To address your concern about the contribution of the metal (gold) surface to the nonlinear optical processes observed, we conducted SHG experiments on bare gold. The SHG from the surface of metals is almost always dominated by the nonlinear polarizability of the free and bound metal electrons at the interface. Our results indicate that the SHG signal from the bare Au film is significantly smaller - 20 and 80 times smaller- than the SHG signal we observed from the h-BN on gold film with h-BN thickness of 36 nm and 65 nm (Figures R13). Given this large discrepancy, we believe the impact of the Au film's SHG contribution on the overall enhancement factor is minimal and falls within the error bars.

Nevertheless, we agree that the nonlinear response from metal should be excluded out. Thus, to reassure the readers we mention it explicitly in Page 10 of the new main text, and the SHG measurements on the bare Au films have been added into the supplementary information:

“...To discard any potential contribution to the SHG signal from the metal surface nonlinearity, SHG was recorded under the same conditions from bare gold. The SHG response from the bare metal is about 20 and about 80 times smaller than the SHG signal from h-BN/gold heterostructures with h-BN thicknesses of 36 nm and 65 nm, respectively, as shown by polarization-resolved measurements on these two h-BN flakes (compare Figures S4, S5 and S6). These findings corroborate that the measured SHG signal indeed originates mostly from the h-BN flakes...”

Figure R13: SHG from four locations on bare gold film, 36 nm h-BN on Au film and 65 nm h-BN on Au film.

Comment #7: A major concern, which also relates to the previous two points, is that in fact the measured polarization measurement results in the presence of the metal, shown in Fig 1(c), and also in Fig S6 of the supplementary, do not match the expected 6-fold symmetry of SHG from hBN. The authors explanation of “This phenomenon might be tentatively ascribed to the inhomogeneous strain induced by the irregularities of the gold film (see Figure S6 for more examples of this asymmetry).” is not convincing and is also not supported by any evidence. How irregular is the surface of the gold? What type of strain can it cause? And how is strain expected

to affect the nonlinear tensor in 2D materials? A much more probable cause could be that the SHG is coming from different contributions of materials with different $\chi^{(2)}$ tensors. One strong candidate is the surface nonlinearity of the gold. Yet these are not really investigated. I think the authors should justify their strain argument with some more substantial reasoning and citing some similar studies on that topic, and at the same time do some reasoning on why it cannot be other contributions like gold's surface nonlinearity.

Response #7: Thank you for your insightful observations and for highlighting the observed asymmetry in the SHG signals from hBN on gold substrates, which is a systematic observation. We appreciate the opportunity to further clarify and substantiate our findings.

In response to your concerns, we conducted additional SHG measurements on different areas of the same samples and on different areas of a bare gold substrate. As shown in Figure R13, the SHG response from the Au film alone was 20 to 80 times smaller than on h-BN/gold (compare the scales in Figures R13). Indeed, under exactly the same measurement conditions (pumping power, beam size and integration time), the variation in SHG intensity between different polarization states on the bare gold surface amounts to, at most, about 60 a.u., while the h-BN asymmetries exceeded 900 units (Figure R14, R15). This significant difference suggests that the SHG signals are predominantly influenced by the h-BN layer rather than by the SHG of the underlying gold substrate.

To address potential surface irregularities, we have added AFM images of both the gold and SiO₂ films to the supplementary materials (Figure R16). These images provide important insights into the flatness and surface quality of the films, further clarifying their role in the observed nonlinear optical enhancements. The root mean square roughness value of gold and SiO₂ films is 2.4 nm and 0.6 nm, respectively (0.5nm if we exclude the dust particle, which we have left in the chosen image to pinpoint the presence of isolated particles that one needs to be careful with, and exclude, when performing the measurements). Based on these findings, we attribute the variations to strain effects induced during sample preparation on the different substrates.

Figure R14: SHG measurement from 36 nm h-BN on Au film.

Figure R15: SHG measurement from 65 nm h-BN on Au film.

Figure R16: (a) AFM image of gold film. (b) Height profile of the gold film along the red line indicated in (a). (c) AFM image of SiO₂ film. (d), (e) Height profile of the SiO₂ film along the red(d) and green(e) line indicated in (c). Root mean square roughness is defined as $\sqrt{\langle (h - \bar{h})^2 \rangle}$, where h is the height in each pixel.

To understand deeper the role of strain, we analyzed theoretically the strain dependence of the SHG signal in h-BN. The SHG intensity as a function of strain can be expressed as:

$$I_{\parallel}^{(2)}(2\omega) \propto \frac{1}{4} (A \cos(3\varphi) + B \cos(2\theta + \varphi))^2$$

With $A = (1 - \nu)(p_1 + p_2)(\varepsilon_{xx} + \varepsilon_{yy}) + 2\chi_0$ and $B = (1 + \nu)(p_1 - p_2)(\varepsilon_{xx} - \varepsilon_{yy})$. p_1 and p_2 are the photoelastic parameters, ε_{xx} and ε_{yy} denote the principal strains, θ is the principal strain orientation, φ the polarization angle, and χ_0 the nonlinear susceptibility parameter of the unstrained crystal lattice. Although data on the photoelastic parameters for h-BN are scarce, analogous studies on monolayer MoS₂ have shown that tensile strain can reduce the SHG response in certain directions by up to half compared to unstrained conditions. We thus cite it to draw parallels and support our hypothesis regarding strain effects in hBN. This aspect of our research opens an interesting direction for future studies, particularly in accurately quantifying the

photoelastic parameters of h-BN. Based on this analysis we propose that the asymmetry in SHG results from our samples is likely due to local strain effects incurred during sample preparation, rather than contributions from gold's surface nonlinearity.

Figure R17: Strain effects on SHG measurement from monolayer MoS₂. Reproduced from Nature communications 9.1 (2018): 516.

In order to address these concerns, we have added the following information in Page 10 of the main text (shown in *italic* below) as well as the AFM images of gold and SiO₂ surfaces employed as substrates in our study, which are shown now in the Supplementary Information:

“Noticeably, the polar plot of the SHG is fully symmetric on the flat SiO₂/Si substrate while an asymmetry is observed systematically on gold films. *This asymmetry degree varies, however, from location to location within the same flake (Figures S4 and S5). This observation is tentatively ascribed to the inhomogeneous strain induced by the irregularities of the gold film whose roughness, while in the nanometer scale, is five times larger than that of SiO₂/Si substrates (root mean square roughness of 2.4 nm Vs 0.5 nm, respectively). Indeed, to minimize the effect of the polarization-resolved SHG response asymmetry of h-BN on gold, each experimental data in Figure 2a corresponds to the SHG value averaged over the six maxima in the polarization-resolved measurements, both for data on gold and on SiO₂/Si substrates.*”

Comment #8: In plots like 1(b) or 1(e), why is the signal generated for the case of hBN on SiO₂ around 0? I assume the authors subtracted the average value for this measurement. But for an analysis where one is interested in the ratio of two signals, one should not do a subtraction from both signals, otherwise how can the ratio be calculated? I think the authors should show the actual measured values and not the average-subtracted values, which is not useful for the analysis of enhancement.

Response #8: Thank you for your detailed observation regarding the data processing used in our study, particularly in plots 1(b) and 1(e) of the original manuscript (note that these data are now incorporated into the new Figure 2). Your question highlights an important aspect of our data presentation that merits clarification.

In the following we explain in detail how we calculate the enhancement factor for the SHG response from h-BN on various substrates, which is an essential part of our analysis.

First, we measure the SHG response of h-BN on gold and SiO₂ films, assessing the response for different pump polarizations to provide polarization-dependent SHG results. Rather than subtracting the background, we apply a Lorentz fitting directly to the raw data, which helps us to accurately and systematically extract the maximum SHG signal values at different polarizations. Figure R18 (which corresponds to a new panel in the new Figure 2 of the main text) presents the polarization-dependent SHG measurements of 18 nm h-BN and of 36 nm h-BN thin films on gold and on SiO₂, respectively.

One can see that the SHG signal from the 36 nm h-BN on SiO₂ substrate, which displays the smallest SHG signal of all the measured samples in our study, is completely isotropic. This indicates that its intensity is within the background of our current measurements, but still nonzero. On the other hand, for all the other samples, both on gold and on SiO₂, we do observe the six polarization-dependent lobes.

Figure R18: SHG response from 18 nm and 36 nm h-BN on gold and on SiO₂ films.

Furthermore, to minimize the effect of the asymmetric polarization-resolved SHG, especially on gold, each data point in the new Figure 2a represents the SHG value averaged over the six maxima in the polarization-resolved measurements. This averaging process has been conducted for samples

on both gold and SiO₂/Si substrates. To clarify this point we have added the description of our methodology in Page 10 of the main text and Page 23 of the Methods section, and added into the new Figure 2 the raw data of polarization resolved SHG measurements on 18 nm and 36 nm thick h-BN on gold and on SiO₂ substrates.

Page 10: *“Indeed, to minimize the effect of the asymmetry on the polarization-resolved SHG response of h-BN on gold, each experimental data in Figure 2a corresponds to the SHG value averaged over the six maxima in the polarization-resolved measurements, both for data on gold and on SiO₂/Si substrates.*

Overall, the results in Figure 2 confirm that the SHG intensity of h-BN on gold originates from the h-BN and that it is enhanced by one to two orders of magnitude thanks to the interplay between enhanced dipolar and enhanced quadrupolar contributions.”

Page 23: *“To ensure a more precise and reliable calculation of the SHG enhancement factor, we have utilized Lorentz fitting directly on the raw data to extract the maximum SHG signal values, ensuring that no artificial baseline adjustments impact the analysis. This method allows us to accurately capture the polarization-dependent SHG response for hBN on different substrates, including gold and SiO₂. For each substrate, we identify six maximum values that correspond to the six-fold symmetry characteristic of hBN. By performing the same measurements on hBN on and off the gold substrates, we calculate the enhancement factor at each angle. We then compute the mean value and standard deviation to provide a robust estimate of the enhancement factor and its associated uncertainty. This approach ensures that the enhancement factor is derived from the true physical properties of the system, free from the influence of any background subtraction, and accurately reflects the influence of different substrates on the SHG response.”*

Comment #9: Why is there Ti layer under gold? Is that needed for bonding? Would be good if this is explained too.

Response #9: We appreciate your attention to this detail and thank you for the opportunity to clarify the role of the titanium layer. The titanium layer serves a critical function as an adhesion promoter between the gold layer and the underlying substrate. Gold does not adhere well to many substrates, including common materials like glass or silicon. Ti, however, forms a strong chemical bond with both the substrate and the gold layer, effectively anchoring the gold in place.

In the revised manuscript we have included in Page 8 of the main text and in Page 22 of the Methods section details on the role of the Ti layer:

Page 8: *“Note that Ti is used here to facilitate metal adhesion and plays no particular role in terms of the optical design. A femtosecond...”*

Page 22: “*The primary role of the Ti layer is to act as an adhesion promoter, ensuring a strong bond between the gold film and the underlying substrate.*”

Comment #10: In the experiments, how do the authors determine/measure if a hBN flake has odd or even number of layers? Is it just from the SHG measurements or is it some direct measurement approach like AFM? This should be mentioned.

Response #10: We thank you for such an insightful comment. We have not employed the results of AFM in this study to determine the parity of layers. Instead, we just measure the thickness and SHG response of sample and plot them in the figure. Interestingly, they can either be odd or even layers. This demonstrates the capabilities of SHG as a diagnostic tool for assessing layer parity in 2D materials.

The only section where we intentionally add one additional monolayer to a “thick” h-BN is in Section “Innovative parity analysis of thick h-BN flakes using second harmonic generation” of the Supplementary Information, where we wanted to illustrate that, thanks to our approach, we can easily distinguish using SHG between N-monolayers and N+1 monolayers even when N is large, which is not straightforward when using a standard glass substrate.

Comment #11: In Fig 2(e), a combination of bad image quality, bad choice of colors, and lack of legends, makes this plot completely non-understandable and consequently any physics that is tried to imply from it is lost.

Response #11: We thank you for the comment, and we sincerely apologize for the confusion. We have updated Figure 2 (now Figure 3) to provide clearer data and to illustrate the sample fabrication process and SHG measurements performed on the twisted sample deposited on gold film, demonstrating our method's compatibility with other enhancement techniques. We also revised the section “Compatibility of our approach with twisting strategies” to reflect these updates.

Figure 3a now illustrates the fabrication process for generating twisted samples on gold film in a controlled manner, and Figure 3d has been redrawn with distinct colors to enhance clarity. It now shows a bright-field optical microscopy image of h-BN homostructures, with SHG intensity scanned along the red-yellow-green-blue path, encompassing all h-BN homostructure (AA' and AB) and substrate (gold and SiO₂/Si) combinations. The corresponding SHG results along this closed loop are presented on the right side of Figure 3d. By combining twist methods with our metal enhancement, we achieve a 357-fold increase in SHG signal compared to AA' on SiO₂/Si. This enhancement depends on both the location of the non-centrosymmetric interface and the parity of the top and bottom flakes, as stated explicitly in the main text.

Figure R19: SHG enhancement in twisted h-BN on gold film. *a*, Detailed process for preparing the h-BN homostructure. It illustrates the step-by-step method used to assemble the h-BN homostructure. *b*, Schematic of h-BN homostructures (blue) on gold film (yellow). A homostructure is formed by rotating the top h-BN flake relative to the bottom one by an angle. Inset: Rotation of h-BN lattice with a θ angle. *c*, The experimental results of SH intensity as a function of polarization angle for the homostructures on gold and SiO₂/Si; AA' stack corresponds to 0° and AB stack corresponds to 60° stacking angle. *d*, Bright field optical microscopy image of h-BN homostructures and SH intensity along a closed loop including all the h-BN homostructures (AA' and AB)/substrates(gold and SiO₂/Si) combinations. Different combinations are represented by unique colors and symbols, each corresponding to the path depicted by a line of the same color in the image. *e*, Summary of the enhancement factors from different areas in *d* with respect to the 0° homostructure on SiO₂/Si substrates, measured at a 72° polarization angle: AB (60°) homostructure on gold; AA' (0°) homostructure on gold; AB (60°) homostructure on SiO₂/Si; AA' (0°) homostructure on SiO₂/Si. The error bars represent the standard deviation of the enhancement factor within each region.

In the revised manuscript we have included in the main text a detailed description for preparing the h-BN heterostructure, which is also illustrated in the new Figure 3a:

Page 12: “To do so we investigated the SHG of h-BN homostructures with two different stacking

angles on gold and on SiO₂/Si substrates (Figures 3a and 3b). Specifically, we selected a 49 nm thick h-BN flake and separated it into two pieces. One piece of 49 nm h-BN layer was picked up while the other piece of 49 nm h-BN layer, left on the initial substrate, was rotated by 180° (equivalently, 60°) before stacking it side by side with the unrotated one. This pair of rotated h-BN layers were subsequently and simultaneously transferred on top of a third 46 nm h-BN layer, which had a rotation of 0° with respect to the two top initial layers (Figure 3a). In this way, h-BN homostructures with 0°- and 60°-rotation interfaces can be compared within the same sample and under exactly the same experimental conditions, as evidenced by the polarization-dependent SHG (Figure 3c).

Comment #12: Another major issue is the following: The enhancement factor, as shown in Fig 2(a) is rather a meaningless figure of merit. If I understand correctly, what is measured is how much SHG is generated for a certain thickness of a material, comparing the same thickness when placed on SiO₂ and gold. For such a system, where the authors are claiming that everything is dependent on the intensity of the field at the surface, it could easily be that for a specific thickness, the gold substrate has a dramatically different field at the surface compared to SiO₂ substrate, due to simple Fabry Perot effects. For example, I can imagine if the thickness of the SiO₂ layer is systematically changed, this enhancement factor could easily change. So that does not mean that the gold substrate is creating a dramatic enhancement in the field when considering all thicknesses. This can in fact even be seen in the simulations shown in Fig. 2(b), where the overall enhancement of generated SH, when considering all thicknesses and averaging the results in a way, seems to be about an order of magnitude. This order of magnitude enhancement more or less makes sense, since reflection from the interface makes the electric field generally 2 times stronger at some points, and this makes the intensity of fundamental harmonic pump 4 times stronger, which makes SHG intensity about 16 times stronger. Then one can consider that reflection is collecting also waves generated in the transmission direction and sends them to the reflection side, creating a factor of 32. So such enhancement factors are easily explained by the effect of the mirror, which is not a really an interesting effect. I suggest the authors perform such a study, where the thickness of hBN is fixed but the SiO₂ thickness is changed, to find a fair ratio between the efficiency of the system with and without gold. Overall, I think that this enhancement factor at equal thicknesses is not a reasonable figure of merit to judge the enhancement, and a more general figure of merit should be defined that can somehow take out this very sensitive thickness-dependent interferences to be able to judge how much the gold substrate is affecting the system.

Response #12: Thank you for your insightful feedback regarding the enhancement factor in Fig. 2(a). We fully agree that Fabry-Perot effects and interferences on the SiO₂/Si samples could influence the surface electric field for a specific material thickness, potentially impacting the observed enhancement factor. Before going into the discussion let us clarify that the substrates we used in the manuscript are SiO₂/Si with a 285 nm thick SiO₂ layer, rather than pure SiO₂. We have updated the manuscript to reflect this more clearly, even if it was already explicitly stated.

First of all, to prevent any confusion with the enhancement factor and because it seemed to us, reading all three referees' comments, that it would be clearer for the reader, we now plot the experimental data in the new figure 2a as the SH intensity, so that we show explicitly and in the same scale the SH signals of h-BN on gold and of h-BN on SiO₂/Si. Furthermore, in order to show in the same plot the numerical simulation curves, we normalized all experimental points by the experimental SH signal of h-BN on gold displaying about 70 nm thickness, for which the theoretical SH curves for odd and even number of monolayers cross each other. The new Figure 2a displaying all SH data of h-BN (on gold and on SiO₂/Si) is:

Figure R20: Experimental and simulated SH intensity for h-BN with different thicknesses on gold films and SiO₂/Si substrates. Symbols are experimental points while solid and dashed lines are nonlinear transfer matrix simulation results. The pump field in the simulation is 890 nm. All data are normalized to the value for 70 nm h-BN on gold.

Beyond this modification of the way we present the data, and to answer clearly your question, we performed additional simulations with varying SiO₂ thicknesses for nonlinear material/SiO₂/X structures, where X represents a Si substrate (i.e. a bulk wafer), various metals (Au, Pt, Ag, Pb), or a fused silica (SiO₂) wafer. We analyzed four different h-BN thicknesses in these simulations. As shown in Figure R20, the SiO₂ thickness indeed affects the SHG signal due to interference within the SiO₂ “dielectric cavity”, but the nonlinear material/SiO₂/metal structures outperform systematically the SiO₂/Si and pure silica substrates. Even with the optimal SiO₂ thickness maximizing the signal from the SiO₂/Si sample, enhancement thanks to the metal surfaces is achieved. These new calculations (Figure R21) have been added to the Supplementary Information.

Figure R21: Simulated SHG responses from material/SiO₂/X structures and fused silica at various hBN thicknesses.

One can notice that we have separated in the previous figure the effects of the SiO₂/metal structures depending on the monolayers parity, as we wanted to emphasize the role of the electric field gradient, i.e. the quadrupolar contribution, to the nonlinear response of centro-symmetric materials. The maximum SHG intensities for each optimal scheme are summarized in Table R6, which shows that material/SiO₂/metal structures consistently provide better SHG performance than SiO₂/Si and pure silica.

	Odd Layer			Even Layer		
Thickness of hBN	hBN/SiO₂/Au	hBN /SiO₂/Si	hBN /Silica	hBN /SiO₂/Au	hBN /SiO₂/Si	hBN /Silica
10 nm	227.7	26.5	4.0	4.5	0.6	0.2
30 nm	91.3	10.0	1.4	24.0	4.0	1.0
60 nm	6.7	0.5	0.2	32.7	6.6	1.1
100 nm	15.6	2.6	1.3	33.2	3.0	0.2

Table R6: The maximum value of SHG intensity obtained for the optimum layers thicknesses.

Regarding your back-of-envelope calculation and your hypothesis about the sole effect of the enhanced reflectivity, we agree that the enhancement of field magnitude is due to larger reflectivity and Fabry-Perot effects (whenever a dielectric spacer is used), but we believe it does not fully capture the underlying physics. In particular, if the enhancement was solely due to reflection, the SHG signal from even-layer h-BN (which is centrosymmetric) should be theoretically zero, which is not (as shown in Fig. 2a). Not only it is not observed, but actually the response of h-BN/Au structures is almost parity independent, which means that there is a second effect as important as the increased reflectivity and enhancement of the electric field amplitude.

As already mentioned in Response#3.1, we have included a detailed discussion on the two ingredients necessary to explain our observations in the new section **“General strategy for enhancing SHG in centrosymmetric and noncentrosymmetric thin materials”**, in Pages 6-8 in the new manuscript and have added a new section **“Effect of SiO₂ Thickness on the SHG Enhancement”** into the supplementary information showing the numerical calculations displaying the SH intensity as a function of the SiO₂ thickness in between the nonlinear material and the substrate/metal film below (i.e. Figure R21).

Comment #13: This one order of magnitude enhancement is also stated for the SPDC measurement (line 237-238), where again the same explanation as above follows, where the main contribution to enhancement is coming from reflection, and has not profound connection (that I can see) to the dynamics of dipolar, surface, or quadrupolar nonlinearity.

Response #13: For NbOCl_2 , which is a noncentrosymmetric material, one can apply same considerations explained in Response #3.2 for the case of odd-number monolayers h-BN. In such a situation both the dipolar (intensity of the electric field) and quadrupolar (amplitude of the electric field and its gradient) moments contribute to the second-order nonlinear response.

Comment #14: A major issue for analysis of SPDC: Line 267-269 the authors say “The observed biphoton emission rate has an inverse relation to the pump power, which is a typical indication of nonclassical photon pair emission [16].” Where is this shown? The authors are just showing CAR as a function of pump power. Those are not the same thing. Also, why are the authors analyzing CAR for studying the polarization dependent response in figs 3(f) and (g)? Fundamentally the CAR should not be polarization dependent. The fact that it is here raises many questions. The effects that the authors are trying to describe would only make sense by using the absolute coincidence rate as the quantity for interpretation. This analysis with CAR is very confusing and raises many questions about why CAR is polarization dependent. CAR is the coincidence to accidental ratio, and having more CAR does not mean a more efficient process.

Response #14.1: Thank you for your comment regarding our analysis of SPDC enhancement. We recognize the confusion caused by our initial description and appreciate your suggestion for clarity. Indeed, our original intent was to show that the correlation shows an inverse relation to the pump power, indicative of nonclassical photon pair emission. This has now been clarified in the revised manuscript. Besides, in the new figure 4 all data (including the polarization dependence) correspond to coincidence- counts rate, except for panel f, where we discuss the CAR dependence on pump power explicitly. The new figure 4 is:

Figure R22 | Characterization of photon pair generation from NbOCl₂/metal heterostructures. *a*, Schematic drawing of the crystal structure of NbOCl₂, denoting the polar axis *b*, nonpolar axis *c*, and out-of-plane axis *a*. Red, oxygen atoms; yellow, chlorine atoms; blue, niobium atoms., Inset: An illustration of the quantum SPDC process, in which a photon (ω_p) incident upon a nonlinear crystal is spontaneously converted into signal and idler photons of lower frequency (ω_i and ω_s). *b*, Coincidence counts rate from 275nm thick NbOCl₂ on gold (orange), NbOCl₂ on SiO₂/Si (blue) and the bare SiO₂/Si substrate (dark blue). *c*, Coincidence rate in H-V basis from NbOCl₂ on gold substrate. H-axis is defined as the polar axis (*b*-axis), and the fundamental wave is also polarized along the H-axis. *d*, Coincidence counts rate from NbOCl₂ on gold film with different wavelength integration ranges. *e*, Power dependent coincidence counts rate from NbOCl₂ on gold film, showing a linear response. *f*, Power dependent coincidence-to-accidental ratio (CAR) from NbOCl₂ on gold film, indicating that a clear correlation peak above the classical limit (CAR > 2) is obtained from the sample. The results in *d-f* are measured under co-polarized configuration. *g*, Coincidence counts rate as a function of the fundamental wave polarization in the co-polarized detection configuration. Points are experimental results, and the broad line is a theoretical fitting.

Response #14.2: We recognize that while CAR itself may depend on polarization, it is clearer to show that the coincidence rates vary with polarization due to the nonlinear optical properties of the materials involved, as we do now in the new Figure 4 (see figure Figure R22 panel g).

For completeness, we derived the polarization-dependent SHG response first, given that SPDC is essentially the reverse process of SFG. Our analysis considers the orientation of crystal axes and the interaction of the pump field with the material, leading to variable SHG and hence SPDC responses based on polarization orientation.

Let us denote the crystal coordinate system as (a, b, c) , where a is the out-of-plane axis, b is the polar axis, and c the non-polar axis. In this frame we can write the generated signal field as:

$$\begin{bmatrix} P_a(2\omega) \\ P_b(2\omega) \\ P_c(2\omega) \end{bmatrix} = \begin{bmatrix} 0 & 0 & 0 & d_{14}^{(2)} & 0 & d_{16}^{(2)} \\ d_{21}^{(2)} & d_{22}^{(2)} & d_{23}^{(2)} & 0 & d_{25}^{(2)} & 0 \\ 0 & 0 & 0 & d_{34}^{(2)} & 0 & d_{36}^{(2)} \end{bmatrix} \begin{bmatrix} E_a(\omega)^2 \\ E_b(\omega)^2 \\ E_c(\omega)^2 \\ 2E_b(\omega)E_c(\omega) \\ 2E_a(\omega)E_c(\omega) \\ 2E_a(\omega)E_c(\omega) \end{bmatrix}$$

where $P(2\omega)$ is the up-converted field at frequency 2ω , and $E(\omega)$ is the excitation field. Assuming that the pump field is normally incident on the crystal, so that the fields are oscillating in the crystallographic bc -plane, we get the following relation for the polarization field along the polar and nonpolar axes:

$$P_b(\omega) = d_{22}^{(2)} E_b(\omega)^2 + d_{23}^{(2)} E_c(\omega)^2$$

$$P_c(\omega) = 2d_{34}^{(2)} E_b(\omega)E_c(\omega)$$

In the laboratory (x, y, z) coordinate system, we can take advantage of the symmetry $d_{23}^{(2)} = d_{34}^{(2)}$. Assuming that the laboratory direction y is rotated by an angle θ with respect to the crystal direction b , our up-converted field in the new coordinate system is:

$$P_z(2\omega) = d_{22}^{(2)} E_b(\omega)^2 \sin\theta + d_{23}^{(2)} E_c(\omega)^2 \sin\theta + 2d_{23}^{(2)} E_b(\omega)E_c(\omega)\cos\theta$$

After that, we transform the pump and the seed fields also into the laboratory coordinates, assuming that the fields are polarized along the z -axis:

$$E_b(\omega) = E_\omega \sin\theta$$

$$E_c(\omega) = E_\omega \cos\theta$$

Finally, we obtain the SHG intensity parallel to the excitation field:

$$I_{SHG}^{//} \sim |P_z(\omega)|^2 \sim \sin^2\theta (d_{22}^{(2)} \sin^2\theta + 3d_{23}^{(2)} \cos^2\theta)^2$$

Since $d_{23}^{(2)}$ is much smaller than $d_{22}^{(2)}$, the overall dominating response is seen from the component of all the fields oscillating parallel to each other, exhibiting a large two-lobed pattern. The peak response is seen at the maximum of the term $\sin^2\theta$, which corresponds to the pump being oriented parallel to the crystallographic b -axis.

The coincidences-to-accidentals ratio can be expressed as $CAR = (C - A)/A = g^2(0) - 1$, where C is the coincidence count rate and A is the accidental coincidence rate. When SPDC is efficiently occurring, the $g^2(0)$ value exceeds 2 due to the presence of true coincidences, which translates to a CAR value greater than 1. When the pump and collection optics are aligned with the b -axis of the crystal (the polar axis in our material system), we observe maximized SPDC efficiency. This alignment maximizes the coincidence rate and, consequently, the CAR value. Conversely, alignment with the c -axis (the non-polar axis) results in reduced efficiency of the SPDC process. Here, the coincidence rates are closer to accidental counts, leading to minimized CAR values.

In any case, we agree with you that showing coincidence count as a function of polarization is more straightforward to understand. We have revised the corresponding figures in the main text as shown above in this response.

In summary, you raised highly valuable and insightful questions, which have given the opportunity to explain the novelty/significance of our work more clearly and to clarify the origin of the enhancement more thoroughly. We have answered them very carefully. We hope that you find our answers thorough, clear and consistent. Below, we summarize the key achievements and the novel contributions of our work, along with the mechanisms underlying the enhancement:

Main Achievements and Novelty:

- (1) Realization of 2-orders of magnitude enhancement of SHG for h-BN and 1-order of magnitude enhancement of SPDC for NbOCl₂ with metal-nonlinear material heterostructures.
- (2) Nonlinear responses become substantially less sensitive to the number of monolayers parity (Even-layer h-BN exhibits similar nonzero SHG responses to odd-layer h-BN, despite its centrosymmetry).
- (3) The enhancement is broadband, spanning a wide range of pump laser wavelengths, which provides a fundamental platform for exploring more interesting phenomena in the future, such as time/frequency entanglement (Commun Phys 6, 278 (2023)).
- (4) Demonstration of the thinnest nonlinear medium as a source of quantum entanglement, furthering the potential for integrated quantum technologies.

The origin of the enhancement:

(1) For Material/Metal configuration:

- (i) The enhancement of $E_{\beta}^{\omega,j}$, $E_{\beta}^{\omega,j}$ is due to a Fabry-Perot (FP) cavity effect, enabled by the high reflectivity of the metal surface. This contributes to both dipole and quadrupole moments.
- (ii) The increased electric field gradient, $\frac{\partial E_{\gamma}^{\omega}}{\partial z}$, originating from the zero phase induced by the metal film, enhances the quadrupole moment.

(2) For Material/SiO₂/Metal configuration:

- (i) In addition to $E_{\beta}^{\omega,j}$, $E_{\beta}^{\omega,j}$ field enhancement, we optimize the position of the h-BN layer by adjusting the SiO₂ layer thickness, allowing for controllable enhancements.
- (ii) Similar to the metal configuration, larger electric field gradient $\frac{\partial E_{\gamma}^{\omega}}{\partial z}$ contributes to the quadruple moment.

Conclusion: The dominant enhancement mechanism in both configurations arises from the enhanced light-matter interaction enabled by engineering the pump field's amplitude, gradient, and phase. This sophisticated manipulation of the optical field is key to the significant improvements we observed in nonlinear responses.

Reviewer #3 (Remarks to the Author):

General comments:

In this manuscript, the nonlinear optical properties of both hBN and NbOC12 flakes are investigated with different substrates, which make distinct nonlinear optical behaviours, especially the nonlinear optical enhancement effect. The authors studied SHG and SPDC for hBN and NbOC12 flakes respectively. They found that the Au substrate show obvious signal augmentation, compared to SiO₂/Si substrate. Further, SiO₂/Au/Si substrate demonstrate outstanding performance, which can be used to enhance nonlinear optical signal for very thin flakes. The theoretical explanation of field amplitude and gradient is reasonable. There are many literatures about the nonlinear optical enhancement effect of van der Waals materials. The enhancement factors of some of them are larger than the results reported here. However, this work is interesting and systematic, which could be considered for publication in NC. My questions and comments are as follows.

General response: Thank you for your thoughtful assessment of our manuscript and the acknowledgment of our “systematic” approach to investigating the nonlinear optical properties of h-BN and NbOC12 flakes on different substrates. We are pleased that the findings regarding signal enhancement on Au substrates and the performance of SiO₂/Au/Si substrates were highlighted as “interesting” and “outstanding”.

Comment #1: The nonlinear optical signal from Au film should be considered. That is, to measure the signal of Au substrate simultaneously, and the enhancement factor needs to be reevaluated.

Response #1: We thank you for this valuable suggestion, and we fully agree. In response to your comment, we have conducted additional experiments to directly measure the SHG response from the Au film (Figure R23). The results indicate that the SHG signal from the bare Au film is significantly smaller -20 and 80 times smaller- than the SHG signal we observed from the h-BN on gold film with h-BN thickness of 36 nm and 65 nm (Figure R24, R25). Given this large difference, we believe the impact of the Au film’s SH contribution on the overall enhancement factor is minimal.

Still, we agree that it will be better to clarify the actual value of the nonlinear gold response, so that there is no doubt that the effect is mainly governed by the nonlinear response of h-BN. Therefore, the quantitative comparison between the nonlinear responses of bare Au or h-BN/Au is now explicitly included in Page 8 of the main text. Meanwhile, the SHG data from the bare Au film and the h-BN films with 36 nm and 65nm thickness on different locations have been added to the supplementary information as Figures S4, S5 and S6.

“...some TMDs [4, 22, 33]. To discard any potential contribution to the SHG signal from the metal surface nonlinearity, SHG was recorded under the same conditions from bare gold. The SHG response from the bare metal is about 20 and about 80 times smaller than the SHG signal from h-BN/gold heterostructures with h-BN thicknesses of 36 nm and 65 nm, respectively as shown by polarization-resolved measurements on these two h-BN flakes / (compare Figures S4, S5 and S6). These findings corroborate that the measured SHG signal indeed originates mostly from the h-BN flakes. Furthermore, since...”

Figure R23: SHG from four locations on bare gold film.

Figure R24: SHG measurement from 36 nm h-BN on Au film.

Figure R25: SHG measurement from 65 nm h-BN on Au film.

Comment #2: Are there possibilities for the proposed substrates to enhance single layer van der Waals material?

Response #2: We sincerely thank you for your interest and for raising this technologically interesting question. As shown in Figure 5e, our simulation results indicate a significant enhancement of over 300 times with monolayer BN using our proposed substrate configuration with an SiO₂ thickness around 64 nm. Additionally, by optimizing the thickness of the SiO₂ layer, we anticipate that similar enhancements can be achieved across a range of TMD materials and that our approach is broadly applicable.

To emphasize this point and encourage other teams to try such architectures, we have begun the discussion about figure 5e highlighting the enhancement possibilities for “vanishingly thin h-BN”. The new text is:

“...For a monolayer-thick h-BN (i.e. for vanishingly thin h-BN with odd number of monolayers), the optimum SiO₂ thicknesses are around ~76 nm and 166 nm, so very close to the resonant condition for the SH waves and close to the condition for a maximum of the electric field pump amplitude (maximizing the dipolar contribution)). Optimizing either ...”

Comment #3: In Fig. 1, why to change the thickness of hBN, which makes the results be not comparable.

Response #3: Thank you for your careful observation regarding the change of thickness of the hBN flake in the original Figure 1, whose information has been now transferred (and enlarged) in the new Figure 2. The choice to change the thickness of h-BN between different panels was to showcase in one figure different situations.

We fully agree that this could bring up confusion to the reader. To prevent it, we have decided to show in the new Figure 2a all the experimental SH data, i.e. for different h-BN thicknesses on gold and on SiO₂/Si. This enables to present all the data on the same footing. Besides, in the new panel Figure 2c we display the polarization-resolved SH intensities for two specific h-BN thicknesses (highlighted in Figure 2a), which showcase the experimental thickness for which the enhancement is maximum (36 nm). It is important to note that the SHG signal from the 36 nm-thick h-BN flake on SiO₂/Si substrate was negligible, as attested by the absence of any polarization-dependent pattern for this h-BN thickness on SiO₂/Si. Because for this thickness the h-BN is just at the level of the background, we decided to add the polarization-dependent response for a second h-BN thickness to show that also on SiO₂/Si substrates the nonlinear response of h-BN displays the six lobes.

Figure R26. SHG enhancement in centrosymmetric-nonlinear material/metal heterostructures. *a*, The experimental and simulated SH intensity for h-BN with different thicknesses on gold films and SiO₂/Si substrates. Symbols are experimental points while solid and dashed lines are nonlinear transfer matrix simulation results. The pump field in the simulation is 890 nm. All data are normalized to the value for 70 nm h-BN on gold. *b*, The SHG of h-BN on gold film under variable pumping wavelengths spanning more than 100 nm. *c*, Polar plot of h-BN SH intensities as a function of the polarization angle of the incident laser on the same 18 nm and 36 nm thick h-BN flake transferred onto a gold film and a SiO₂/Si substrate. The corresponding experimental points are highlighted by a shadowed region in Fig. 2a.

Comment #4: In Fig. 2, the 49 nm thick hBN flake was separated into 4 pieces, not 2 pieces, right?

Response #4: We appreciate your attentive observation and thank you for pointing out the need for clarification regarding our experimental setup. We apologize for any confusion caused by our initial description and appreciate the opportunity to clarify the experimental process and create a new figure.

Actually, the 49 nm thick h-BN flake was separated into 2 pieces. Here, we provide a detailed explanation of the process, which is depicted clearly in Figure R27 below (which is now panel (a) of the new Figure 4 in the main text):

- (1) We initially selected a single h-BN flake with a visible crack at the center.

- (2) We then used a PC (polycarbonate) film to pick up one portion of the h-BN flake.
- (3) The remained piece of the h-BN flake (BN2) was rotated 180 degrees to align it with an antiparallel crystal orientation relative to the h-BN piece on the PC film.
- (4) The rotated BN2 flake was picked up with the same PC film, resulting in two h-BN flakes with antiparallel crystal orientations side-by-side.
- (5) The bottom h-BN flake was then transferred onto a gold-coated SiO₂/Si wafer.
- (6) The top h-BN flakes, still on the PC film, were deposited onto the bottom flake by aligning the edges (parallel to well-defined crystallographic planes) of top and bottom flakes.
- (7) Finally, the PC film was carefully removed to complete the assembly.

To ensure that this procedure is clear, we have added a new detailed illustration in the maintext. This diagram effectively represents the preparation and precise placement of the h-BN flakes, facilitating a better understanding of the methodological intricacies involved. We also added a more detailed description in Page 13 of the main manuscript, explaining the series of steps:

Figure R27: Detailed Process for Preparing the h-BN Heterostructure. It illustrates the step-by-step method used to assemble the h-BN heterostructure, highlighting key stages including initial selection, rotation for alignment, and subsequent layering on a gold-coated SiO₂/Si substrate.

“To do so we investigated the SHG of h-BN homostructures with two different stacking angles on gold and on SiO₂/Si substrates (Figures 3a and 3b). Specifically, we selected a 49 nm thick h-BN flake and separated it into two pieces. One piece of 49 nm h-BN layer was picked up while the other piece of 49 nm h-BN layer left on the initial substrate was rotated by 180° (equivalently, 60°) before stacking it side by side with the unrotated one. This pair of rotated h-BN layers were subsequently and simultaneously transferred on top of a third 46 nm h-BN layer, which had a rotation of 0° with respect to the two top initial layers (Figure 3a). In this way, h-BN homostructures with 0°- and 60°-rotation interfaces can be compared within the same sample and under exactly the same experimental conditions, as evidenced by the polarization-dependent SHG (Figure 3c).

Comment #5: The phenomenon of broadband enhancement is a little bit overemphasized.

Response #5: Thank you for your observation concerning our discussion of broadband enhancement in the manuscript. We value your opinion, as it assists us in maintaining a balanced and accurate discussion of our findings.

In contrast to systems employing multi-resonant metasurfaces, which are limited by the specific resonant modes that can be enhanced, our approach offers a distinct advantage. Besides, broadband capabilities are crucial for a variety of applications that benefit from robust operation across a wide range of frequencies. These include broadband optical parametric amplifiers and supercontinuum generation, which are vital for advancing optical technologies. Moreover, there is growing interest in ultra-thin entangled photon sources for integrated quantum computing and communication devices. Particularly, frequency entanglement, which utilizes the relaxed momentum conservation inherent in thin films, is increasingly recognized for its potential in encoding quantum information. In such systems, enhancing the generation of photon pairs across a broad spectrum of frequencies is crucial.

However, we do note that there are other applications requiring narrow-bandwidth nonlinear optical systems, such as quantum memories. Therefore, a discussion on the application-specific advantages of large and narrow bandwidths, which introduce them in the same footing, is now included in Page 19 of the main text:

“The desired operational bandwidth of a nonlinear optical system is highly dependent on the specific application. For instance, quantum memory of single photons requires an extremely narrow bandwidth to ensure high fidelity and efficiency in photon storage and retrieval. This contrasts with other applications, such as broadband optical parametric amplifiers and supercontinuum generation, which rely on robust operation across a wide range of frequencies and benefit from broader bandwidths. In addition, the growing interest in ultra-thin entangled photon sources, particularly for integrated quantum computing and communication devices, highlights the importance of broadband capabilities. Frequency entanglement, which leverages the relaxed momentum conservation in thin films, is increasingly recognized for its potential in encoding information. In such systems, it is essential to enhance the generation of photon pairs over a broad spectrum of frequencies to maximize entanglement efficiency. Our material-on-metal configurations are particularly well-suited for such broadband operations, with at least 100 nm large bandwidth.”

Comment #6: For SiO₂/Au/Si substrate, the field distribution has a dependence on wavelength. Whether such SiO₂/Au/Si structure also maintains broadband characteristics like Au substrate?

Response #6: Thank you for your inquiry regarding the wavelength dependence of the field distribution and the broadband characteristics of the SiO₂/Au/Si substrate used in our experiments.

In response to your question, we have performed new experiments and investigated the broadband capabilities of the SiO₂/Au/SiO₂/Si structure. Our findings indicate that the SiO₂/Au/SiO₂/Si substrate maintains robust broadband characteristics, effectively supporting a wide range of wavelengths without significant losses in field enhancement, as illustrated in the Figure R28. This broad bandwidth can be attributed to the subwavelength thickness of the active nonlinear material.

Figure R28: Wavelength dependent of SHG measurement from material on SiO₂/Au configuration.

The broadband character is now mentioned in Page 16 of the main text:

“The measured two-orders of magnitude enhancement of the SH signal compared to the same flake deposited directly onto gold (Figure 5d) confirms the efficiency of the SiO₂/Au structure in *enhancing the nonlinear response of very thin h-BN, which still remains broadband (Figure S12).*”

In addition, we included the above new data as Figure S12 in the new section “**Broadband SHG on SiO₂/Au configuration.**” in the supplementary material.

Comment #7: I suggest the authors use picosecond laser to conduct measurements, with higher wavelength resolution.

Response #7: Thank you for your suggestion to utilize a picosecond laser to conduct our measurements with higher wavelength resolution. We appreciate the potential benefits that such an approach could offer in terms of enhanced range in our experiments.

We did indeed explore the use of a picosecond laser during the preliminary stages of our research. However, we encountered a significant limitation regarding the output power of our available picosecond laser system. Specifically, the laser we tested provided an output of only 0.8 mw, which is substantially lower than the 12 to 15 mw of femtosecond laser used in our experiments as reported in the paper.

Due to this much lower power output, we were unable to obtain any measurable signal across a systematic range of thicknesses when using the picosecond laser, making it impractical for our experimental study.

Comment #8: The SEM/AFM images of Au and/or SiO₂ films can be provided, whose flatness may have an impact on the nonlinear optical process.

Response #8: We appreciate your suggestion to include SEM/AFM images of the Au and SiO₂ films. We agree that the surface morphology of these films could significantly influence the nonlinear optical responses we are investigating. To address this, we have added AFM images of both the Au and SiO₂ films to the supplementary materials (Figure R29). These images provide crucial insights into the flatness and surface quality of the films, helping to further elucidate their role in the observed nonlinear optical enhancements. The root mean square roughness value of gold and SiO₂ films is 2.4 nm and 0.6 nm (0.5nm if we exclude the dust particle).

Correspondingly, we have added the following sentences in the main text:

Page 10: *“This observation is tentatively ascribed to the inhomogeneous strain induced by the irregularities of the gold film whose roughness, while in the nanometer scale, is five times larger than that of SiO₂/Si substrates (root mean square roughness of 2.4 nm Vs 0.5 nm, respectively).”*

Figure R29: (a) AFM image of gold film. (b) Height profile of the gold film along the red line indicated in (a). (c) AFM image of SiO₂ film. (d), (e) Height profile of the SiO₂ film along the red(d) and green(e) line indicated in (c). Root mean square roughness is defined as $\sqrt{\langle (h - \bar{h})^2 \rangle}$, where h is the height in each pixel.

Comment #9: As for the writing, there are many long sentences, which lead to difficulty in reading.

Response #9: Thank you for pointing out the readability issues caused by long sentences in our manuscript. We acknowledge that clear and concise writing is crucial for effectively communicating scientific findings. We have thoroughly reviewed the manuscript and revised the text to break down complex and lengthy sentences into more digestible parts. These revisions aim to enhance clarity, improve flow, and make the content more accessible to readers, ensuring that the scientific arguments are presented in an easily understandable manner.

In summary, we wish to sincerely thank you for spending your valuable time and for the insightful comments. We are thrilled to see that you recognized the novelty and the robustness of our work. We found the questions/comments extremely inspiring and greatly helped us to improve our manuscript. We hope that you found our answer clear and thorough.

Reviewer #1 (Remarks to the Author):

General comments:

The authors comprehensively revised the manuscript, including the text and figures. Additional experiments and theoretical analysis are provided to support their claims. The updated results regarding nonlinear enhancement with a metal substrate appear quite interesting. While most comments from all reviewers have been addressed appropriately, the following points need to be clarified before I could recommend it for publication.

General response: We thank you for recognizing the changes we made. We also thank you for thinking that our updated results are “interesting”. We provide herein thorough and clear answers to each of your questions.

Comment #1: The authors discussed several different structures and many different experimental conditions. The physical mechanism and the achieved enhancement are different for these structures. I recommend including a table summarizing the features of these structures, including the nonlinear enhancement factors, physical mechanisms, benefits, and limitations.

Response #1: We sincerely thank you for making this suggestion, which will help the readers apprehend the novelty of the work and will give an overall view of the quantitative results. We entirely agree with you that the features of structures, nonlinear enhancement factors, physical mechanisms, benefits and limitations deserve to be more clearly summarized in a table.

Table R1 provides a comprehensive summary of all the SHG and SPDC processes discussed in our work, along with the corresponding sample structures used to demonstrate them. For all h-BN samples with an odd number of layers we attribute the observed enhancement to the combined contributions of both dipolar and quadrupolar moments. In contrast, for h-BN samples with an even number of layers, where inversion symmetry is preserved, the enhancement primarily originates from the quadrupolar moment. One of the key advantages of this approach is its flexibility in excitation wavelengths, compatibility with various structural configurations, and the ability to achieve large-area enhancement.

In response to your suggestion, we have now included Table R1 in the revised manuscript, which systematically summarizes the properties of the different structures under investigation. We have also cited the table in the conclusion section: “...By engineering the light component we achieved a substantial boost in the light-matter interaction resulting in a notable increase in the nonlinear response, as observed for both SHG and SPDC (Table S6). For centrosymmetric materials like h-BN, which typically exhibit lower second-order...”. The table is reproduced below:

Structures	Parity of nonlinear material layers	Process	Enhancement factors compared to SiO ₂ /Si (Experiment)	Physical mechanisms	Benefits	Limitations
h-BN/Au	Odd	SHG	Up to 738	Dipole, Quadrupole	Broadband excitation, compatible with other enhancements (e.g., twisted structures), large-area enhancement	Not suitable for few-layer materials
h-BN/Au	Even	SHG	Up to 27	Quadrupole		
NbOCl ₂ /Au	-	SPDC	10	Dipole, Quadrupole		
Twisted h-BN/Au	-	SHG	357	Symmetry breaking, Dipole, Quadrupole		-
h-BN/64nm SiO ₂ /Au	Odd	SHG	238	Dipole, Quadrupole		Requires specific SiO ₂ /nonlinear material thickness for optimal enhancement
NbOCl ₂ /SiO ₂ /Au	-	SHG	300	Dipole, Quadrupole		
NbOCl ₂ /SiO ₂ /Au	-	SPDC	Up to 100	Dipole, Quadrupole		

Table R1: Summary of the key characteristics of different structures, including nonlinear enhancement factors, physical mechanisms, benefits, and limitations.

Comment #2: In Line 117, the authors provide an equation describing the nonlinear polarization with contributions from dipole and quadrupole moments. Can the authors provide references for this equation? Additionally, have previous works in any nonlinear system proposed other approaches to enhance the quadrupolar term?

Response #2: We sincerely appreciate your insightful comment regarding the nonlinear polarization equation provided in Line 117. The references for the equation are *Yao et al., Sci. Adv. 2021; 7 : eabe8691* and *Nat. Phys. 17, 777–781 (2021)*. We have added these references as Ref.23 and Ref. 37 into the corresponding sentences in our main text.

Regarding previous studies exploring quadrupolar enhancement in nonlinear optical systems, various strategies have been investigated: Yu Zhang et. has demonstrated SHG response in centrosymmetric graphene, where SHG is usually forbidden under the electric dipole approximation (Physical Review Letters, 2019. 122(4): p. 047401). In this work the inversion symmetry is broken by optical dressing the pumping photon with an in-plane photon wave vector i.e. by using oblique incidence excitation. Fu Xiang et al. has used two-beam SHG technique to enhance the nonlinear response of metal films (New Journal of Physics, 2010. 12(6): p. 063009). In this case the effective quadrupolar nonlinear response originates from field retardation across the surface, whose random roughness introduces unequal retardation for beams impinging the sample from different orientations. While these approaches rely on specialized excitation techniques, our strategy focuses on simple designs that maximize both dipolar and quadrupolar contributions in a straightforward manner.

In response, we have modified the main text with the following sentences:

“Previous studies have demonstrated quadrupolar enhancement of SHG by using optical dressing with an in-plane photon wave vector (i.e. by breaking inversion symmetry by oblique incidence excitation) [38] and a two-beam SHG technique on a rough metals surface, on which unequal retardation effects for each beam breaks the initial symmetry [39].”

Comment #3: The authors mention that the enhanced quadrupolar response in non-centrosymmetric materials can be 2-3 times smaller than the dipolar response. Does this indicate that the proposed approach to enhancing the nonlinear response is more suitable for centrosymmetric materials?

Response #3: Thank you for your insightful question. We believe that the proposed approach is suitable for both centrosymmetric and non-centrosymmetric materials, as it enhances both dipolar and quadrupolar responses. Obviously, its impact is stronger on centrosymmetric materials for which the quadrupolar term is the dominant one in the polarization expansion.

To clarify further, we discuss the effects separately for h-BN stacks with an even or odd number of monolayers:

(1) **For even parity h-BN:** Even-layer h-BN retains centrosymmetry, similar to bulk h-BN that, yet, exhibits a weak SHG signal. This is entirely attributed to the nonzero quadrupolar moment, as the dipole moment vanishes due to centrosymmetry. In our approach, the quadrupolar moment is enhanced through two mechanisms:

- i) Increasing the gradient of the electric field $\frac{\partial E_Y^\omega}{\partial z}$.
- ii) Amplifying $E_\beta^{\omega,j}$ within the material.

These enhancements, which contribute to both mechanisms (i) and (ii) in our framework, are demonstrated in Figure 2a of the main text.

(2) **For odd parity h-BN:** Both the dipole and quadrupolar moments contribute to the SHG signal. Structurally, this system consists of two adjacent layers (like an even-layer material) plus an additional single layer:

- i) In the two-layer configuration, the dipole moment cancels out due to symmetry, while the quadrupole moment remains nonzero.
- ii) In the single-layer configuration, the dipole moment is present, but the quadrupole moment is zero.

As a result, the total nonlinear polarization $P_{\alpha}^{2\omega}$ can be expressed as $P_{\alpha}^{2\omega,d,SL} + P_{\alpha}^{2\omega,q,2L}$, where SL denotes one single layer and 2L denotes two adjacent layers. By placing h-BN on a gold substrate, enhancements are observed in all E_{β}^{ω} , E_{γ}^{ω} and $\frac{\partial E_{\gamma}^{\omega}}{\partial z}$, effectively boosting both dipole and quadrupole contributions. Note, however, that while for even parity h-BN one should try to maximize both, amplitude and gradient, for odd parity, one might prefer to minimize the quadrupolar term if this is not in-phase with the dipolar one. This is why in the main text we state “Thus, to enhance the SHG response of even number of monolayers h-BN (i.e. centrosymmetric) by one to two orders of magnitude, we need to sacrifice a factor $\sim 2-4$ in the SHG signal of odd number of monolayers h-BN (i.e. non-centrosymmetric).”

Comment #4: In Line 183, the authors claim that the dipolar enhancement can reach two orders of magnitude for an odd number of monolayers. Why is the enhancement so large for just a reflection-based mechanism?

Response #4: We sincerely thank you for raising this important question (also raised by the second referee in the first review round). In fact, this potential “two orders of magnitude enhancement of the intensity” (so one order of magnitude enhancement of the amplitude) is not so much related to the reflectivity of the gold, but rather to the fact that the second harmonic signal of h-BN on SiO₂/Si nullifies for certain h-BN thickness close to about 30 nm, so that the enhancement ratio can be up to two orders of magnitude (in fact, the SHG being zero for such h-BN thicknesses on SiO₂ was also reported in reference 23 of the main text, K. Yao *et al.*, “Enhanced tunable second harmonic generation from twistable interfaces and vertical superlattices in boron nitride homostructures”. *Science Advances*, 2021. 7(10): p. eabe8691.) This is also the main reason why after the first round of revision we decided to plot the SHG intensity in Figure 2(a) rather than the SHG intensity ratios, as suggested by the second referee, to prevent any misunderstanding or misinterpretation of our results.

In fact, if we calculate the gold-to-SiO₂/Si enhancement factor corresponding to the dipole contribution shown in the new Figure 1c of the main text, we readily see that this enhancement factor is mainly governed by the nullifying of the response on SiO₂/Si.

Figure R1: The enhancement factor of second harmonic electric field from h-BN on gold film compared to SiO₂/Si.

To be sure that the reader does not get the wrong impression that there is some hidden effect or some unrealistic artifact, we have explicitly indicated in the text that the enhancement goes from about one order of magnitude to up to two orders of magnitude, but that for the particular case of odd number of monolayers the reason for such strong enhancement is related to its poor response on the standard SiO₂/Si configuration.

The new text is:

“On the other hand, for odd number of monolayers, the potential enhancement if only the dipolar contribution was considered would be typically ~~two~~ one to two orders of magnitude (attaining its maximum for an h-BN thickness of about 30 nm, for which the dipolar response on SiO₂/Si approaches zero). However, due to the out-of-phase quadrupolar contribution (see Figure S2), the overall enhancement is rather a factor 20-30 ~~100~~.”

Comment #5: In Fig. 2c, the polarization dependence of SHG in the hBN-on-SiO₂/Si platform looks totally different for 18 nm and 36 nm thicknesses. What is the underlying physical mechanism behind this difference? Are these results for odd layers or even layers?

Response #5: Thank you for your insightful question. We know that the SHG intensity in hBN is thickness dependent, and that for h-BN layers (with odd number of monolayers) around 30 nm thick on SiO₂/Si the SHG intensity decreases by one order of magnitude compared to layers 20 nm or 40 nm thick. In Figure 2a, three of the experimental points corresponding to h-BN on SiO₂/Si lie on this thickness region, where the SHG intensity is so low that its value is near or even within the detection background of our current measurements. To showcase this low SHG intensity, we decided to show the polarization-resolved of one of these samples (the 36nm tick h-BN on SiO₂/Si), where the six lobes can be hardly distinguished (if seen at all). On the other hand, as soon as we go away from this minimum of SHG intensity, even for thinner layers, we recover the six lobes on the polarization resolved measurements of h-BN on SiO₂/Si substrates. For the samples of h-BN on gold, all samples display the six lobes in the polarization-resolved measurements.

Based on Figure 1a, we attribute both the 18 nm and 36 nm h-BN samples to odd-layered structures.

To clarify this in the manuscript, we have revised the Figure 2c caption:

“...The corresponding experimental points are highlighted by shadowed regions in Fig. 2a, where we assume that both flakes have an odd-layer structure...”

Comment #6: In Line 253, the authors show that the enhancement in twisted hBN layers can reach an enhancement of 357 times, which is very impressive by using gold only. However, the physical mechanism is not well described. Is this enhancement compared to the off-gold AB stack? For the AB stack where the inversion symmetry is broken, the dominant contribution should be dipolar. How does the gold bring such a high enhancement?

Response #6: We sincerely appreciate your insightful question and apologize for not clearly stating the physical mechanism behind the observed enhancement.

The reported 357-fold enhancement is compared to the off-gold AA' stack. This enhancement arises from two key factors: (1) the additional dipole moment originating from the AB stack interface, and (2) the enhanced bulk effects (both dipolar and quadrupolar contributions) due to the gold substrate.

To clarify further:

- (1) When comparing the AB stack off-gold to the AA' stack off-gold, the enhancement factor is approximately 8 times, which primarily results from the interface SHG contribution of the AB stack.
- (2) When placing the AB stack onto the gold film, both the electric field magnitude and the electric field gradient across the structure are significantly enhanced. These are bulk effects that contribute additional enhancement beyond the interface-driven SHG from the dipole moments.

The mechanism of gold film enhancement in this structure follows the same principles as in other cases discussed in our work. Importantly, this result demonstrates that gold-enhanced SHG can be effectively combined with other enhancement mechanisms, such as the twisting of layered materials, where interface SHG can also be generated.

We appreciate your question, which has helped us refine our explanation, and we have updated the manuscript accordingly to clarify this point:

“In addition to the enhanced bulk effects from dipolar and quadrupolar contributions -arising due to the increased electric field magnitude and field gradient-, the additional twisted interface-related SHG further amplifies the overall SHG response.”

Comment #7: In Fig. 1c, the authors present the SHG contributions of dipole and quadrupole moments for the gold substrate. I recommend including the results for the SiO₂ substrate for a direct comparison.

Response #7: We sincerely appreciate your valuable suggestion. We agree that providing a direct comparison between the gold substrate and the SiO₂ substrate would enhance the clarity of our discussion and further support our conclusions.

In response, we have now included the SHG contributions of dipole and quadrupole moments for the SiO₂ substrate in Fig. 1c of the revised manuscript.

Figure R2: The simulation results of dipolar (green) and quadrupolar (orange) contributions to the magnitude of SHG electric field of h-BN flakes with odd number of monolayers on gold (line) as well as SiO₂/Si (dashed line). Dip: dipolar; Quad: quadrupolar.

Interestingly for the reader, by adding the dipolar and quadrupolar responses on SiO₂ one can more easily understand the origin of the maximum intensity enhancement, which could sound otherwise too large (cf. Comment 4).

Comment #8: What is the operating bandwidth for the SiO₂/Au/SiO₂/Si structure where the Fabry Perot effect is involved? Is the operating bandwidth reduced as compared to the hBN-on-Au structure?

Response #8: The operating bandwidth of the SiO₂/Au/SiO₂/Si structure, where the Fabry-Pérot effect is involved, exceeds 100 nm (Figure R3).

Importantly, the operating bandwidth is not reduced compared to the h-BN-on-Au structure, at least within the 100 nm of pump wavelength that we are able to test, as shown in Figure R3. As explained in the supplementary information, this is because the actual cavity is a low-Q one and the metal displays a relatively large reflectivity over a very broad spectral range, in particular both at pump and SHG wavelengths.

Figure R3: (a) Experimental wavelength dependent SHG measurements from material on SiO₂/Au configuration. (b) Simulated wavelength dependent SHG measurements from material on SiO₂/Au and SiO₂/Si substrates for an SiO₂ thickness of 64 nm. Inset: the enhancement of SHG of h-BN on SiO₂/Au compared to h-BN on SiO₂/Si.

Comment #9: Please correct the typos in citing the figures: (1) Line 203, Fig. 1c -> Fig. 2c. (2) Line 412, Fig. 4f -> Fig 5f.

Response #9: We sincerely appreciate your careful attention to detail and for pointing out the citation errors in the manuscript. We have carefully corrected the typos in the main text already.

Reviewer #2 (Remarks to the Author):

General comments:

The authors have presented very detailed responses to my questions, including new results and new explanations added to the manuscript, and have resolved my main concerns about the manuscript. The results and the main message of the paper are now much clearer. I am now convinced that the results have novelty and will be of interest to the field. Hence, I can now support the publication of this work in Nature Communications. I only have a few comments that I think the authors should consider in their final version.

General response: We sincerely appreciate your thoughtful feedback and your positive evaluation of our manuscript. We are grateful for the time and effort taken to provide such detailed and constructive comments, which have significantly helped improve the clarity, depth, and presentation of our work. We are pleased to hear that our revisions have addressed your main concerns and that the novelty and impact of our results are now clearer. We also appreciate your support for the publication of our work in Nature Communications. We have carefully considered the remaining suggestions and have incorporated them into the final version of the manuscript.

Comment #1: In line 83 of the manuscript the authors say „hBN, which is the most widespread 2D material today“. I think such statements are rather subjective. Taking out the word „most“ will resolve the problem.

Response #1: We agree that the phrase "the most widespread 2D material" may be subjective. To address this concern, we have revised the statement by removing the word "most", ensuring a more neutral and precise description.

Comment #2: In line 158, the authors said „Furthermore, because the global enhancement of dipolar and quadrupolar contributions described above stems from two general metal properties“. It is not clear here what the two general metal properties are.

Response #2: We appreciate your comment and agree that the statement in Line 158 was unclear. Although we have discussed these two metal properties in the section “General strategy for enhancing SHG in centrosymmetric and noncentrosymmetric thin materials”, we recognize the need to restate them explicitly for clarity.

To address this, we have now explicitly summarized the two general metal properties in Line 158 as: 1) High reflectivity over a broad wavelength range, which enhances optical interactions. 2) Strong electric field gradient near the metal, which amplifies both dipolar and quadrupolar nonlinear effects.

In response, we have revised the main text as follows:

“Furthermore, because the global enhancement of dipolar and quadrupolar contributions described above stems from two general metal properties—high reflectivity over a broad wavelength range and the ability to generate a strong electric field gradient near the metal—its effect is observed when depositing the nonlinear active material on a variety of metals (Figure S1, S16, and S17).”

Comment #3: In the last paragraph of the „Methods“ section „Sample preparation“, lines 468-473, I think the figure numbers should be updated. I think the description there refers to the figure numbers from the previous version of the manuscript.

Response #3: We sincerely appreciate your thorough review, which has helped improve the accuracy and clarity of our work. In response, we have carefully reviewed and corrected the figure numbers to ensure consistency with the revised manuscript. These updates have been implemented in the final version:

“As determined by AFM, the thickness of gold/titanium substrate used in Figures 2, 3 and 4 is 200 nm/18 nm, and the thickness of SiO₂ in the SiO₂/Au heterostructure of Figure 5 is 64 nm. The primary role of the Ti layer is to act as an adhesion promoter, ensuring a strong bond between the gold film and the underlying substrate. The thickness of SiO₂ layer on SiO₂/Si substrate is 285 nm. The thickness of NbOCl₂ shown in Figure 4 is 275 nm while in Figure 5 it is 16 nm.”

Comment #4: In lines 3-5 for page 6 of the Supplementary file, the authors say „This enhancement is demonstrated in Figure 2c of the main text. This enhancement is demonstrated in new Figure 2a of the main text.“ I think these sentences need to be updated for the new figure numbers. Probably there is also a repetition in sentences, and also the word „new“ in there is probably unnecessary. Overall, I suggest the authors look at the figure numbers in the texts again, to make sure there is no inconsistencies with the many new added figures.

Response #4: Thank you for highlighting this inconsistency in our Supplementary file. We acknowledge the need to update the figure references and refine the wording to improve clarity. To address this, we have revised the sentences to: “This enhancement is demonstrated in Figure 2a of the main text.”

Additionally, we have conducted a thorough review of the manuscript and Supplementary Information to ensure that all figure references are accurate and consistent.

Comment #5: After equation (30) in the supplementary, the authors say „The propagation wavevector of matrices Φ for 3ω is $3k_0$ and the refractive indexes need to be reevaluated accordingly,“. I am not sure where the third harmonics are coming into this calculation. If it is an error then it should be corrected. If not, maybe a short explanation helps, since I cannot see where the third harmonic is getting into the calculations.

Response #5: Thank you for pointing out this discrepancy. We acknowledge the typo and appreciate your careful observation. The correct term should indeed be 2ω and $2k_\omega$, rather than 3ω and $3k_0$. We have corrected the corresponding text in the Supplementary Information as follows:

“The propagation wavevector of matrices Φ for 2ω is $2k_\omega$ and the refractive indexes need to...”

Comment #6: In line 508-510, the definition for calculating the lower bound for the enhancement factor of SPDC seems rather an unusual definition as an enhancement factor. It would be good if the authors can add a short explanation there about why this definition is used for enhancement factor and what it means.

Response #6: We appreciate the reviewer’s insightful question regarding the definition of the enhancement factor for SPDC. The standard enhancement factor is typically defined as: $\frac{C_{real}^{on\ gold}}{C_{real}^{off\ gold}}$, where C_{real} represents the real coincidence counts, calculated as $C - \langle A \rangle$, with C being the measured coincidence counts and $\langle A \rangle$ the average of accidental counts.

However, in our case the SPDC signal from the material off the gold film is at or below the background level. Because of this, the standard enhancement factor as defined above cannot be directly applied, as the denominator approaches zero (if we correct it for the background level) and then we would be pushed to claim a giant enhancement. To prevent this, we define a lower bound for the enhancement factor using the signal-to-noise ratio (SNR), ensuring that the actual enhancement factor is bounded at least by our accidental counts background.

Specifically, the signal is defined as $(C^{on} - \langle A \rangle^{on})$, which is the real coincidence counts while the noise is defined as the standard deviation of the accidental counts (background). We compared the standard deviation of the accidental counts for both material on gold and on SiO_2/Si . The results indicate that the calculated lower bound of the enhancement factor is approximately 100 in both cases.

In response, we have revised the Methods section to explicitly state this definition:

“The lower bound of the enhancement factor for SPDC is determined using the signal-to-noise ratio (SNR), as the SPDC signal from the material off the gold is at or below the background level. Specifically, it is calculated as: $(C^{on} - \langle A \rangle^{on})/\delta A^{on}$ and $(C^{on} - \langle A \rangle^{on})/\delta A^{off}$, where C is the coincidence counts per second and A is the accidental counts per second for on and off structures, $\langle \rangle$ means mean-value and δ represents the standard deviation. Using this approach, the calculated lower bound values of the enhancement factor are 93 and 153, respectively, confirming a significant enhancement of the SPDC signal.”

Comment #7: In line 69 of the main text, the authors say „See table S2“. I think here they meant table S1, which makes sense compared to the context of the sentences before. I suggest to the authors to carefully examine all these naming for tables and figures in the text to make sure they are referred to correctly.

Response #7: We appreciate your careful attention to detail and for pointing out the incorrect reference in Line 69 of the main text. Upon review, we confirm that the correct reference should be "Table S3" instead of "Table S2", as it accurately corresponds with the preceding discussion.

To ensure consistency throughout the manuscript, we have also conducted a thorough review of all tables and figure citations in both the main text and Supplementary Information, making any necessary corrections.

Comment #8: I think the formula (1) in the supplementary is from the following reference [<https://doi.org/10.1364/OL.44.005792>], where as I understand they had hBN on SiO₂ substrate. I do not think it is valid to use that formula for evaluating χ^2 values for systems of 2D materials on different types of substrates, especially ones like in this work that show extra field enhancement effects. I assume the data now in table S2 for χ^2 values of hBN and NbOCl₂ are based on this formula. If so, maybe the authors can say more clearly in that first supplementary section with formula (1), which of their experimental data for hBN and NbOCl₂ are actually used to find these χ^2 value. I am not sure if this formula is meant exactly for finding the bulk χ^2 , since it is used in [<https://doi.org/10.1364/OL.44.005792>] for finding the surface χ^2 for hBN. So, the authors should explain how they derived bulk χ^2 values for table S2 based on formula (1) (maybe state if they used an effective thickness) and state explicitly which of their experimental data they are using for this task.

Response #8: We appreciate your critical assessment and fully agree that Formula (1) may not be directly applicable for evaluating second order susceptibility values for 2D materials on different substrate types. We also apologize for any misunderstanding caused by our previous explanation.

To clarify any potential misunderstanding: the $\chi^{(2)}$ values of h-BN and NbOCl₂ listed in Table S2 are not derived from our experimental data but are instead taken from the following literature sources: Optics Letters 44.23 (2019): 5792-5795; Physical Review B 89.8 (2014): 081102; Nanophotonics 13.21 (2024): 4029-4035; Nature 613.7942 (2023): 53-59. These values were used in our simulations and these materials were selected to illustrate the enhancement mechanism in both centrosymmetric and non-centrosymmetric materials.

To ensure clarity, we have revised the Supplementary section “Second Harmonic Susceptibility” to explicitly state the purpose of Table S2 and our methodology for selecting these values:

“Our selection of h-BN and NbOCl₂ is intentional, as they represent two distinct nonlinear regimes. In centrosymmetric materials like h-BN, the second-order susceptibility is inherently constrained by its structure (Table S2). For even-layered h-BN, dipolar contributions vanish due to symmetry constraints, making the quadrupolar contribution dominant. This highlights the potential of methods designed to enhance quadrupolar nonlinear processes. Conversely, in non-centrosymmetric materials like NbOCl₂, the dipolar contribution is typically dominant, and the second-order susceptibility is intrinsically larger (Table S2). Our method effectively enhances this response, similar to approaches using microcavities or metasurfaces.

Additionally, our approach provides broadband enhancement, which is particularly beneficial for ultra-thin entangled photon sources such as NbOCl₂. This broadband response enables the generation of frequency-entangled photon pairs, thereby increasing the number of available quantum communication channels from a single source.

In our simulations, we used a second-order susceptibility of 42 pm/V for h-BN and 400 pm/V for NbOCl₂. To further facilitate a direct comparison with existing literature, we have benchmarked our nonlinear materials by comparing their second-order susceptibility values, as presented in Table S2.”

Comment #9: The caption for figure S11 in the supplementary needs to be updated. It refers to a figure in the main text that seems not to be there now. Also, it compares to the result on gold film, which is not shown in the same figure, and it would be good to remind the reader in that caption where that result is.

Response #9: Thank you for your suggestion and apologize for the mistakes. We have added the result on gold film in Figure S11 for a clear comparison (Figure R4) and revised the captions.

Figure R4: Simulation for SH Intensity emanating from even-layered h-BN on a S-A structure and gold film utilizing nonlinear transfer matrix calculations. A marked enhancement can be seen in signal strength for h-BN flakes comprising an even number of monolayers on S-A structure (a), when compared to the

results obtained on the gold film (b). The simulation corresponding to odd number of layers is shown as Figure 5b in the main text.

Comment #10: Caption of figure S12 says „Wavelength dependent of SHG measurement from material on SiO₂/Au configuration.“ The authors should mention in this caption which material is that, hBN or NbOCl₂, and for which SiO₂ thickness.

Response #10: Thank you for your valuable suggestion. We have revised the caption to provide a clearer description of the material and SiO₂ thickness. The updated caption now is:

“Wavelength dependence of SHG measurement from 8 nm h-BN on a SiO₂/Au configuration, where the SiO₂ thickness is 64 nm.”

Comment #11: In caption of figure S13, for (a), is that a microscope image? That should be mentioned in the caption.

Response #11: Thank you for your suggestion. We apologize for this missing information. To clarify, Figure S13(a) is a microscope image, and *we have revised the caption accordingly to:*

“Optical microscope image of a 17 nm h-BN flake (in the blue box) on a S-A substrate with an extra monolayer transferred onto it (in the red curve).”

Comment #12: In caption of figure S17, for (b), the authors should mention what is the dielectric and what is its thickness.

Response #12: Thank you for pointing this out. We recognize the importance of specifying the dielectric material and its thickness for clarity. In response, *we have updated the figure caption to explicitly mention this information:*

“Simulations of SHG intensity as a function of thickness of h-BN from flakes on metallic (a) and SiO₂/metallic stack (b) substrates, where the SiO₂ layer has a thickness of 64 nm.”

Reviewer #3 (Remarks to the Author):

General comments:

As it has been improved a lot, I recommend that this manuscript be accepted for publication in Nature Communication.

General response: We sincerely appreciate your thorough evaluation and constructive feedback, which have significantly improved the quality and clarity of our manuscript. We are especially grateful for the positive recommendation.